# Inadvertent human genomic bycatch and intentional capture raise beneficial applications and ethical concerns with environmental DNA

Liam Whitmore[1,2,6], Mark McCauley [1,3,6], Jessica A. Farrell[1,4,6], Maximilian R. Stammnitz [5], Samantha A. Koda [1], Narges Mashkour[1], Victoria Summers[1], Todd Osborne [1], Jenny Whilde[1] & David J. Duffy [1,4] ✉

The field of environmental DNA (eDNA) is advancing rapidly, yet human eDNA applications remain underutilized and underconsidered. Broader adoption of eDNA analysis will produce many well-recognized benefits for pathogen surveillance, biodiversity monitoring, endangered and invasive species detection, and population genetics. Here we show that deep-sequencing-based eDNA approaches capture genomic information from humans (*Homo sapiens*) just as readily as that from the intended target species. We term this phenomenon human genetic bycatch (HGB). Additionally, high-quality human eDNA could be intentionally recovered from environmental substrates (water, sand and air), holding promise for beneficial medical, forensic and environmental applications. However, this also raises ethical dilemmas, from consent, privacy and surveillance to data ownership, requiring further consideration and potentially novel regulation. We present evidence that human eDNA is readily detectable from 'wildlife' environmental samples as human genetic bycatch, demonstrate that identifiable human DNA can be intentionally recovered from human-focused environmental sampling and discuss the translational and ethical implications of such findings.

The field of environmental DNA (eDNA) research has been rapidly expanding in recent years, resulting in unprecedented advances in a range of biological monitoring applications. Environmental DNA research provides a non-invasive and cost-effective approach for the study and management of wild populations and invasive species, by using a forensics approach to the extraction and identification of DNA fragments released as organisms travel through and interact with the environment[1–7]. Environmental DNA analysis is also being applied to issues of human and animal health—for example, in pathogen, parasite and pollen monitoring[1,8–10]. This includes the rapidly emerging field of human eDNA-based pathogen detection from human wastewater. Such approaches developed quickly during the early stages of the COVID-19

[1]Whitney Laboratory for Marine Bioscience and Sea Turtle Hospital, University of Florida, St. Augustine, FL, USA. [2]Department of Biological Sciences, School of Natural Sciences, Faculty of Science and Engineering, University of Limerick, Limerick, Ireland. [3]Department of Chemistry, University of Florida, Gainesville, FL, USA. [4]Department of Biology, College of Liberal Arts and Sciences, University of Florida, Gainesville, FL, USA. [5]Centre for Genomic Regulation (CRG), Barcelona Institute of Science and Technology, Barcelona, Spain. [6]These authors contributed equally: Liam Whitmore, Mark McCauley, Jessica A. Farrell. ✉e-mail: duffy@whitney.ufl.edu

pandemic and have already been repurposed for other pathogens such as monkeypox, poliovirus and tuberculosis[1,11–15]. Environmental DNA has been successfully obtained from a range of sample types including air, soil, terrestrial and aquatic sediments, water (marine, freshwater and wastewater), permafrost, snow and ice cores[10,16,17].

Environmental DNA research has traditionally relied primarily on targeted methodologies, such as quantitative PCR (qPCR) and metabarcoding-based next-generation sequencing, and early applications focused on bacterial communities[18]. However, continued improvements in deep sequencing technology and novel bioinformatics refinements mean that untargeted shotgun-sequencing-based approaches are becoming feasible (Extended Data Fig. 1a), which more fully capture the true extent of genetic diversity within a sample[1,8,19]. Shotgun sequencing is set to become more labour- and cost-effective than qPCR or metabarcoding in the near future while providing the least biased biodiversity assessments, thus providing the broadest possible presence and abundance information across all taxa. We have recently shown that untargeted shotgun deep sequencing (the direct sequencing of total eDNA with no prior enrichment or selection) can provide both host and pathogen sequence data[8,17], while also simultaneously capturing all other biodiversity within an environmental sample. Similar to biodiversity assessments, shotgun sequencing of wastewater samples could be applied to monitor all human pathogens simultaneously but would also probably capture a large volume of human genomic data.

While there is a plethora of beneficial applications of eDNA, we postulate that an unintended negative consequence of eDNA approaches might be the capture of human genomic information (human genetic bycatch (HGB); Fig. 1a). Beneficial applications of human-focused eDNA sampling can also be envisaged. Currently, human DNA is rarely (if ever) the intended target of eDNA studies, leaving the field with a lack of specific human-related regulatory guidelines or ethical approvals. Current targeted qPCR and metabarcoding-based eDNA approaches do not recover any substantial human genomic information. However, as eDNA shifts towards shotgun sequencing, potentially large volumes of human eDNA will be retrieved, including sufficient data to identify and phenotype human individuals. Obtaining genetic data from identifiable persons requires informed consent[20]. Legal and ethical frameworks are common in studies involving humans and studies that generate patient data, albeit with continued debate regarding whether such policies are sufficiently rigorous in relation to informed consent, data ownership and data protection[20–24].

To ascertain whether human genomic DNA could be harvested from eDNA data, we aligned the sequencing data previously generated[8,17] as part of our wildlife and pathogen eDNA projects against the human reference genome. Having demonstrated the occurrence of HGB, we next applied species-specific qPCR to quantify the level of human eDNA in environmental water samples from sites distant from and close to human habitation, from human footprints in beach sand and from occupied and unoccupied room air (Supplementary Fig. 1). Finally, we applied long-read shotgun sequencing and short-read sequencing human exome enrichment to obtain human-aligning sequences to reconstruct informative human haplotypes (genetic ancestry and mutations) from eDNA.

## Results

Human-aligning reads were detected in all samples (Fig. 1b and Supplementary Table 1) of untargeted shotgun deep sequencing from water and sand eDNA generated for wildlife and pathogen monitoring[8,17]. Furthermore, in some wild (non-rehabilitation tank) water samples, human-aligning reads were detected at levels almost as high as that of our main study species, the green sea turtle[17] (green sea turtle reads per ten million total reads (RPTM)/human RPTM per wild water sample average ratio, 1.39; minimum ratio, 1.03; maximum ratio, 2.54). As expected, the rehabilitation sand and tank water samples had more

sea turtle eDNA than human eDNA present (turtle RPTM/human RPTM ratio, 63.50 and 18.50, respectively). The high rate of human eDNA recovered from wild samples (Fig. 1b and Supplementary Table 1) is particularly relevant as our study sites were not directly adjacent to areas of dense human habitation (that is, cities or towns). The relative paucity of human-aligning eDNA reads in the water negative field control and the rehabilitation tank samples (Supplementary Table 1) confirms that the vast majority of human-aligning reads did not originate from contamination during sample processing. In total, 1.8 million paired-end human-aligning reads (300 base pairs (bp) per read pair) were recovered as by-catch from this eDNA study (Fig. 1b and Supplementary Table 1).

The human Y chromosome is a fast-evolving reduced chromosome, which is divergent in structure and gene content from even humans' closest extant relative, the chimpanzee[25]. Y chromosomes are not shared among all vertebrate species; they are specific to therian mammals. Sea turtles (our target study animal) do not possess Y chromosomes, instead having temperature-dependent sex determination. The human Y chromosome is therefore a useful genome region for confirming the presence of genuine human reads in these complex metagenomic eDNA sequencing samples. Human Y chromosome genomic bycatch reads were detected in all samples (Extended Data Fig. 1b), despite the Y chromosome's small size and the default exclusion of reads originating from human females. Estuarine samples tended to have higher human Y chromosome RPTM values than oceanic samples (Extended Data Fig. 1b).

Having determined that HGB can occur from wildlife eDNA sampling, we next investigated the feasibility of intentional human eDNA recovery from sampling sites of suspected high and low human eDNA release. Species-specific qPCR assays for the quantification of human eDNA revealed that human DNA was readily recoverable from water sites located close to towns (Figs. 1c and 2 and Extended Data Figs. 1c, 2 and 3). These findings were replicated in both subtropical Florida and temperate Ireland. Minimal human eDNA was detected in a mountain tributary (Goldmines River) of the Avoca River, Co. Wicklow, Ireland, close to its source and above the line of human habitation (Fig. 1c, Extended Data Fig. 1c and Supplementary Table 2). In contrast, human eDNA was detected once the Avoca River enters Arklow town, with human eDNA levels in the river increasing as it flows through the town, before being diluted when the river enters the Irish Sea (Fig. 1c and Extended Data Figs. 1c and 2a,b). More human eDNA was present in the samples within the town even though less than half of the volume of water could be filtered compared with the non-town samples, due to increased filter clogging induced by more turbid water (Supplementary Table 2). Similarly, no human eDNA was detected from a private well, above the human habitation line (Fig. 1c and Extended Data Fig. 1c). Non-human eukaryotic eDNA was recovered from all Irish samples, demonstrating successful eDNA extraction (Extended Data Fig. 3). Similar to the Irish samples, human eDNA was readily detectible from water samples taken near the city of St. Augustine, Florida, while no human eDNA was detectible by qPCR from ocean water collected on an incoming tide from the ocean beyond the Matanzas Inlet, Florida (Fig. 2a and Extended Data Fig. 4).

We next compared human eDNA recovery from beach sand footprints with that from an island with restricted human access. No human eDNA was detected by qPCR in sand samples from a remote area of Rattlesnake Island, Florida, part of the Fort Matanzas National Monument (Fig. 2b and Extended Data Fig. 5). This part of the island is inaccessible to the general public; our access was facilitated by the US National Park Service. By contrast, human eDNA was readily detectible from beach sand samples recovered from human footprints (Fig. 2b).

Then, we compared the recovery of human eDNA from air in rooms where humans were present with that from air in rooms where humans were absent. No human eDNA was detected by qPCR in negative field controls (eDNA filters left in rooms with humans present

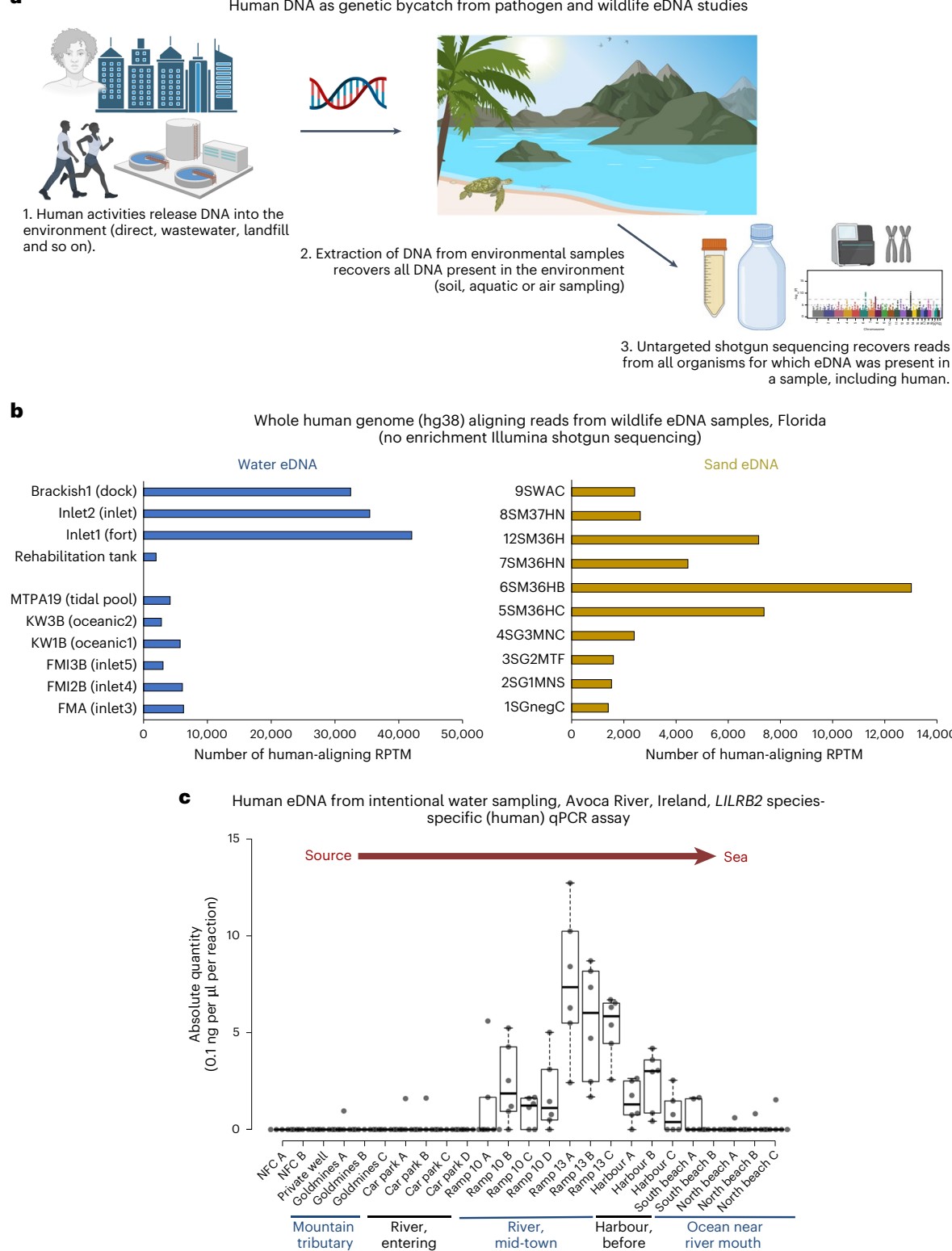

**Fig. 1 | Recovery of human eDNA from field samples. a**, Schematic overview of how human DNA can enter the environment and be inadvertently sequenced as HGB from pathogen- and wildlife-focused eDNA studies. Schematic created with BioRender. **b**, Whole human genome aligning reads from a wildlife eDNA shotgun Illumina sequencing study. **c**, qPCR-based species-specific quantification of human eDNA from Avoca River water sampling. The absolute quantity (0.1 ng per µl per reaction) of human eDNA per sample is shown. Each qPCR reaction is a 10 µl reaction containing 1 µl of extracted eDNA template. The samples were quantified with *LILRB2* human-specific assays. For filtered water volumes and elution volumes, see Supplementary Table 2. For matching samples quantified with *ZNF285* human-specific assays, see Extended Data Fig. 1c. Tukey whiskers (extend to data points that are less than 1.5 × interquartile range (IQR) away from 1st/3rd quartile) were utilized for each boxplot. The median for each sample is shown as a horizontal line within each box, and box edges are the upper and lower quartiles. One box is graphed per single sample, consisting of all qPCR technical replicate wells for that sample. Biological replicates are not pooled on any boxplots, with each sample being denoted by its own box.

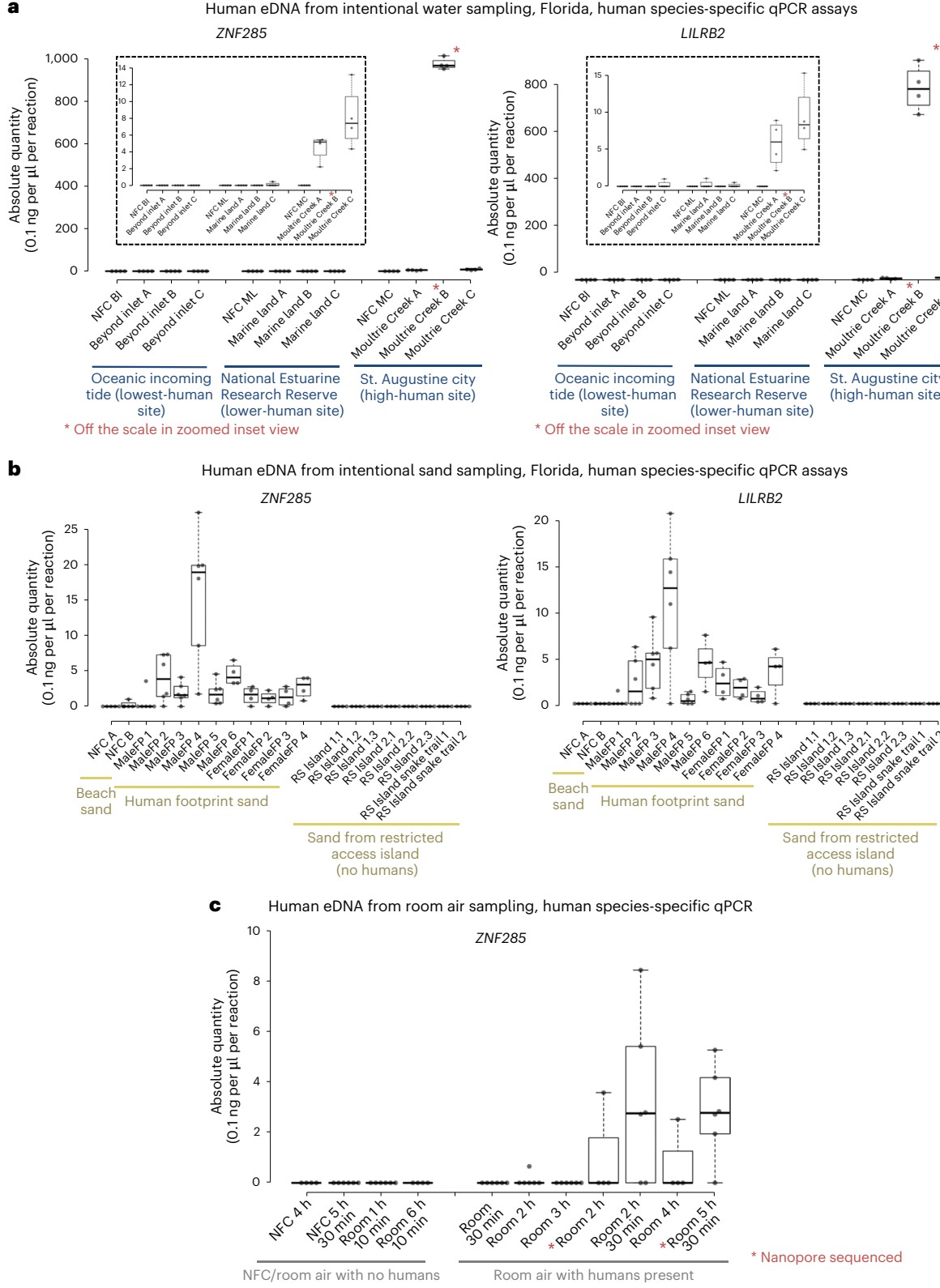

**Fig. 2 | qPCR-based species-specific quantification of human eDNA from Florida water, sand and air sampling. a**, Water eDNA sampling, qPCR, quantified with *ZNF285* (left) and *LILRB2* (right) human-specific assays. Insets: enlarged section plot. BI, beyond inlet; ML, marine land; MC, Moultrie Creek. **b**, Sand eDNA sampling, qPCR, quantified with *ZNF285* (left) and *LILRB2* (right) human-specific assays. **c**, Room air eDNA sampling, qPCR, quantified with *ZNF285* human-specific assays. The absolute quantity (0.1 ng per µl per reaction) of human eDNA per sample is shown. Each qPCR reaction is a 10 µl reaction containing 1 µl of extracted eDNA template. For filtered water volumes and elution volumes, see Supplementary Table 2. Tukey whiskers (extend to data points that are less than 1.5 × IQR away from 1st/3rd quartile) were utilized for every boxplot. The median for each sample is shown as a horizontal line within each box, and box edges are the upper and lower quartiles. One box is graphed per single sample, consisting of all qPCR technical replicate wells for that sample. Biological replicates are not pooled on any boxplots, with each sample being denoted by its own box.

during sampling, but not connected to a vacuum pump) or from vacuum-pumped air from rooms in which no humans were present (Fig. 2c). Human eDNA was recovered from rooms with humans present as participants went about regular working activities (Fig. 2c). Human eDNA was recovered even though air samples from rooms with humans present were collected from a sterile veterinary hospital environment.

Differences in the level of human eDNA detected were broadly in close agreement between the two independent species-specific eDNA assays (*LILRB2* and *ZNF285*). This indicates that both are suitable for qPCR-based human eDNA applications (Figs. 1c and 2a,b and Extended Data Fig. 1c).

Having quantified the level of human eDNA in each sample by qPCR, we selected seven samples for Oxford Nanopore MinION sequencing (Supplementary Table 3) to confirm that human genomic information could be derived from intentional human eDNA sampling. MinION shotgun sequencing was conducted with six intentional human eDNA samples (collected in 2022) and our longest-travelled water negative field control sample (collected in 2018, travelled over 904 km). Oxford Nanopore sequencing was selected as it generates longer read lengths (Extended Data Fig. 6a,b) than Illumina sequencing. Without any enrichment (shotgun sequencing), the samples of human eDNA in water, human footprint sand and room air returned thousands of human-aligning reads, while negative field control water and no-human-site sand eDNA samples (from a snake trail) had only 2 to 26 human-aligning reads (Fig. 3a and Supplementary Table 3). For all human-positive substrates, coverages were relatively even across the human reference genome, with reads from all chromosomes being detected (Fig. 3b). Nanopore sequencing revealed that eDNA is not necessarily fragmented DNA, as even without employing high-molecular-weight extraction methodology or long-read library preparation protocols (short-read nanopore buffer was used), we recovered human eDNA single reads up to 148,969 bp long (air eDNA; 120,998 bp for water and 39,229 bp for sand) (Fig. 3b and Supplementary Table 4). The longest human-mitochondrial-aligning single read was 16,535 bp, only 34 bp shorter than the full-length mitochondrial reference genome (Supplementary Table 4). The average read length across all human-positive nanopore samples was 1,514 bp (Supplementary Table 4).

To demonstrate that human eDNA could be used for applications beyond mere quantification, we examined known human genetic variants in the nanopore eDNA data to determine whether eDNA-based ancestry and disease susceptibility applications may be feasible. Deletions annotated in gnomAD (v.2.1)[26] could be detected in all three shotgun human-positive eDNA substrates, mainly in water but also to a lesser extent in sand and air samples (Fig. 4a and Supplementary Table 5). The longest deletion detected within a single read was 40,738 bp, a common copy-number polymorphism in European and Latino populations, located on human chromosome 2 (Fig. 4a). Genes associated with the deletions (deletions within or adjacent to the gene) from the Moultrie Creek B water sample have a range of functions including neuron differentiation, the regulation of double-strand break repair and the regulation of proteolysis. Deletions in or near prominent cancer-associated genes (for example, *ALK*, *LIN28B*, *PDGFD* and *WNT7B*) were also detected (Extended Data Fig. 7a and Supplementary Table 5). Other structural variant types (insertions and duplications) were also detectable, when analysed with the EPI2ME Sniffles-based structural variant caller (Extended Data Fig. 7b). Human mitochondrial reads from the shotgun nanopore sequencing were also assessed for confirmed mitochondrial pathogenicity-associated alleles (MitoTip[27] and ClinGen[28]); we detected seven mitochondrial mutations (six from water eDNA and one from air eDNA) associated with a range of diseases, including autism, diabetes, eye diseases and cardiac diseases (Supplementary Table 6).

We next assessed the feasibility of human exome enrichment on water and sand eDNA samples. Five eDNA samples were human exome enriched and then sequenced on an Illumina NovaSeq6000. Three of

these samples had also been subjected to long-read sequencing (Supplementary Table 4). Despite exome enrichment, the negative field control samples (NFC sand eDNA Rattlesnake Island site 2 sample 1 and NFC Moultrie Creek) generated very few human-aligning reads (Extended Data Fig. 7c and Supplementary Table 4). For all three human-positive eDNA samples sequenced, exome enrichment increased the proportion of human-aligning reads (Fig. 4b and Extended Data Fig. 7c). Post-exome enrichment, the water eDNA sample had 48% of sequenced reads aligning to the human genome (20.72 billion human-aligning bases), representing 473× coverage of the targeted human exome regions (Fig. 4b and Supplementary Table 4).

We then examined the human population-genomic-level resolution present within our eDNA samples. The detected deletions themselves (Fig. 4a) occur at varying allele frequencies in different human populations (Supplementary Table 5). We also conducted haplogroup and haplotype analysis on all sequenced intentional human-eDNA-positive samples: four nanopore samples, three exome-enriched samples and the wildlife-focused sand and water Illumina sequencing with the highest number of human-aligning reads. Every eDNA sample assessed enabled population genomic analysis, with proportions of specific haplogroups and haplotypes varying between samples (Fig. 4c). Generally, haplotypes from European–Indian ancestry, particularly H2a2a1, were the most widely recovered from each substrate type (Fig. 4c). For samples with known participants (footprint and room air), the haplotyping matched the participant profile, while for the water eDNA samples, the results broadly matched the demographics of the area from which the sample was taken (https://www.census.gov/quickfacts/fact/table/staugustinesouthcdpflorida,staugustinecityflorida/PST045221). Even with no prior enrichment, reads containing information on genetic ancestry and parental origin have been recovered (Fig. 4c and Supplementary Table 5).

Each sample also contained a complex mix of microbial reads, demonstrating the utility of eDNA (including air eDNA) to simultaneously detect humans, animals and microbes (Extended Data Fig. 8a,b). This included the room-air sample (from a sea turtle hospital), which had airborne pathogenic (tumour-implicated) sea turtle herpesvirus and papillomavirus DNA as well as sea turtle DNA (Fig. 4d and Extended Data Figs. 8c and 9a).

## Discussion

The rapid expansion of the field of eDNA coupled with increasingly cost-effective deep-sequencing technologies promises an exponential increase in the generation of shotgun data from environmental samples over the coming years[1,16,19,29,30]. This heightens the likelihood of unintended large-scale HGB from the general public in eDNA studies. In the present study, we report that HGB was found in all field eDNA samples. These samples had been collected primarily for the detection of non-human species, marine turtles, animal pathogens and metagenomics[8,17]. With no human enrichment prior to shotgun sequencing and with sampling having been conducted in areas of relatively low human habitation densities, we nevertheless inadvertently captured a substantial amount of human genomic data. There is likely to be a continuum across which HGB may be beneficial, neutral or exploitative—for example, samples collected from a popular tourist beach are likely to be less ethically concerning than wastewater monitoring of small, defined, stable populations. Similarly, areas with greater human population numbers are more likely to return more human data. When the sites were intentionally tested, far more human eDNA was indeed recovered from aquatic sampling sites in which water had passed through populated areas. Furthermore, targeted sequencing of human eDNA from complex metagenomic samples was readily achievable with existing exome enrichment approaches. Having demonstrated the propensity of eDNA approaches to capture human eDNA equally well as that of other species, we will now consider the potential beneficial applications and ethical implications of these findings (Box 1).

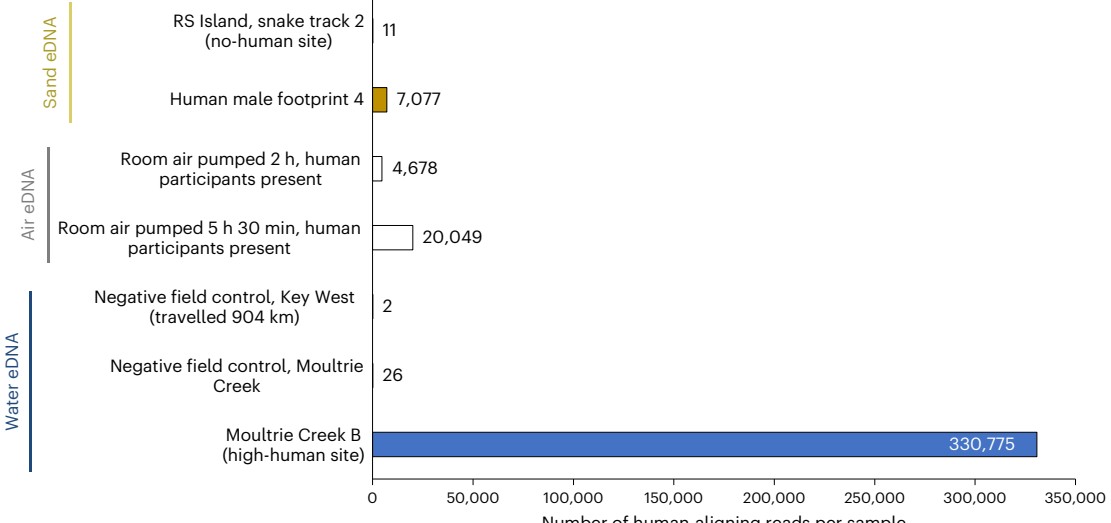

**a** Whole human genome (hg38) aligning reads from intentional high-human and low-human eDNA samples
(no-enrichment Oxford Nanopore shotgun long-read sequencing)

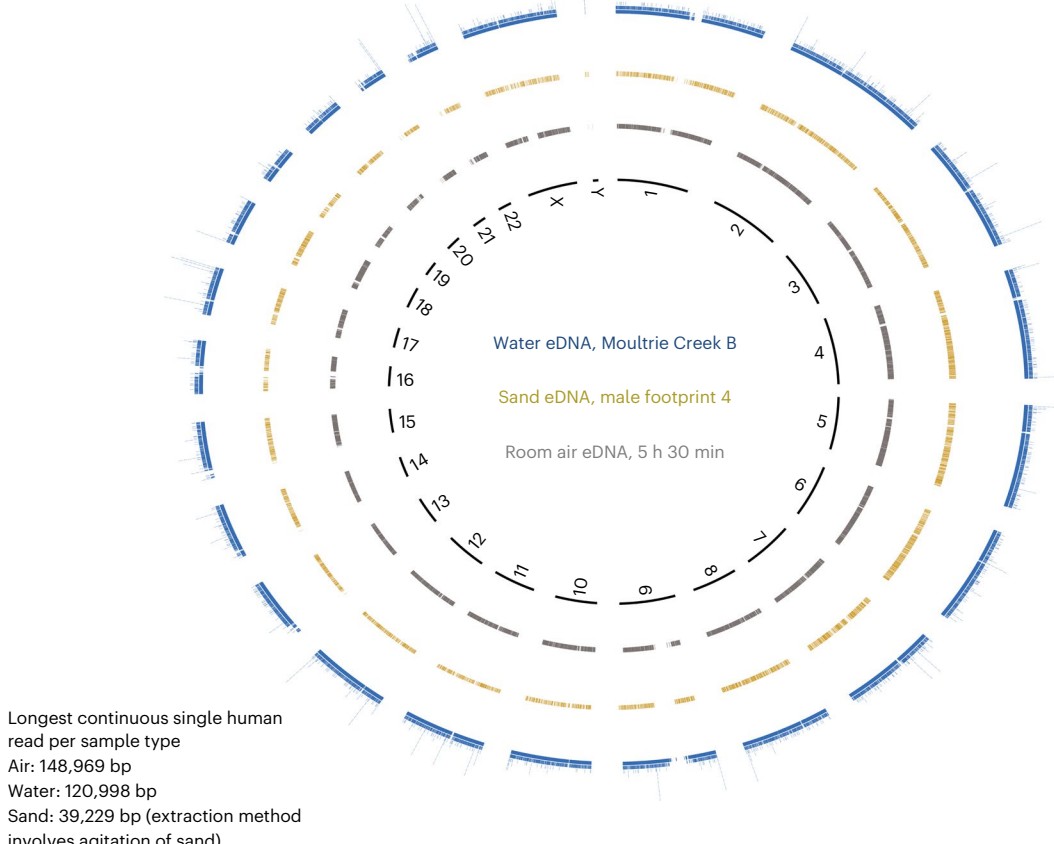

**b** Genome mapping of whole human genome (T2T) aligning reads from intentional human eDNA samples
(no-enrichment Oxford Nanopore shotgun long-read sequencing)

Water eDNA, Moultrie Creek B

Sand eDNA, male footprint 4

Room air eDNA, 5 h 30 min

Longest continuous single human
read per sample type
Air: 148,969 bp
Water: 120,998 bp
Sand: 39,229 bp (extraction method
involves agitation of sand)

**Fig. 3 | Oxford Nanopore long-read shotgun sequencing of eDNA samples.**
**a**, Total number of human-aligning reads (Bowtie2) from each sequenced field,
room and negative field control eDNA sample. **b**, Human genome alignment map
of the human-aligning reads (minimap2) from the high-human-site eDNA water
sample (Moultrie Creek B), the human male footprint 4 eDNA sample and the
room air 5 h 30 min eDNA sample.

At a minimum, HGB may complicate approvals for eDNA-based
deep-sequencing projects, potentially requiring them to be reviewed
by human-focused ethical review boards, even when humans are not
the intended target. As eDNA approaches are non-invasive, limited
approvals are currently needed, even when investigating endangered
species. Added layers of regulation may therefore complicate the
implementation and widespread adoption of eDNA approaches, ham-
pering conservation and non-human genome research applications.

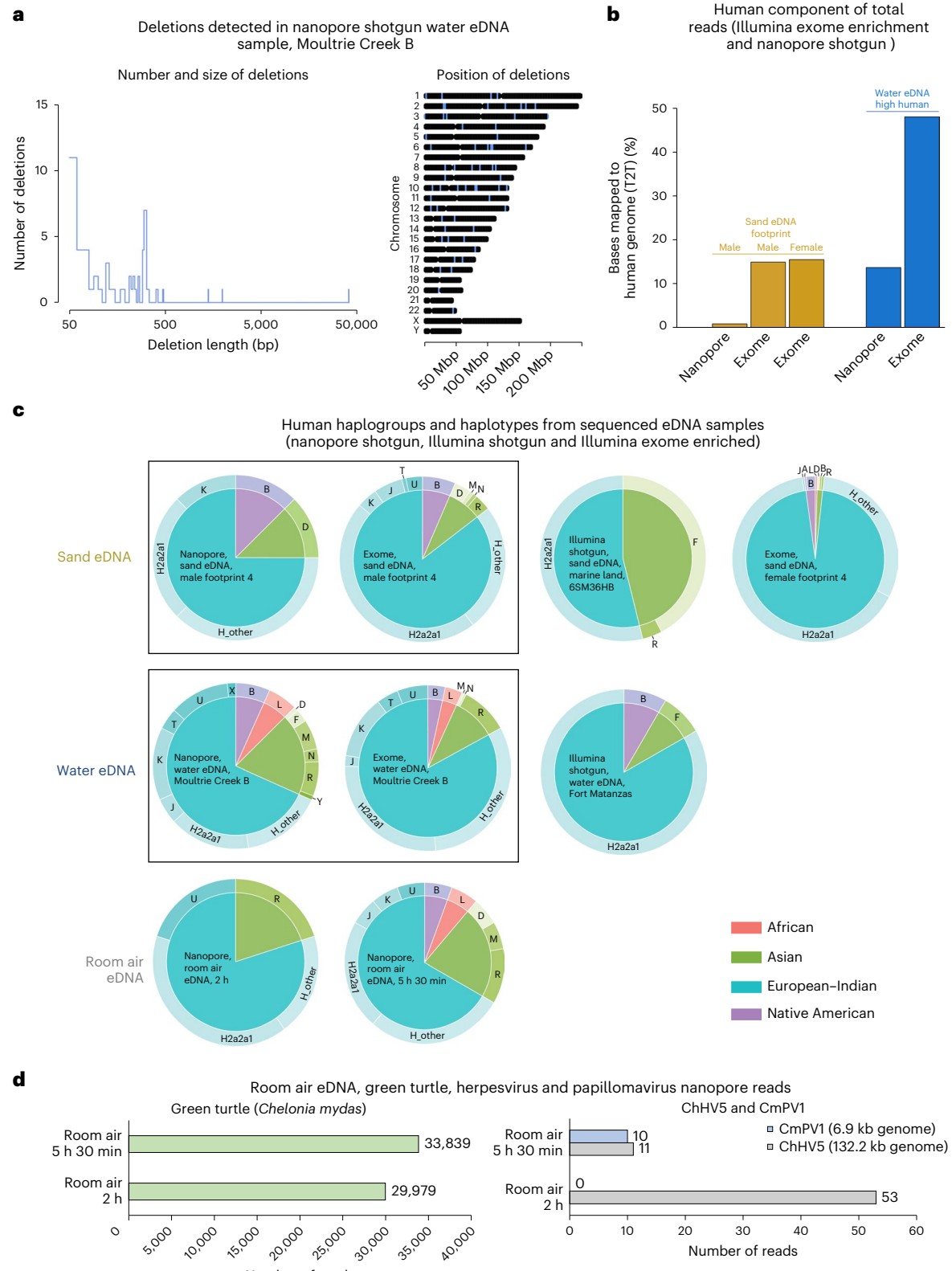

**Fig. 4 | Mutation and genetic ancestry analysis of shotgun and exome-enriched eDNA samples. a**, Known human genomic deletions (gnomAD database) detected in a nanopore shotgun water sample, including the number and size of deletions (left) and the genomic position of each deletion (right); deletion locations are denoted by blue shading. The full details of the detected deletions can be found in Supplementary Table 5. **b**, Percentage of sequenced reads from eDNA metagenomic samples that aligned (minimap2) to the human genome, compared with all reads generated for each sample (regardless of species of origin). Nanopore shotgun sequencing is compared with Illumina exome enrichment. Note that the same male footprint sand eDNA sample and the same high-human water eDNA sample was used for both shotgun sequencing (nanopore) and exome enrichment (Illumina). **c**, Haplogroup and haplotype analysis of human-mitochondrial-aligning reads for each sequenced (shotgun or exome-enriched) intentional human sample and for the water and sand bycatch Illumina shotgun-sequenced samples (sea turtle focused) with the highest number of human-aligning reads. Pie charts within the same box are from sequencing data for which the same original sample was used (that is, for shotgun and exome enrichment). **d**, Quantification of green sea turtle and sea turtle herpesvirus (ChHV5) and papillomavirus (CmPV1) aligning reads (minimap2) from the same room-air eDNA nanopore shotgun sequencing data as were used for identifying human-aligning reads (Fig. 3a).

## BOX 1

# Potential problematic implications of the capture of human genomic eDNA data

Unintended consequences:
- Requirement of human-study-related ethical approvals for wildlife studies
- Lack of human subject consent/breach of privacy
- Public deposition of eDNA data including human genomic data
- Inadvertent location tracking
- Inadvertent genome harvesting
Potential malicious applications:
- Genome harvesting—the ability to illegally/unethically harvest human genomic data from local populations/ethnic groups without their knowledge or consent
- Covert accumulation of human genetic data for malicious or commercial purposes (for example, genomic surveillance or big-data-fuelled discovery)
- Genetic surveillance—individual tracking (similar to forensics/wildlife applications)
- Genetic surveillance—unethical tracking/locating of ethnic groups/populations
- Genetic surveillance—potential for involuntarily genetic surveillance from investigative applications, including the recovery of bystander shed genetic information or intentional overreach
- Bio-piracy of human genetic data from populations and countries (akin to flora/fauna genetic bio-piracy)

The capture of non-genetic human information such as speech (eco-acoustic data) or human images (camera traps) has posed similar conundrums for conservationists as HGB. In those cases, codes of conduct, enhanced education and automated data filtering have been suggested to alleviate privacy and consent concerns[31–34].

The deposition of sequencing data in a public repository is a prerequisite for scientific publication and open data initiatives. However, if eDNA sequencing datasets also include human genomic information, this may produce a potential conflict. An open ethical question is therefore whether any such data would have to be pre-filtered to remove human sequencing data prior to deposition, similar to the de-identification of patient data[35]. This in turn raises issues relating to transparency and the correct filtering criteria. Furthermore, with whom should the responsibility of policing such a solution lie: journals, data repositories, ethical committees or investigators themselves? Another alternative is attempting to actively block the sequencing of human DNA from eDNA samples. However, a recent study reported that even when human blockers were applied, human reads were still detected[36].

Current deep-sequencing approaches suffer from a somewhat similar issue as HGB, whereby genomic information from personnel who process samples may be inadvertently sequenced[37,38]. However, in terms of ethical considerations, this inadvertent genome capture from contamination occurs from experienced personnel who are well versed in the technology. Environmental-DNA-based inadvertent capturing of genomic information from the general public is a more complex ethical conundrum, as the genomic information captured comes from individuals mostly unaware of the technology use and oblivious to the fact that their genetic information has been inadvertently obtained. Complicating matters, researchers are already

exploring the potential of eDNA for individual-level identification and tracking in wildlife populations[39–41]. HGB in eDNA metagenomics studies therefore raises the intriguing question of whether individual humans could also be identifiable from eDNA data; this is likely, given the unfragmented nature of eDNA reported here. This could require eDNA studies to obtain prior informed consent, a near-impossible task. Given the large geographical range over which eDNA can travel[1,6,42,43] (particularly for aquatic samples[44–46]), it is impractical to imagine that prior informed consent could realistically be obtained. Even with no targeting or enrichment, our MinION shotgun sequencing was able to identify the genetic ancestry within pooled human populations and to identify variants associated with disease susceptibility. The long eDNA reads sequenced here suggest that with targeted enrichment of informative genomic locations, one could achieve individual-level identification even from pooled samples. Given the advanced targeting approaches already in existence for genomic regions of interest (for example, from disease and population genomics research, nanopore adaptive sequencing[47–51] or the exome enrichment used in this study), targeting specific regions or variants from human eDNA samples is already completely feasible.

While our data demonstrate the possibility of human genomic bycatch in wildlife-oriented eDNA studies, they also highlight the feasibility of targeted human eDNA applications. Particularly sensitive or informative regions of the human genome could be intentionally targeted. Human privacy issues are particularly relevant to the increased use of pathogen detection from human wastewater during the COVID-19 pandemic[1,11,12]. Even in relation to pathogens, the legal and ethical implications of wastewater monitoring have not been adequately considered. Wastewater is currently utilized to detect illicit drugs, antidepressants, stress markers and alcohol consumption[52]. Our findings show that such ethical dilemmas need to be considered not only for wastewater sampling but for the field of eDNA as a whole, with HGB feasible from air, aquatic and substrate environmental sampling.

Given the demonstrated feasibility of human eDNA analysis, including disease-associated loci calling, it is likely that beneficial applications may exist for such human-orientated eDNA approaches (Box 2). For example, one could utilize wastewater or air eDNA-based sampling to correlate the level of pathogens with the abundance of susceptibility loci in a given population[1,13,15,53,54]. This may be especially feasible given the ever-increasing portability of DNA sequencing/surveillance technology[1,29]. It is also immediately feasible to consider intentional human and pathogen eDNA sampling from specific sources, such as substrate footprints or filtered-air eDNA (as we report here) from specific locations–for example, rooms in the home, public indoor or outdoor spaces, hospital wards, or even special diagnostic rooms designed to filter the host and pathogen eDNA shed by a single patient over a short period. Here we report that room-air sampling in a clinical setting (a veterinary hospital) recovered host (animal patient), human (staff) and vertebrate viral pathogens, suggesting future novel monitoring approaches in human and animal medical settings and beyond. Alternatively, water eDNA collected before wastewater leaves the home could be utilized, linking to continual health monitoring and chronic disease management initiatives[55]. This is particularly pertinent to routine continual health monitoring for somatic mutations, which arise later in life and can be drivers of life-threatening diseases such as cancer. Human eDNA approaches can be combined with advances in the ongoing genomic medicine revolution[24] to enable novel applications, as evidenced here by the ability to detect specific human disease-associated mutations from eDNA and to enrich for human genomic regions of interest (for example, exome) and simultaneously detect microbes, including pathogenic viruses.

Haplotyping and phylogenetic analysis of individuals (whale sharks, sea turtles and kakapo) from eDNA samples have already been achieved, although to date these have tended to be samples dominated by a single individual[17,47,51,56]. Our study adds humans to

## BOX 2

# Potential beneficial applications of human eDNA as the nascent field matures

- The discovery of novel human genetic variation can help redress the historical imbalance in human genomic databases not spanning the range of human diversity.
- Population-based disease risk susceptibility studies can be carried out, particularly wastewater- or air-based when coupled with active pathogen surveillance.
- Non-invasive monitoring of host genetics, pathogen load and transmission studies can be conducted, including from pooled settings (including being able to disentangle the number of pooled individuals contributing to non-controlled-environment samples).
- Human eDNA can be a new tool in continual health monitoring and continual personalized medicine biomarker monitoring for chronic disease management initiatives (particularly pertinent to somatic mutations, which arise spontaneously and can be drivers of life-threatening diseases such as cancer).
- It can enable sensitive quantification and source identification of human effluent entering and polluting waterways and aquifers (from septic tank leaching and release of improperly treated wastewater).
- Forensic and criminal investigative applications can aid in solving crime. Air eDNA holds particular novel promise.
- Human eDNA can assist in the recovery of missing persons/deceased remains (particularly drone-enabled air eDNA in remote locations, especially if coupled with real-time eDNA detection technology and remote reporting).
- It can help locate sites of archaeological importance (cryptic human remains, such as sacrificial sites in remote bogs).
- It can serve as a roadmap for future wildlife (fauna and flora) eDNA studies. Trialling approaches and tools with human eDNA and existing rich human genomic databases will be highly informative for what can be achieved for wildlife eDNA population genomics (haplotyping to individual, disease risk and so forth) once sufficient wildlife genomic resources/databases are established.

this list and highlights how haplotyping can be achieved from complex multi-individual (metagenomic) samples with diverse population structures. Given its less biased nature, human eDNA may also help redress the balance in relation to the lack of sequenced genomes from diverse populations and underrepresented rare human alleles in genomic databases. Traditionally, these databases have been skewed towards populations of European ancestry[57].

Separately, genetic analysis is already being applied to combat the illegal wildlife trade by identifying the source animal populations[58,59]. Similarly, it may be possible to adopt such approaches in targeting human eDNA recovered from illegal wildlife parts (poacher/trader handling) to broadly identify the geographic human populations through which the illegal parts transited. Human eDNA could also be employed to identify sites of archaeological importance; such possibilities include sampling of remote stagnant waterbodies (for example, bogs) to identify undiscovered sacrificial sites, or eDNA sampling to help in the recovery of more recent human remains. Even search-and-rescue missions could use eDNA techniques if currently

in-trial air eDNA sampling by drones is successful. Recently, it has been postulated that human eDNA extracted from the air could also have novel forensic and criminal investigation applications[60–62]. Our human population genomics assessment of airborne DNA is an early step in this direction, providing proof-of-concept of the recovery of shotgun-sequencing-quality human DNA from room air. Both ancient and contemporary DNA have their own established ethical procedures, although in those disciplines too, the pace of advancement has outpaced the dialogue about research ethics[63–66]. Vertebrate air eDNA is a recent field of study, and improved capture devices is an active area of research from which human airborne DNA applications could benefit[36]. Water-based eDNA approaches have also been postulated to be beneficial for forensic investigations[67,68]. Forensic science has well-established ethics procedures, although issues pertaining to racial discrimination persit[69,70]. Taken together, forensic applications can be envisaged for air, soil and water eDNA approaches, especially in light of the intact non-fragmented nature of eDNA reported here.

While benefits of human eDNA analysis may exist, there could conceivably be more worrying applications of such technology. The accumulation of population-level genomic databases is currently highly desirable, being a valuable research and commercial commodity[71,72]. While some projects maintain the genomic data and subsequent findings in public ownership, the lucrative economic potential of such databases means that a host of private companies have also been rapidly accumulating them (for example, DNA ancestry companies and genomic medicine companies) and selling access[72,73]. Countries and corporate entities are racing to create ever-larger pan-genomic patient/population datasets. Examples include the 100,000 Genomes Project and the new Genome UK Strategy (https://www.gov.uk/government/publications/genome-uk-the-future-of-healthcare); the Biobank project, which aims to sequence half a million genomes (https://www.sanger.ac.uk/collaboration/uk-biobank-whole-genome-sequencing-project/); and the National Institutes of Health All of Us Research Program (https://allofus.nih.gov/), a project to sequence the genomes of one million US citizens. Such pan-genomic activities have the potential for great public good, including advanced medical and pharmaceutical applications, but they have been an ethical minefield regarding ownership, data protection, insurance coverage and privacy issues[24,72,74–76]. Examples from digital databases also clearly show that wherever valuable databases exist, there is an ever-present temptation to exploit the data for financial gain, whether legally or otherwise. The Cambridge Analytica scandal, which involved complex harvesting of data from Facebook profiles[77], is one such example, and Google's harvesting of medical records with individual identifying information still intact is another[78].

Human eDNA applications present similar concerns: they could be employed for the surveillance of individuals, minority groups (genetic ancestry) or genetically driven disabilities, or to obtain genomic information from local populations without their knowledge or consent, including from 'valuable' genetically diverse indigenous populations (uncontacted tribes, particular ethnic groups and so on). Such scenarios expand the current issues regarding the commodification of genomic information, which is already a particularly acute concern in relation to indigenous peoples[72,79–81]. These novel human eDNA applications may also enable unscrupulous (though possibly not technically illegal) entities to perform covert mass capture of genomic data from populations, to further expand population-level genomic databases. Furthermore, such resources can then be utilized for enhanced monitoring of individuals for less savoury purposes. Genetic/genomic surveillance is a serious ethical concern, with documented human rights abuses having already occurred whereby national DNA databases were used with other surveillance data to monitor minority populations[70,82]. The application of human eDNA approaches could further undermine genetic consent, limiting the ability of threatened minorities to withhold their genetic information. In the future, human eDNA could also be

utilized to determine whether members of a genetically distinct group were present in a given population—for example, through wastewater monitoring or air filtering at checkpoints, in urban areas or in private dwellings. Such potential is particularly chilling given the propensity of humans to carry out ethnic persecution and genocide throughout our history.

Many of the ethical issues raised here are more likely to arise when sampling sites are closer to human urban areas. Similarly, issues pertaining to individuals or ethnic groups are more likely to be pertinent in areas with small, stable populations of humans contributing to the recovered eDNA—for example, a specific watershed or portion of a sewage system. It is clear that in the short term the convergence of massively powerful deep-sequencing technologies with improvements in ecological eDNA sample capture approaches raises serious questions about human genomic bycatch, requiring discussion and regulatory consideration from scientific, ethical, regulatory and social perspectives. Regulators, researchers, funders and other stakeholders should develop responses to the ethical implications of HGB and intentional human eDNA applications. Such planning should be initiated immediately, pre-empting the technology becoming even more widespread, affordable and entrenched, be it for beneficial or exploitative applications. Such ex ante planning is crucial for ensuring that laws and ethics stay ahead of emerging technology. Conversely, these same eDNA approaches can open up novel beneficial applications in areas from human health to criminal forensics.

## Methods

### Sample collection, DNA extraction and sequencing

All laboratory procedures from sampling through final analysis (sequencing or qPCR) were conducted in a way that minimized any human DNA contamination from investigators. Although the Illumina HiSeq eDNA samples had also been sampled and utilized for sea turtle and pathogen research, the original study design for these samples included assessment of the human-aligning reads from these shotgun data. Therefore, all appropriate precautions to avoid contamination were taken. This included no contact of investigators or samplers with the substrates being sampled (water or sand), new nitrile gloves being used for each sample collection, frequent glove changes throughout sample processing, cleaning of equipment and benchtops with bleach prior to use, and negative field controls being treated identically to genuine field samples throughout all processes from extraction through qPCR/sequencing (see below). All eDNA samples were extracted in labs that do not process human samples: a sea turtle lab in Florida and a chick/mouse developmental biology lab in Ireland. The Irish qPCRs were set up in a laminar flow cabinet after a protracted 1 h exposure of UV of the cabinet and equipment. It was originally postulated that the sea turtle rehabilitation tank water sample (2017, HiSeq) would contain more human DNA than the field water samples, due to hospitals staffs' interactions with the tank water and sea turtle patients. For all subsequent eDNA sampling, we had already established that the 2017 tank sample contained less human eDNA than the 2017 field samples. We therefore continued to observe strict procedures to avoid investigator contamination in all subsequent sampling in order to be able to investigate HGB. While we intended in advance to assess the level of human eDNA recovered from the 2017 and 2020 samples, no human factors (such as population density) were considered when selecting sampling sites. Site selection was purely based on the wildlife eDNA considerations of these sites—that is, site selection was not biased towards areas with high human densities. Conversely, site selection for all 2022 samples was intentionally directed towards sites with high or low human perturbations (primarily based on human population density).

Previously extracted[8] green and loggerhead sea turtle DNA from tissue samples was used for species specificity qPCR tests. DNA was extracted from tissue using a Qiagen DNeasy Blood and Tissue kit

(Qiagen, cat. no. 69504) according to the manufacturer's instructions. SH-SY5Y cells (ATCC, cat. no. CRL-2266) were gifted by the Loesgen lab (University of Florida), and no human cells were processed in the same lab as the eDNA samples. DNA was extracted using a Qiagen DNeasy Blood and Tissue kit according to the manufacturer's instructions. SH-SY5Y genomic DNA was used for generating standard curves. To avoid any potential contamination, the standard curves were run only after all eDNA qPCRs had been completed.

**HGB samples.** Seawater (rehabilitation tank and oceanic) and beach sand eDNA samples were collected between 2017 and 2021 and sequenced as part of our sea turtle and sea turtle pathogen research[8,17]. All samples originated from Florida, US[8,17]. The tank sample is a pooled[8,17] sample containing eDNA extracted from seawater from five rehabilitation tanks at the University of Florida's Whitney Laboratory for Marine Bioscience and Sea Turtle Hospital. The four main tanks housed juvenile green sea turtles and were 240 cm in diameter and had a full volume of 2,270 l, and a smaller loggerhead post-hatchling tank had a volume of 480.75 l. The separately extracted eDNA from each tank was pooled in equal volumes prior to library preparation.

**Intentional human samples.** River, estuarine, seawater (oceanic) and beach sand samples were collected between May and July 2022 (Supplementary Table 2). Negative field control samples of water and sand were also collected and processed as per the study samples. For negative field control water sampling, 1 l of MilliQ water (Florida) or 1 l of Qiagen Nuclease-free water (Ireland, cat. no. 129117) was transported from the laboratory to the rehabilitation or wild sampling locations and stored in a cool box with the environmental samples to monitor for potential contamination during sampling, transportation and processing. For sea turtle negative field control sand sampling, 50 ml of beach sand was collected away from suspected turtle presence (that is, away from sea turtle tracks or obvious human activity) on each sampling trip. For human negative field control sand sampling, 50 ml of beach sand was collected away from suspected human activity (such as footprints) on each sampling trip and from a restricted-access location on Rattlesnake Island, part of the Fort Matanzas National Monument managed by the US National Park Service. The water and sand negative field controls were filtered and extracted alongside the other collected sand and water samples from each sampling trip and subjected to the same next-generation sequencing conditions and qPCR conditions (intentional human samples). The standard volume of seawater filtered (0.22 µm pore Millipore Sterivex-GP Pressure Filter Units (Merk Millipore, cat. no. SVGPL10RC)) was 500 ml for each DNA sample[8,17]. Samples of less than 500 ml (Supplementary Tables 1 and 3) were a lower volume due to debris clogging the filter and preventing larger volumes being filtered. One sample ('tidal pool') had 1 l of seawater filtered. For sand eDNA, a 50 ml tube was filled with sand from each sampling event, with 10 ml of this sand used per individual eDNA extraction[17]. Human-present air samples were collected from a 280 ft$^2$ room while the participants went about their daily work activities (that is, they could enter and exit the room throughout the sampling period), with a maximum of the same six participants using the room for a portion of the sampling period. The room was air-conditioned (outside air) and had an external door that was opened and closed, and occasionally left open for certain work procedures. Negative field control samples of air were also collected and processed as per the study samples. For air eDNA, two types of negative field controls were collected: (1) a filter kept in the room being sampled for the duration of sampling, but with no air being pulled through it (that is, no pump), and (2) air filtered (with a pump) from a room with no humans present at the time of filtering or during the previous 24 h. Human-related sampling was conducted with University of Florida Institutional Review Board (IRB-01) ethical approval under project number IRB202201336, with all participants providing informed consent. Four participants (three

female and one male) provided sand footprint samples, and six participants (five female and one male) provided room-air samples (pooled room air). Participation was on a voluntary basis, and the participants received no compensation.

Prior to filtration and between every sample, the laboratory surfaces and filtration equipment (all standard laboratory equipment involved in the washing and filtration process, as well as the filtration pump itself) were disinfected with 70% ethanol, and the sampling equipment (collection bottles) was disinfected (washed thoroughly) with 10% bleach and rinsed thoroughly with deionized water.

**Sand.** 1X TE−IDTE pH 8.0 1X TE Solution (Integrated DNA Technologies, cat. no. 11-05-01-09) was added to each individual sand sample in individual 50 ml Falcon conical centrifuge tubes, at approximately two times the volume of sand (10 ml of sand and 20 ml of 1X TE). The samples were shaken gently by hand and then set on a rocking platform for 1 h at room temperature, with additional gentle shaking by hand every 15 min. The samples were rested until sand had sunk to the bottom of each tube; then, the supernatant was immediately pipetted into a 60 ml sterile BD luer lock syringe (Fisher Scientific, cat. no. 136898). The samples were then hand-filtered using 60 ml BD luer lock syringes through 0.22 µm Sterivex-GP Pressure Filter Units (Millipore, cat. no. SVGPL10RC) and capped with B.Braun luer lock caps (Medline, cat. no. BMGTMR2000B). Finally, 740 µl of Buffer ATL and 60 µl of Proteinase K from a Qiagen DNeasy Blood and Tissue Kit (Qiagen, cat. no. 69504) were added to each sample, and they were placed in 50 ml Falcon conical centrifuge tubes in a rolling incubator overnight (24–26 h) at 56 °C.

**Water.** The water samples were pumped (by hand (in Ireland) or electronically (in Florida)) through 0.22 µm Sterivex-GP Pressure Filter Units (Millipore, cat. no. SVGPL10RC) and capped with B.Braun luer lock caps (Medline, cat. no. BMGTMR2000B). Hand-pumping was with sterile 60 ml BD luer lock syringes. Electronic pumping was done using a GeoTech Peristaltic Pump Series II. Then, 740 µl of Buffer ATL and 60 µl of Proteinase K from a Qiagen DNeasy Blood and Tissue Kit were added to each sample, and they were placed in 50 ml Falcon conical centrifuge tubes in a rolling incubator overnight (24–26 h) at 56 °C.

**Air.** For air eDNA sampling, room air was passed through 0.22 µm pore Millipore Sterivex-GP Pressure Filter Units (Merk Millipore, cat. no. SVGPL10RC), using a Welch vacuum pump (2019LD-4112) or a GeoTech Peristaltic Pump Series II. Environmental DNA was extracted[8,17] as for the water and sand samples, except that only 20 µl of Proteinase K was added per filter and the 56 °C ATL Buffer and Proteinase K incubation was conducted for 1 h.

**DNA extraction and sequencing.** For all three sample types (sand, water and air), after the 56 °C incubation, the solutions of Buffer ATL (after water or air filtration, or after sand washing and filtration), Proteinase K and eDNA were transferred from the Sterivex-GP Pressure Filter Units to 2 ml microcentrifuge tubes using 10 ml BD Slip Tip Sterile Syringes (Fisher Scientific, cat. no. 14823434). DNA was then isolated using a modified Qiagen DNeasy Blood and Tissue Kit protocol[8,17,83]. Following incubation, equal volumes of 800 µl (sand and water) of AL Buffer and 800 µl (sand and water) of ice-cold ethanol were added to each sample, and they were vortexed vigorously and microcentrifuged after each addition. For the air samples, 400–500 µl of each solution was used (as a lower volume of ATL solution is recovered from the initially dry Sterivex filters). Each sample was loaded into a DNeasy spin column and centrifuged at 6,000$g$ for 1 min, with flow-through being discarded after each spin (repeated until the entire contents of each sample were spun through the spin column). Then, 500 µl of Buffer AW1 was added, and the samples were centrifuged at 6,000$g$ for 1 min. Next, 500 µl of Buffer AW2 was added, the samples were centrifuged at 16,000$g$ for 3 min, flow-through was removed and they were spun for

an additional 1 min at 16,000$g$. DNA was eluted with 70 µl of AE Buffer (incubated at 70 °C before being added to the spin column), incubated on the column at room temperature for 7 min and centrifuged into a 1.5 ml microcentrifuge tube at 6,000$g$ for 1 min. DNA concentration was measured on a ThermoScientific Nanodrop 2000 Spectrophotometer (Fisher Scientific), and the samples were stored at −20 °C until qPCR or shotgun sequencing.

Library preparation and Illumina shotgun sequencing were conducted at the University of Florida's Interdisciplinary Center for Biotechnology Research Core Facilities. Four water eDNA samples (2017) were sequenced on an Illumina HiSeq 3000, and all subsequent Illumina samples (seven water and ten sand eDNA) were sequenced on an Illumina NovaSeq 6000 (Supplementary Table 1)[8,17]. All Oxford Nanopore samples were sequenced on a MinION in the Duffy lab at the University of Florida's Whitney Laboratory for Marine Bioscience. This device was not previously used for any human samples. We used a personal MinION sequencer instead of the previous core facility high-throughput Illumina sequencer as there were some human-aligning reads in our Illumina negative field control sample, although these were 13 to 38 times fewer human reads than were recovered from environmentally derived eDNA samples (Supplementary Table 1). MinION libraries were prepared according to the manufacturer's instructions, using the following kits: Oxford Nanopore Technologies (ONT) Ligation Sequencing Kit (SQK-LSK110, cat. no. 76487-106), NEBNext Companion Module for ONT Ligation Sequencing Kit (cat. no. E7180S) and ONT Flow Cell Wash Kit (EXP-WSH004, cat. no. 76487-116). They were then sequenced on ONT Minion Flow Cells (cat. no. 76487-106). For run times and the percentage of pores available, see Supplementary Table 3. All sea-turtle-related sequenced samples including raw reads are deposited in the National Center for Biotechnology Information (NCBI) database (https://www.ncbi.nlm.nih.gov/) under BioProject ID PRJNA449022. All human-eDNA-related sequenced samples are under BioProject ID PRJNA874696.

Exome capture libraries were constructed and sequenced at the UF ICBR Gene Expression and NextGen Sequencing Cores. Five water or sand eDNA samples (three intentional human-centred samples and two negative field controls) were used for Illumina DNA Prep with Enrichment, Exome Panel (cat. no. 20020183, captures 45 Mb of exonic content) exome capture analysis, according to the manufacturer's instructions. Briefly, eDNA was fragmentized, adapters ligated and amplified for nine cycles and purified with Ampure XP beads (Beckman Coulter, cat. no. A63881). The five samples were pooled with equal volume for one exome hybridization according to the user guide, and the probe hybridization was performed at 95 °C for 5 min, one cycle of 1 min each, starting at 94 °C for the first cycle, then decreasing 2 °C per cycle for 18 cycles, and a hold for 90 min at the final temperature. After the exome hybridization, the probe capture, wash and elute was performed, followed by library enrichment with ten cycles of PCR amplification. Exome sequencing was conducted on an Illumina NovaSeq 6000 S4 flow cell for 2× 150 bp cycles aiming for 50 million reads per sample (with negative field controls expected to return fewer reads due to a lack of human eDNA).

## Bioinformatic analysis

All bioinformatic tools were utilized using the default parameters, unless otherwise stated. The Galaxy platform (https://usegalaxy.eu/) was used for bioinformatic analysis[84,85], with NanoGalaxy also used for nanopore sequenced data[86]. All samples were checked for quality (FastQC, v.0.73; ref. [87]), adapters and low-quality reads were trimmed (<20 quality score) (Trim Galore! (https://www.bioinformatics.babraham.ac.uk/projects/trim_galore/) v.0.6.7 for HiSeq and NovaSeq data; Porechop v.0.2.4 (ref. [88]) for Oxford Nanopore sequence data), and high-quality reads were aligned (Bowtie2, v.2.4.2; ref. [89]) to the human reference genome (Hg38, https://www.ncbi.nlm.nih.gov/genome/?term=human%20genome%2038) (paired-read alignments, Illumina and single-read

alignments, nanopore) and aligned (minimap2, v.2.24; ref. 90) to the newly released complete human genome (T2T-CHM13v1.1; ref. 91). Trimmed air eDNA nanopore data were also aligned (minimap2, v.2.24; ref. 90) to the following reference genomes: green sea turtle (*Chelonia mydas*) (rCheMyd1, NCBI accession number: GCF_015237465.2 (refs. 92)), ChHV5 (NCBI accession number: HQ878327.2 (ref. 93)) and CmPV1 (NCBI accession numbers: MT179559.1, MT179558.1 and EU493091.1)[94,95]. To examine human genetic reads in the water and sand samples, StringTie (v.2.1.7; ref. 96) was used to identify reads aligning to the human Y chromosome only, by collating Y-chromosome-specific reads (gene abundance per sample abundance). Total reads per sample aligning to the human Y chromosome were also quantified using Samtools idxstats (v.2.0.4; ref. 97), using alignment files as input. The human Y chromosome was selected because it is fast-evolving and can therefore confirm the presence of genuine human reads[25]. Total reads per sample aligning to human nuclear and mitochondrial regions were also quantified using Samtools idxstats (v.2.0.4), using alignment files as input. Human mitochondrial haplogroups were classified in MitoMaster[98], using Haplogrep[99] and Phylotree 17 (ref. 100), with the pathogenicity of mitochondrial variants determined by MitoTip[27] and ClinGen[28]. Human mitochondrial haplogroup charts were produced in RStudio[101], using the webr package v.0.1.5 (https://github.com/cardiomoon/webr), and the human genome coverage plots were produced using the ggplot package v.3.4.0 (ref. 102).

The ONT EPI2ME platform was used for structural variant calling and metagenomics, and adapter and low-quality read trimming was conducted with Porechop. EPI2ME Structural Variant Caller was run using the default parameters (including a minimum of three reads to support each call), with the only exception being a minimum structural variant length of 20 bp. Alignment was to the human reference genome (GRCh38; ref. 103) using minimap2 (ref. 90), and structural variant calling was performed with Sniffles[104,105]. Metagenomic analysis was conducted with What's in My Pot (v.2021.11.26), which utilizes Centrifuge and Dustmaker[106–108].

For the structural variant analyses on ONT sequencing, we used minimap v.2.17-r941 (ref. 90) to align adapter pre-trimmed nanopore eDNA sequencing reads against the human genome GRCh37 with the default parameters. The resulting files were indexed, binarized and sorted using samtools v.1.10 (ref. 97), followed by mapping coverage estimation with samtools coverage. For high-sensitivity rearrangement calling, Sniffles v.2.0.7 (refs. 104,105) was used to detect structural variants in each sample, using the specific settings 'non-germline' and 'minsupport = 1' to increase sensitivity to single split-read resolution. The results of individual runs were used as inputs for a second, combined rearrangement calling by Sniffles' multi-sample input support. To screen for known human deletions, outputs were processed using the vcfR (v.1.12.0) library in R[109]. Deletion call regions were matched against the gnomAD v.2.1 structural variants database[26], using a lenient distance threshold of up to 5% of the total event length around both the 5′ and 3′ deletion break-end positions. We manually verified the resulting copy number variants' ONT sequencing support using IGV[110]. Deletion hits were visualized on GRCh37 chromosome maps using the chromoMap (v.4.1.1) library in R[111]. Functional clustering of deletion gene hits was performed using HumanBase (https://humanbase.net/) functional gene network analysis[112], accessed 21 October 2022.

## Quantitative PCR

Two human Applied Biosystem pre-validated Taqman Gene Expression qPCR assays directed against the *LILRB2* gene and the *ZNF285* gene (assay IDs Hs01629548_s1 and Hs00603276_s1, respectively) were selected for use as species-specific human assays, on the basis of having no cross reactivity with over 27 other species from mice to plants (https://www.thermofisher.com/order/genome-database/; mouse, rat, *Arabidopsis*, *C. elegans*, fruit fly, bovine, dog, Chinese hamster, goat, white-tufted-ear marmoset, guinea pig, zebrafish, horse, chicken,

soybean, cynomolgus monkey, sheep, rabbit, rice, rhesus monkey, baker's yeast, fission yeast, pig, bread wheat, wine grape, western clawed frog and maize) and having both primers and probe within a single exon (that is, detect DNA). We also showed that these assays did not cross-react with green sea turtle or loggerhead sea turtle DNA (Extended Data Fig. 9b). A pan-eukaryotic 18S rRNA gene (Applied Biosystem, 4352930E) pre-validated Taqman Gene Expression assay, which also has both primers and probe within a single exon (that is, detects DNA), was used to quantify the total level of pan-eukaryotic DNA in each of the Irish samples. Green sea turtle species-specific qPCR from air eDNA was conducted as previously developed and validated assays targeting the 16S rRNA gene, including the use of a 499 bp synthetic gene fragment for standard curve generation[8,17] (forward primer, TGCAAAAGCGGGAATAACAC; reverse primer, TCGCCCCAAC-CAAAAATATAG; FAM labelled (ZEN and Iowa Black double quenched) probe, CAACTATCTATACCCACTCACTCTAAGGACCTATAA (synthesized by Integrated DNA Technologies)). For the green sea turtle 16S rRNA synthetic gene fragment sequence, see Supplementary Table 7.

The qPCR reaction mixtures were performed on 384-well plates in a total volume of 10 µl per well: 5 µl of TaqMan Fast Advanced Master Mix (Fisher Scientific, cat. no. 4444557), 3.5 µl of nuclease-free water (Fisher Scientific), 0.5 µl of the respective assay (primer/probe mix, manufacturer-supplied concentration) and 1 µl DNA template (or, for no-template controls, an additional 1 µl of nuclease-free water per well). Depending on the sample type and the sample volume available, each biological sample was run in three to six technical replicates (three technical replicates for tissue samples and four to six technical replicates for eDNA samples). Negative field controls had the same number of technical replicates run as their corresponding eDNA samples. No template controls were run in triplicate on every qPCR plate. qPCR reactions were performed on an Applied Biosystems QuantStudio 6 Pro (Florida samples) or an Applied Biosystems QuantStudio 7 Flex (Irish samples) with the following cycling parameters: 95 °C for 20 s for one cycle, followed by 45 cycles of 95 °C for 1 s and 60 °C for 20 s. The qPCR results were plotted with BoxPlotR[113] (http://shiny.chemgrid.org/boxplotr/) with every datapoint displayed. Tukey whiskers (extending to data points that are less than 1.5× the interquartile range away from the first/third quartile) were utilized for every box plot. One box is graphed per single sample, consisting of all qPCR technical replicate wells for that sample. Biological replicates are denoted by the letters A–D at the end of the sample name. Biological replicates are not pooled on any box plots; each sample is denoted by its own box.

## Permitting statement

Sea-turtle-related sampling was carried out under Florida Fish and Wildlife Conservation Commission permits and with ethical approval from the University of Florida's Institutional Animal Care and Use Committee; see ref. 17 for the full details. Human-related sampling was conducted with University of Florida Institutional Review Board (IRB-01) ethical approval under project number IRB202201336, with all voluntary participants providing informed consent. Sampling at the Fort Matanzas National Monument (Rattlesnake Island) was conducted under a United States Department of the Interior National Park Service permit, permit number FOMA-2022-SCI-0003.

## Reporting summary

Further information on research design is available in the Nature Portfolio Reporting Summary linked to this article.

## Data availability

All Illumina sequenced samples including raw reads are deposited in the NCBI database (https://www.ncbi.nlm.nih.gov/) under BioProject ID PRJNA449022. All Oxford Nanopore sequenced samples including raw reads are deposited in the NCBI database (https://www.ncbi.nlm.nih.gov/) under BioProject ID PRJNA874696.

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

## Acknowledgements

Funding for the initial HGB eDNA study was generously provided by the National Save the Sea Turtle Foundation under project name Fibropapillomatosis Training and Research Initiative (D.J.D.), a Welsh Government Sêr Cymru II and the European Union Horizon 2020 research and innovation programme under the Marie Skłodowska-Curie grant agreement no. 663830-BU115 (D.J.D.). This research was also supported by Gumbo Limbo Nature Center d/b/a Friends of Gumbo Limbo (a 501c3 non-profit organization) through a generous donation through their Graduate Research Grant programme (J.A.F.) and by an Irish Research Council Government of Ireland Postgraduate Scholarship, under project no. GOIPG/2020/1056 (L.W.). Intentional human eDNA research was funded by D.J.D.'s University of Florida start-up funds. M.R.S. was supported by an EMBO long-term fellowship (ALTF 544-2021). We thank M. Q. Martindale, N. Condron, K. Yetsko, S. Creer, A. Pacetti, E. Ryan, C. Eastman and all of our generous co-authors on the wildlife and pathogen research papers for which these HGB samples were originally generated[8,17]. We thank P. Murphy and R. Rolfe for facilitating eDNA extraction of Irish samples in their lab in the Zoology Department, Trinity College Dublin, and A. Krstic, W. Kolch, A. G. Munoz and the Conway Core Facilities staff for facilitating qPCR of the Irish samples at Systems Biology Ireland and the Conway Institute of Biomolecular and Biomedical Research at University College Dublin. We also thank F. Duffy and I. Duffy for assistance with sampling; M. ten Cate and A. Whilde for gifting supplies; K. Foote, A. Rich and the NPS staff of the Fort Matanzas National Monument for valuable assistance with permitting and facilitating access to Rattlesnake Island and site selection; and S. Loesgen (University of Florida) for the SH-SY5Y cells. Finally, we thank the anonymous study participants who permitted us to collect their footprints and room air for human eDNA analysis, with full ethical approval and informed consent.

## Author contributions

D.J.D. designed and supervised the project. D.J.D., J.A.F., T.O., S.A.K., V.S. and J.W. conducted the sampling. J.A.F., D.J.D., S.A.K., N.M. and V.S. generated the data. L.W., M.McC., M.R.S. and D.J.D. conducted the bioinformatics analysis. All authors read and approved the final manuscript.

## Competing interests

The authors declare no competing interests.

## Additional information

**Extended data** is available for this paper at https://doi.org/10.1038/s41559-023-02056-2.

**Correspondence and requests for materials** should be addressed to David J. Duffy.

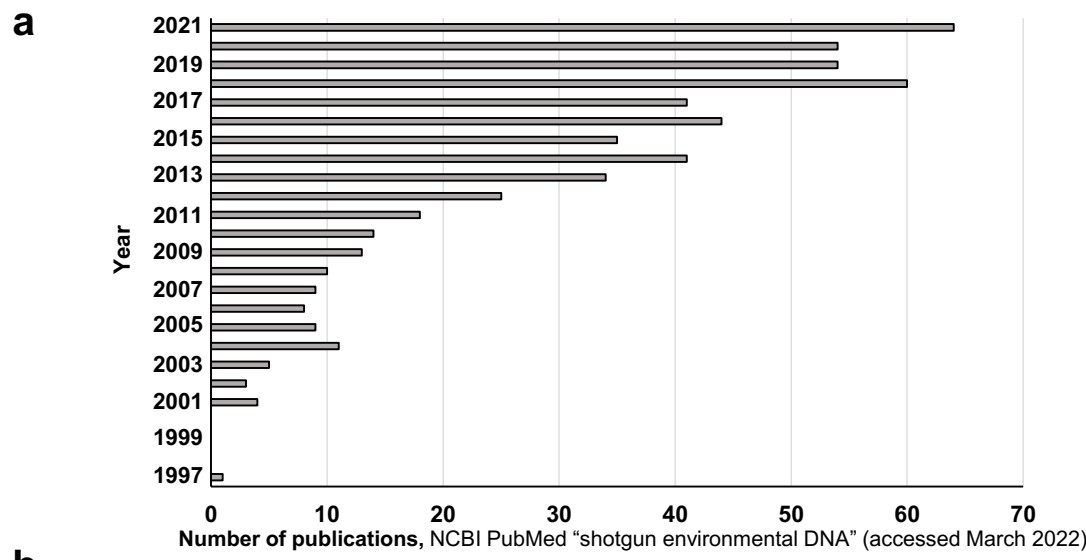

**a**

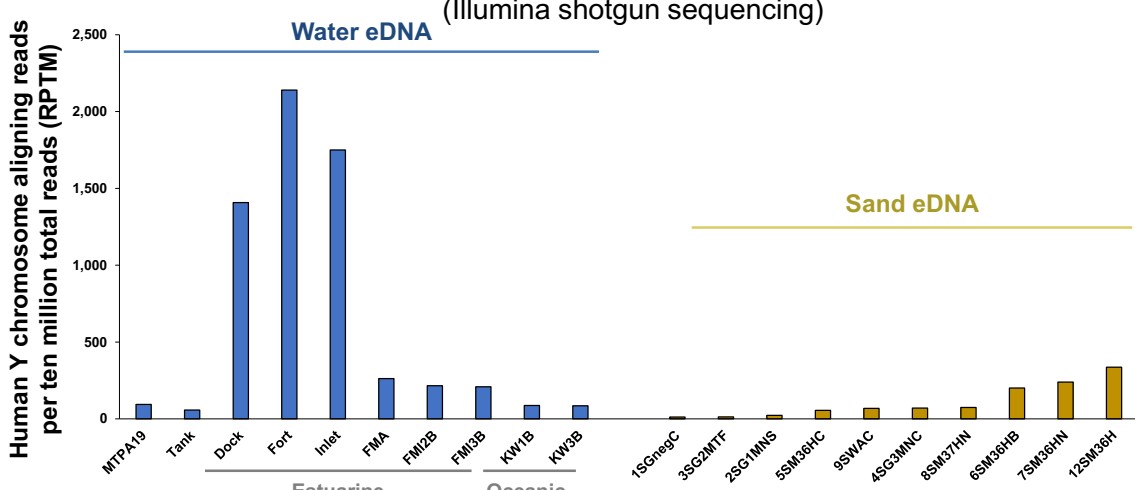

**b**

**Human Y chromosome aligning reads from wildlife eDNA samples**
(Illumina shotgun sequencing)

**c** **Human eDNA from intentional water sampling, Avoca River, Ireland, ZNF285 species-specific (human) qPCR assay**

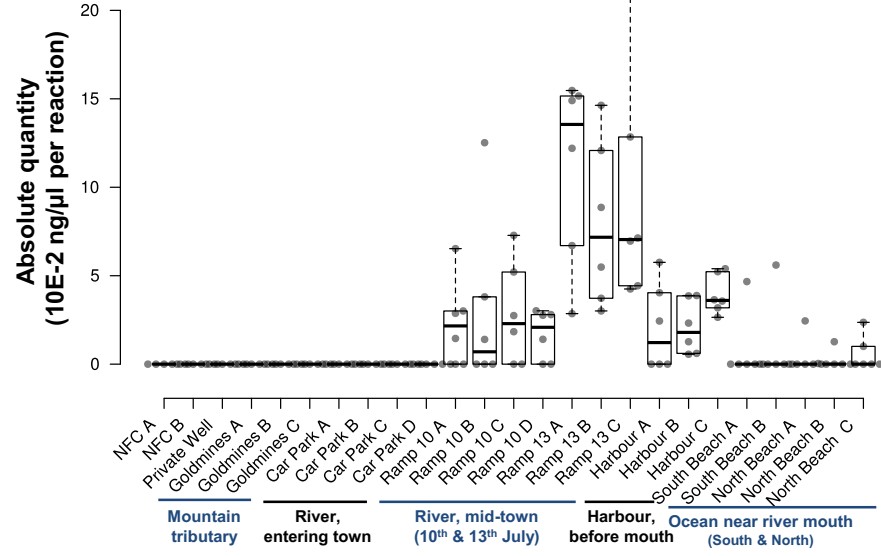

**Extended Data Fig. 1 | See next page for caption.**

**Extended Data Fig. 1 | Additional eDNA publication number, human eDNA bycatch and intentional human eDNA quantification data. 1a)** Graph of 'shotgun environmental DNA' papers in PubMed by year published. A literature search for 'shotgun environmental DNA' returned 499 results in NCBI's PubMed (https://pubmed.ncbi.nlm.nih.gov) and 654 results in Clarivate's Web of Science (www.webofscience.com). Search conducted on March 11th 2022. Note, papers from 2022 were excluded from the graph, as the year had only partially elapsed, but were included in the total tally. **1b)** Human Y chromosome aligning reads per ten million total reads (RPTM) from bycatch shotgun sequenced water and sand eDNA samples. **1c)** qPCR based species-specific quantification of human eDNA from Avoca River water sampling, intentional capture. Absolute quantity (10E-2 ng/µl per reaction) of human eDNA per sample. Each qPCR reaction is a 10 µl reaction containing 1 µl of extracted eDNA template. Quantified with *ZNF285* human specific assay. For filtered water volumes and elution volumes see Supplemental Table 2. For matching samples quantified with *LILRB2* human specific assay see Fig. 1c.

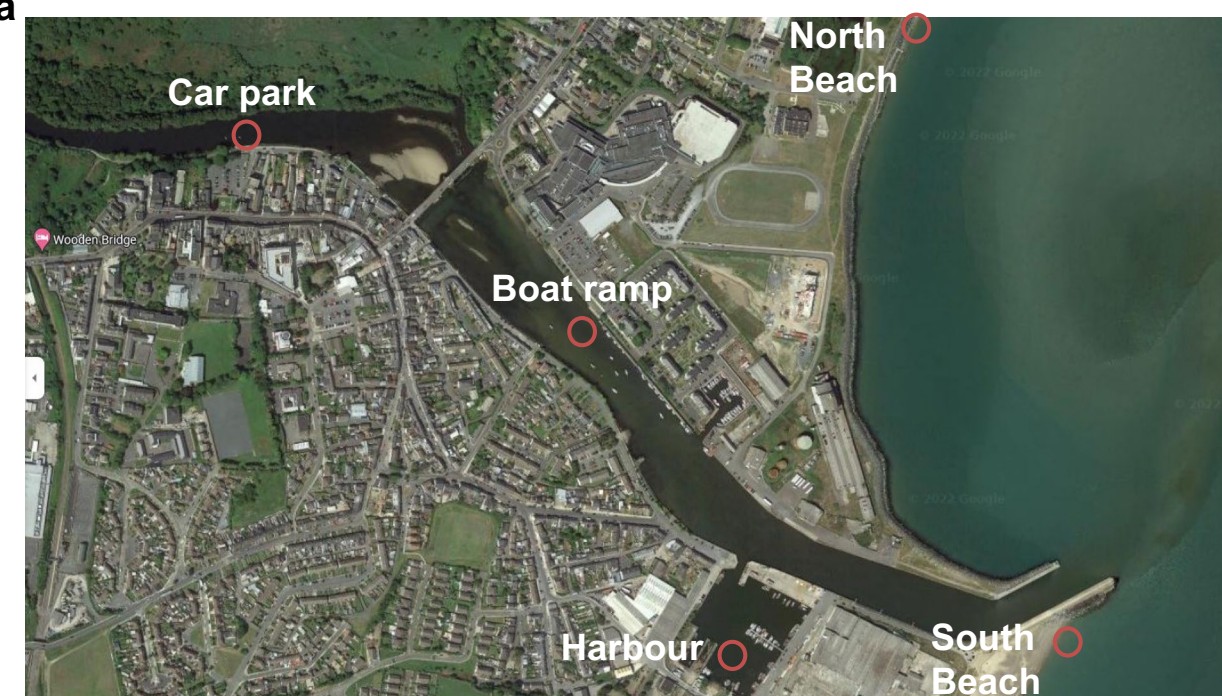

**Extended Data Fig. 2 | See next page for caption.**

**Extended Data Fig. 2 | Satellite imagery view (Google Earth), showing the locations of the Irish 2022 sampling sites.** Satellite imagery view (Google Earth, Map data ©2021 Google, SIO, NOAA, U.S Navy, NGA, GEBCO and TerraMetrics), showing the locations of the Irish 2022 sampling sites. **2a)** Zoomed view of the Avoca River passing through Arklow Town, Co. Wicklow. **2b)** Overview of the full water course from the Goldmines River tributary to the mouth of the Avoca River, Co. Wicklow. The Goldmines River joins the Avoca River, at Woodenbridge, Co. Wicklow.

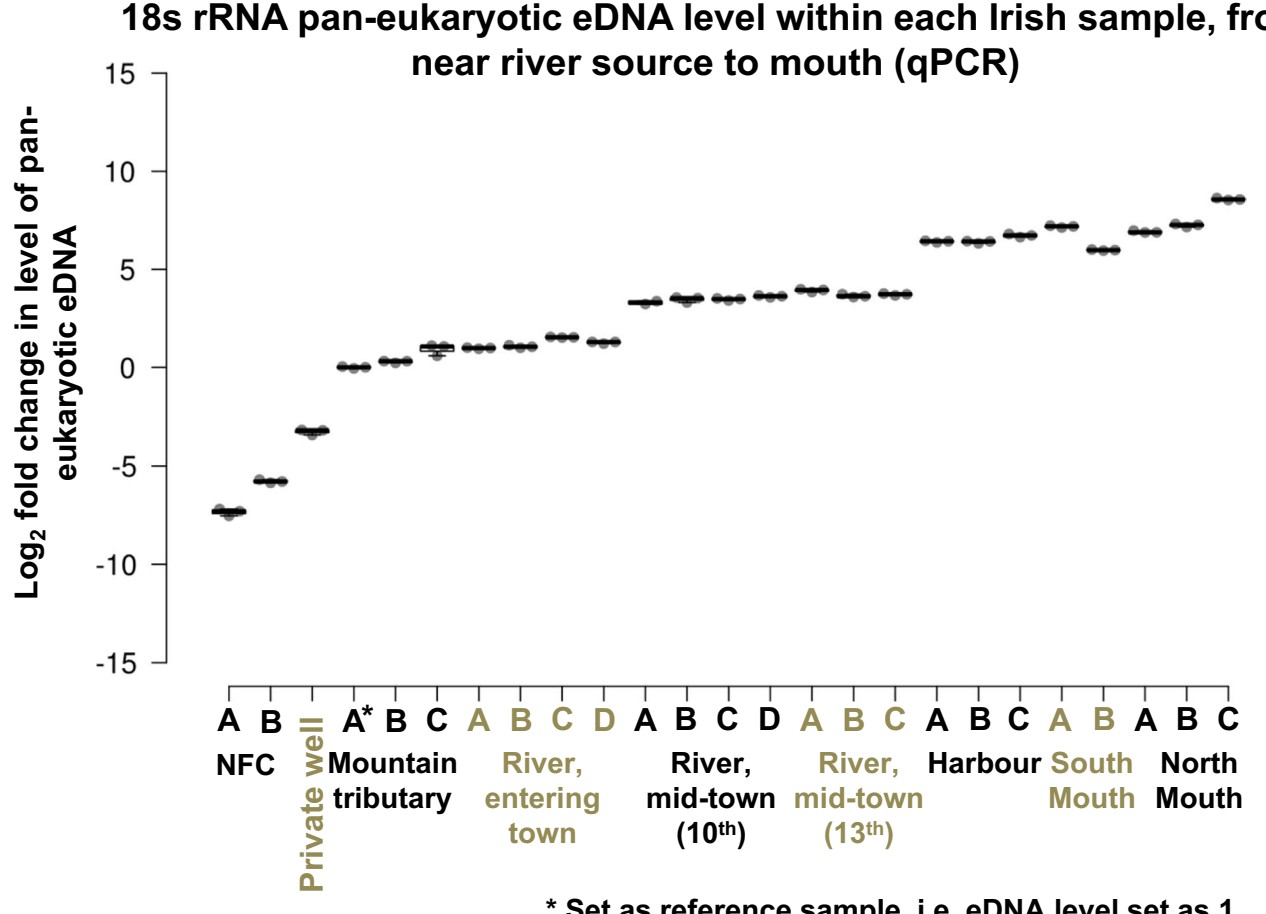

**18s rRNA pan-eukaryotic eDNA level within each Irish sample, from near river source to mouth (qPCR)**

\* Set as reference sample, i.e. eDNA level set as 1.

**Extended Data Fig. 3 | Pan-eukaryotic eDNA levels within each Irish water eDNA sample.** Relative Quantity of 18 s rRNA pan-eukaryotic eDNA levels within each Irish eDNA sample, from near river source to mouth, as assessed by qPCR. Each qPCR reaction is a 10 μl reaction containing 1 μl of extracted eDNA template. For filtered water volumes and elution volumes see Supplemental Table 2.

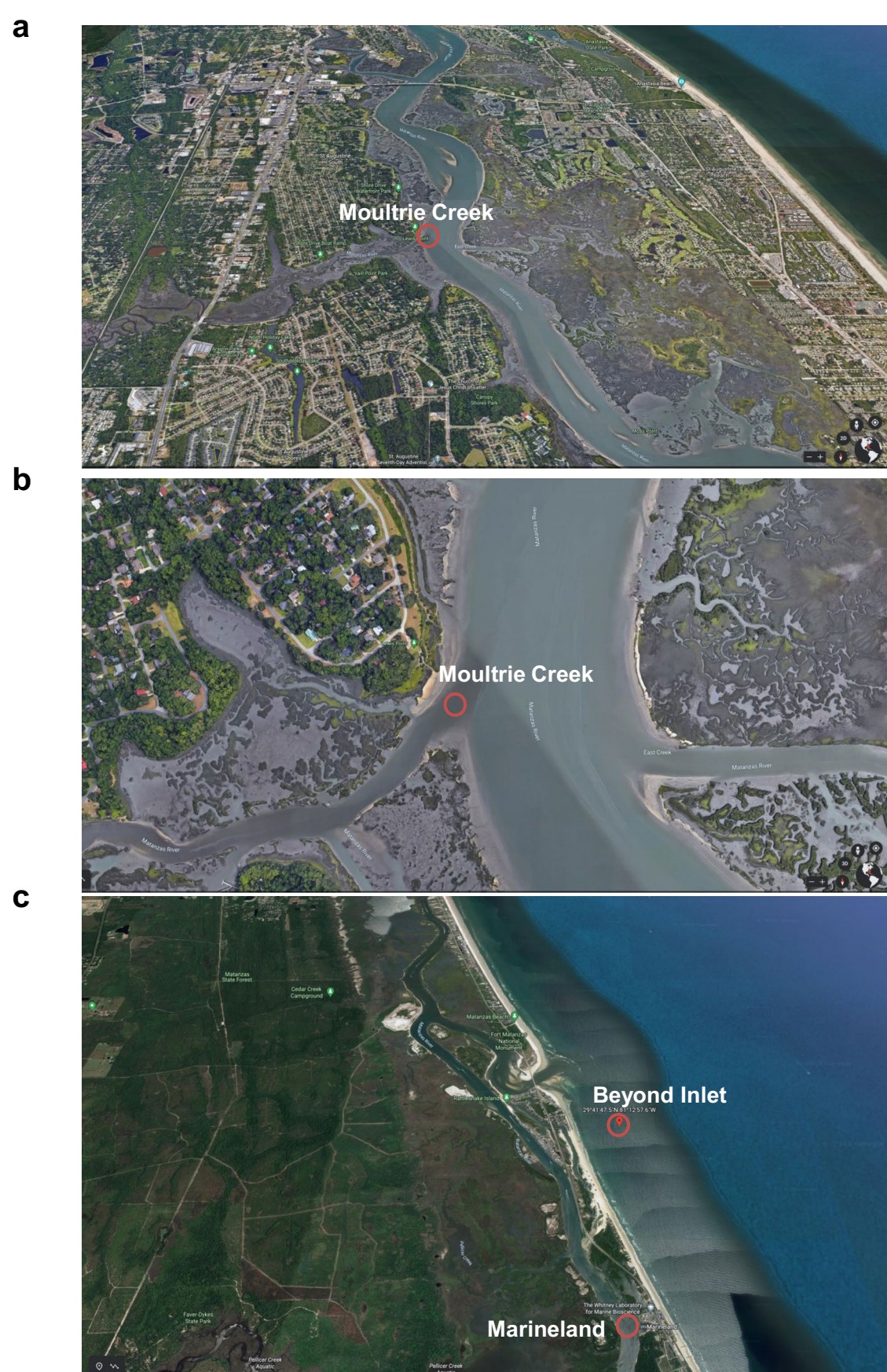

**Extended Data Fig. 4 | See next page for caption.**

**Extended Data Fig. 4 | Satellite imagery view (Google Earth), showing the locations of the Florida 2022 sampling sites.** Satellite imagery view (Google Earth, Map data ©2021 Google and TerraMetrics), showing the locations of the Florida 2022 sampling sites. **4a)** Overview of the Moultrie Creek sampling site, with surrounding human habitation of the city of St. Augustine visible. **4b)** Zoomed view of the Moultrie Creek sampling site. **4c)** Overview of the Beyond inlet and Marineland sampling sites.

**a**

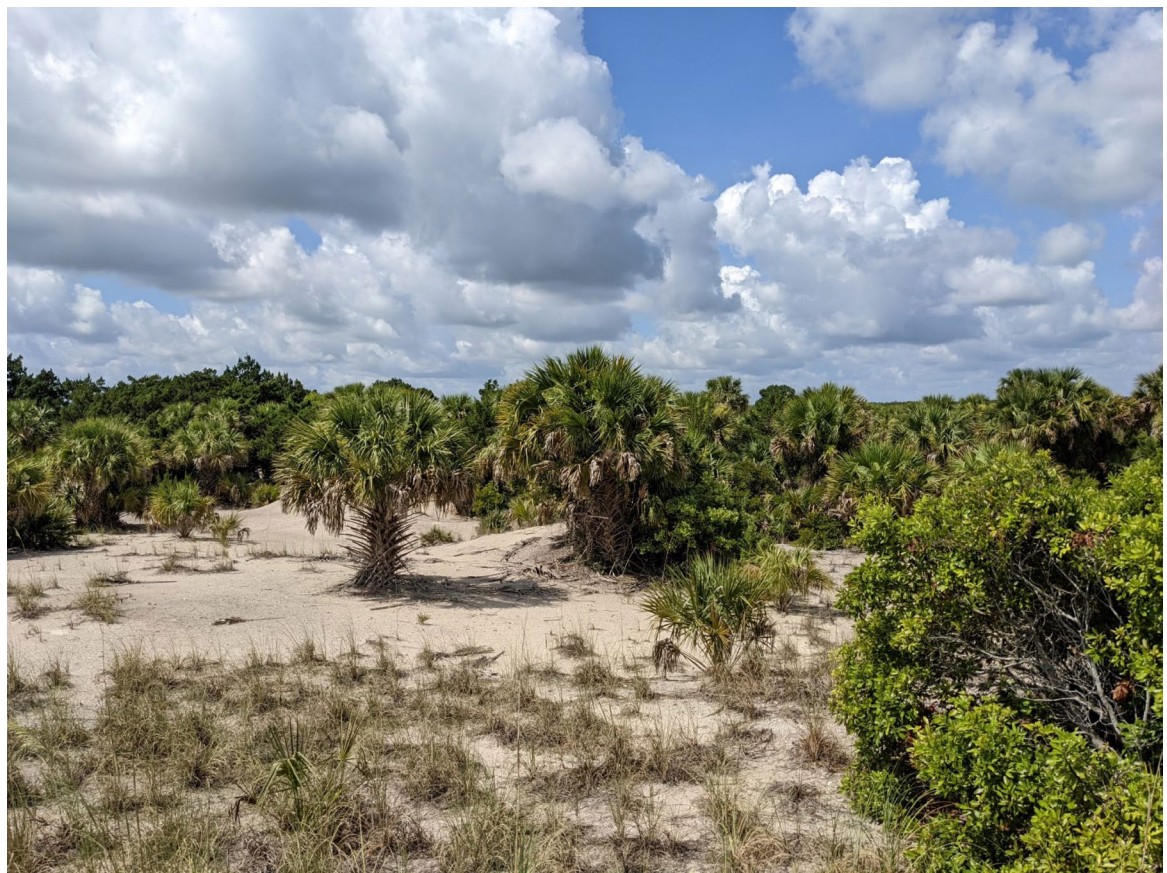

**b**

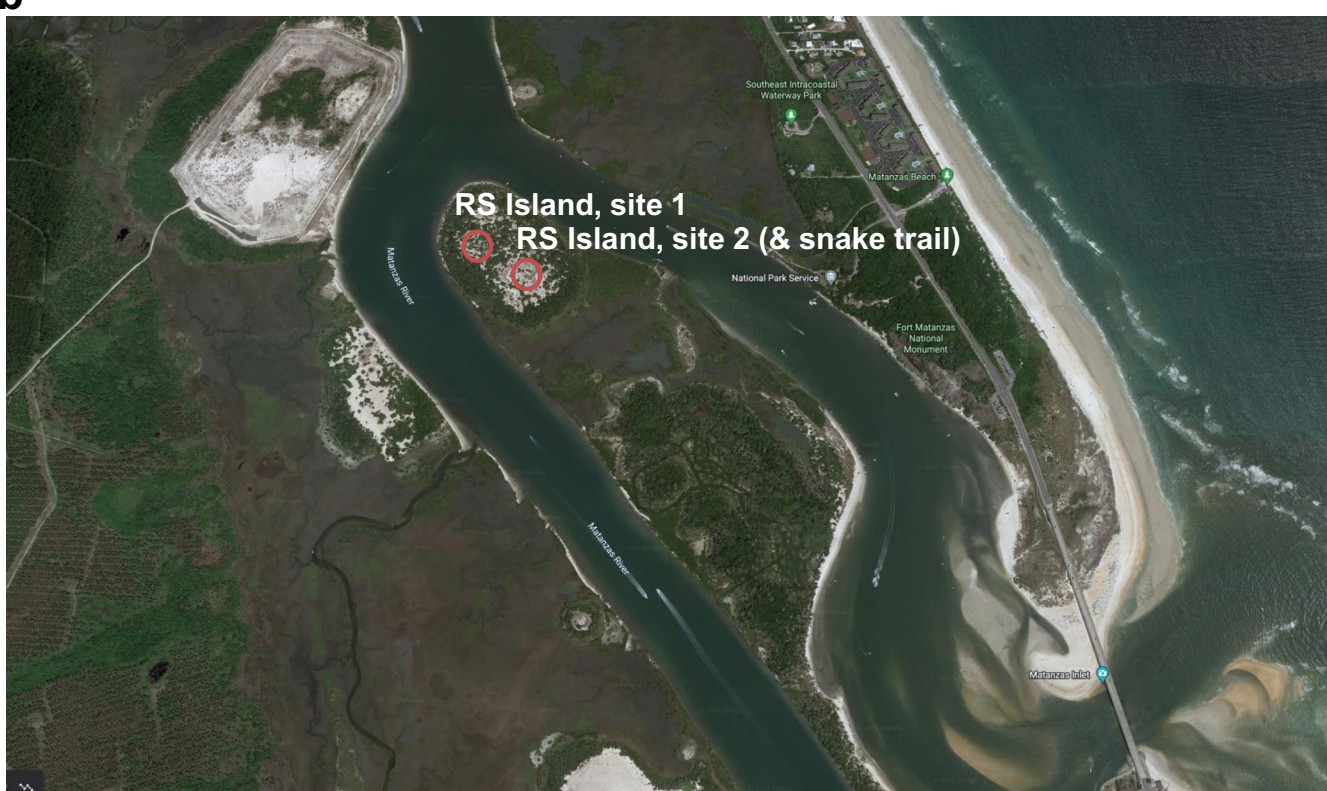

**Extended Data Fig. 5 | Restricted human access sand sampling site. 5a)** Photo of Rattlesnake (RS) Island site 2, at the Fort Matanzas National Monument, managed by the US National Park Service (NPS). **5b)** Satellite imagery (Google Earth, Map data ©2021 Google and TerraMetrics) of the Rattlesnake Island sampling sites. Note that sand samples were also taken from a snake track in the sand at site 2.

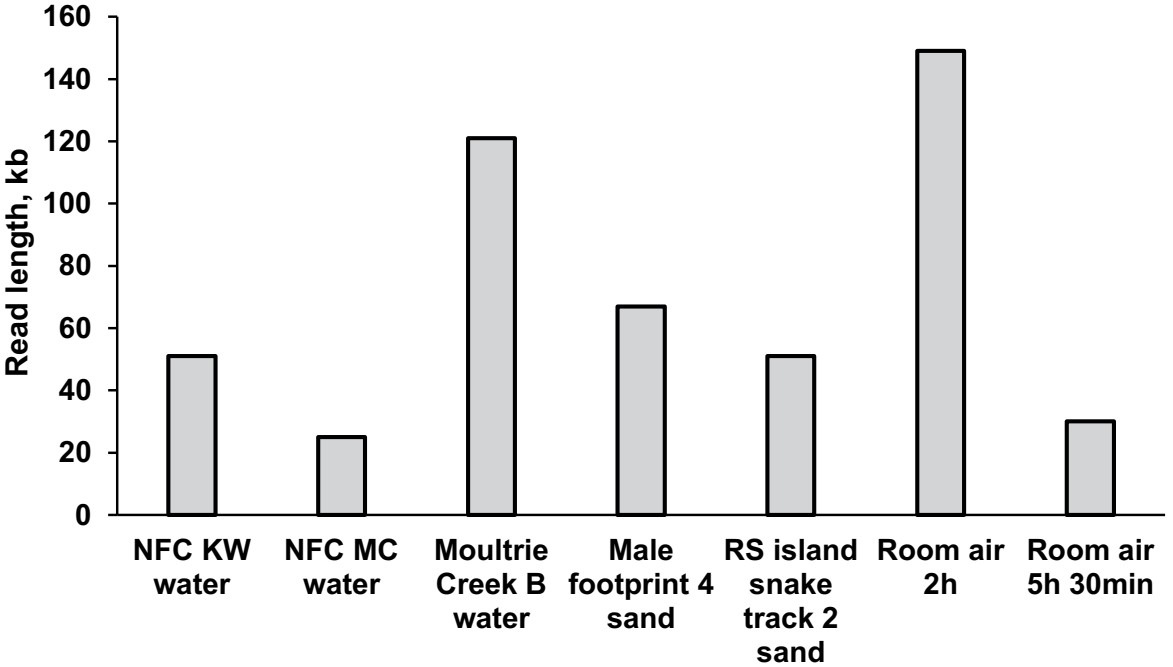

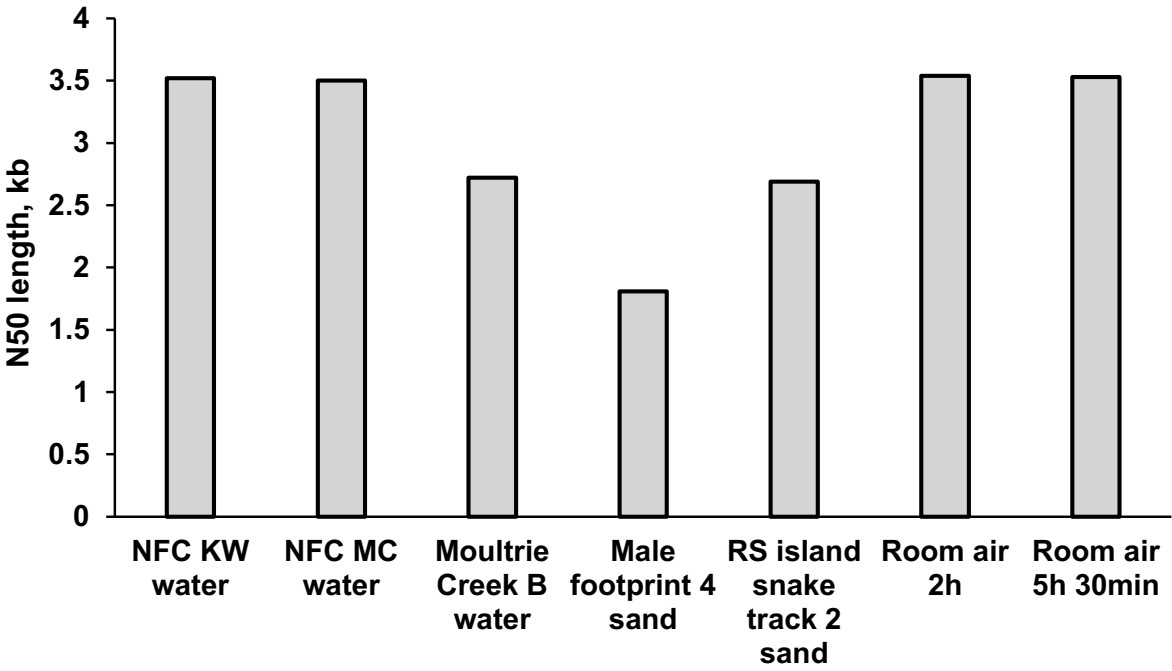

**Extended Data Fig. 6 | Oxford Nanopore long read shotgun sequencing study of eDNA samples post human DNA qPCR-based quantification.** **6a)** Longest individual human reference genome (T2T) aligning read (in kb) generated from each nanopore sequenced sample. **6b)** N50 length (in kb)

from each nanopore sequenced sample, for total sequenced reads, that is pre any reference genome alignment. Note, negative field controls had reduced overall numbers of reads (not reflected in N50 values) reflecting a lack of DNA in the sample (Supplemental Table 3).

**a**

**Prominent cancer-associated genes with or adjacent to deletions detected by gnomAD from Moultrie Creek B water eDNA sample**
(Oxford Nanopore shotgun long read sequencing)

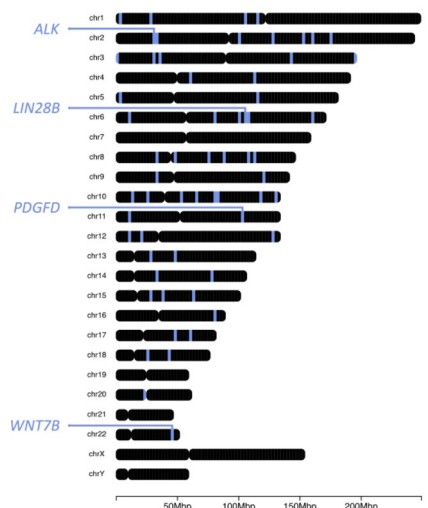

**b**

**Human structural variants (GRCh38 reference genome), detected by EPI2ME structural variant caller**
(no enrichment Oxford Nanopore shotgun sequencing)

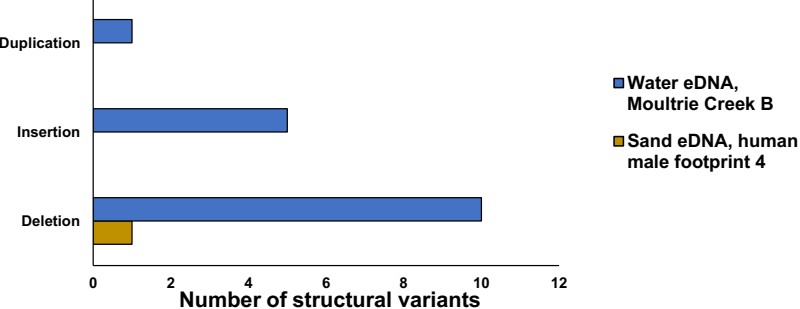

**c**

**Exome enrichment eDNA-seq, human genome (T2T) aligning reads**
(no enrichment Oxford Nanopore shotgun long read sequencing & Illumina Exome enrichment)

**DNA bases that align to human genome, in each exome sample**
(incl. negative field controls)

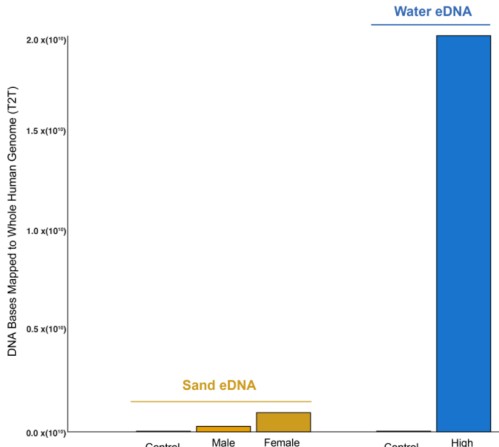

**Total number of bases that align to human genome**

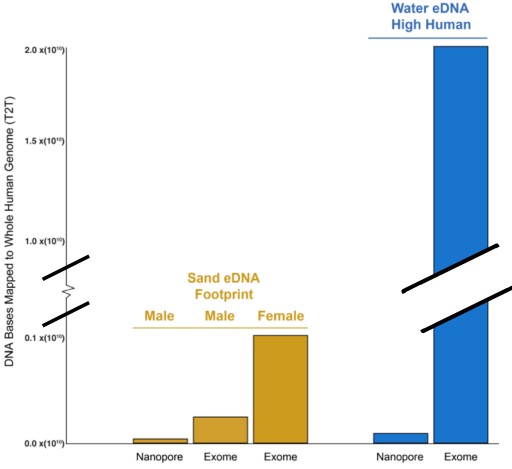

**Extended Data Fig. 7 | See next page for caption.**

**Extended Data Fig. 7 | Additional nanopore deletion and structural variant analysis, and Illumina exome enrichment characterisation.** (7a) Prominent human cancer-associated genes with detections within or adjacent to the gene detected (gnomAD database) detected in Moultrie Creek B nanopore shotgun water sample. Deletion location denoted by blue shading. Full details of all detected deletions can be found in Supplemental Table 5. (7b) Structural variant types identified in nanopore eDNA sequence data identified by the EPI2ME sniffles-based structural variant caller. (7c) Illumina exome enrichment versus shotgun nanopore sequencing data. Left: Total number of DNA bases mapped to the human genome (minimap2) for all exome samples, including negative field controls (Moultrie creek negative field control water eDNA and Rattlesnake Island site 2 sample 1 [no human site] sand eDNA). Right: Total number of DNA bases mapped to the human genome (minimap2) for all human exome eDNA samples, and the corresponding shotgun nanopore data (where the sample was sequenced with both approaches). Note, information is in bases not in reads because bases account for variable read length between nanopore and Illumina sequencing. For mapped read and mapped base information for all analysed samples see Supplemental Table 4.

**a**

### Metagenomics taxonomy (human & microbe) high human water
(no enrichment Oxford Nanopore shotgun long read sequencing, Moultrie Creek B)

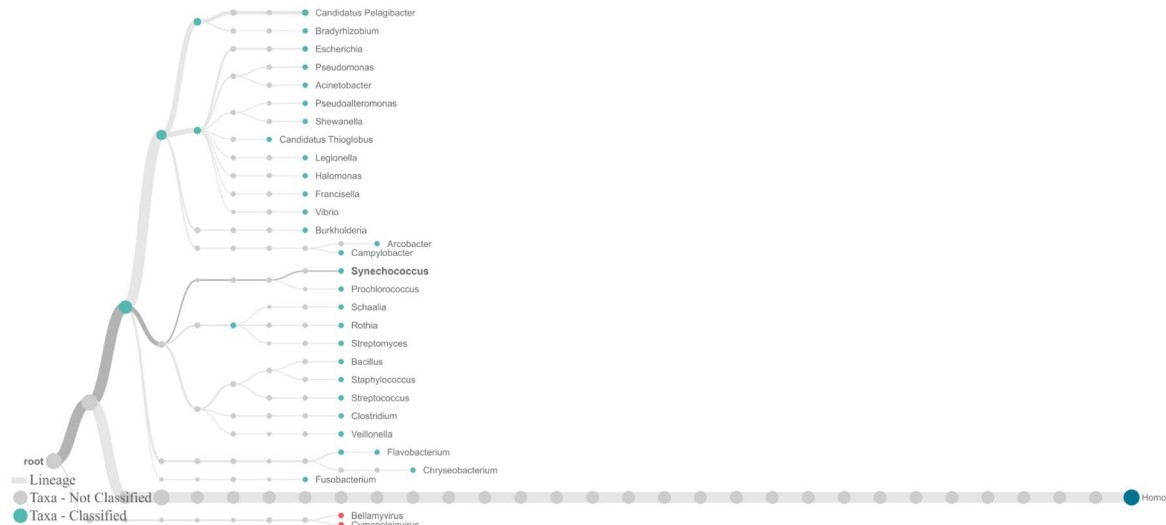

**b**

### Metagenomics taxonomy (human & microbe) human male footprint sand
(no enrichment Oxford Nanopore shotgun long read sequencing)

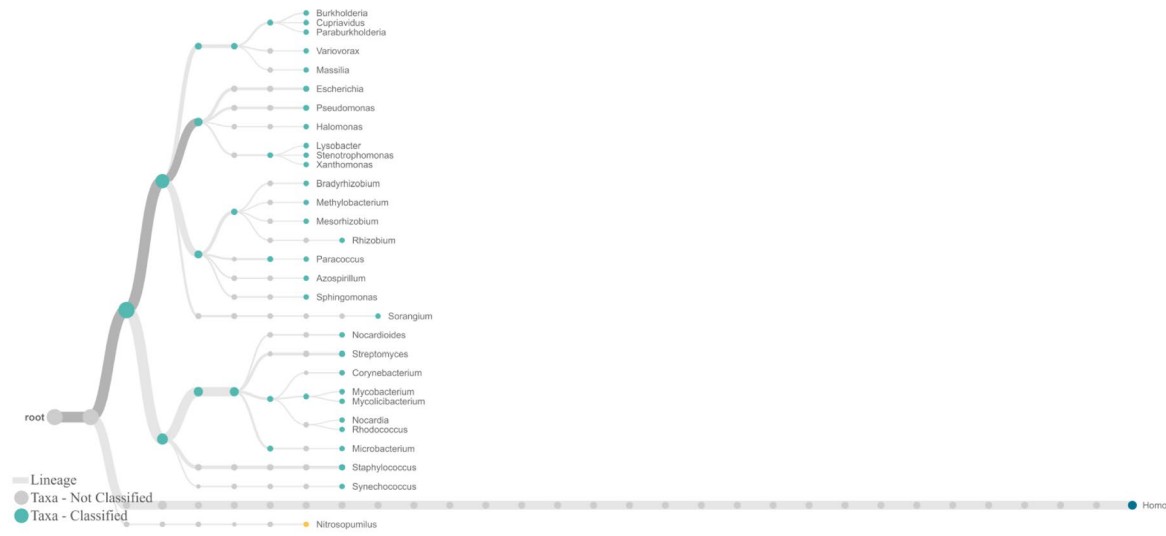

**c**

### Green turtle eDNA from room air sampling, species-specific qPCR
(16s rRNA gene assay)

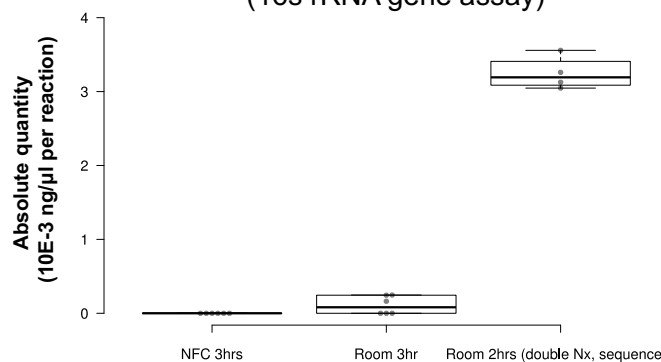

**Extended Data Fig. 8 | See next page for caption.**

**Extended Data Fig. 8 | Metagenomic analysis of nanopore data and qPCR-based green sea turtle species-specific qPCR-based quantification from room air eDNA samples. (8a)** Metagenomic taxonomy of microbial and human aligning reads from the high human site eDNA water sample (Moultrie Creek B), inclusion cut-off on tree, each branch contains at least 0.01% of reads. **(8b)** Metagenomic taxonomy of microbial and human aligning reads from the human footprint sand eDNA sample (human male footprint 4), inclusion cut-off on tree, each branch contains at least 0.01% of reads. **(8c)** qPCR-based species-specific quantification of green sea turtle (*C. mydas*) eDNA from room air samples. Absolute quantity (10E-3 pg/µl per reaction) of green sea turtle eDNA per sample. A 499 bp synthetic gene fragment was used for standard curve generation. Each qPCR reaction is a 10 µl reaction containing 1 µl of extracted eDNA template.

**a**

### Genome mapping of whole green turtle genome aligning reads from room air eDNA samples
(no enrichment Oxford Nanopore shotgun long read sequencing)

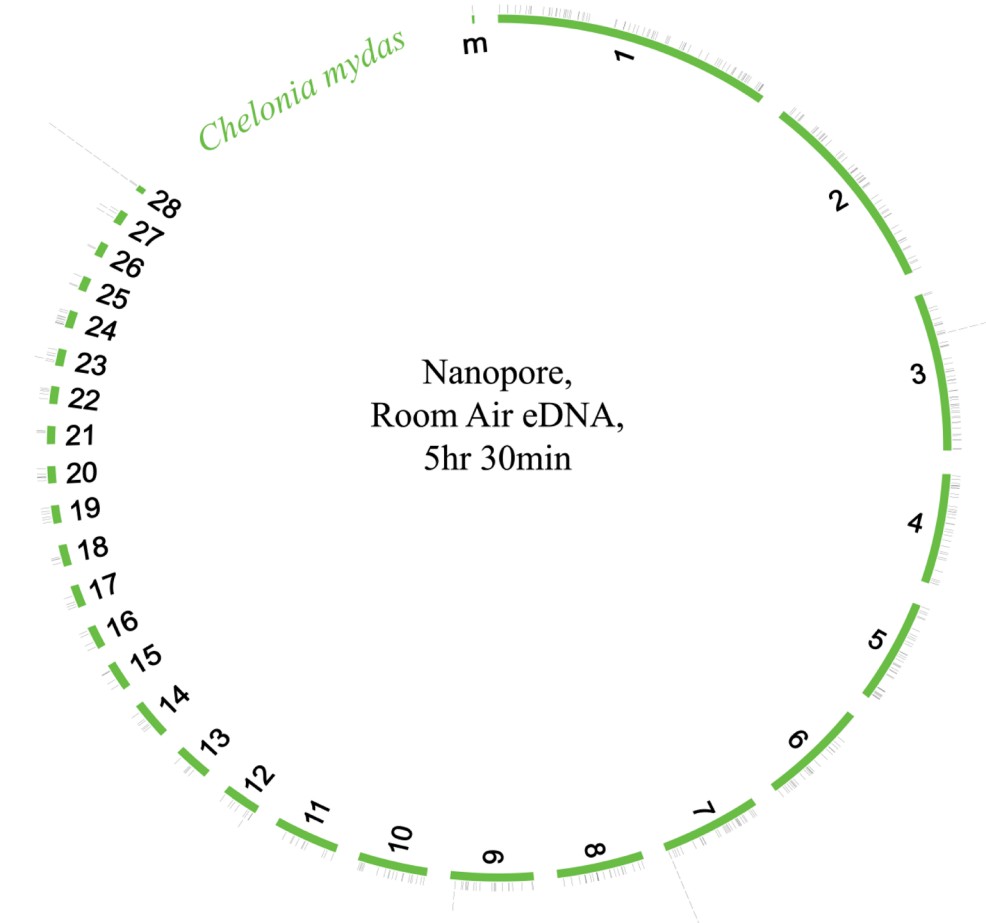

**b**

### Species specificity of human qPCR eDNA assays

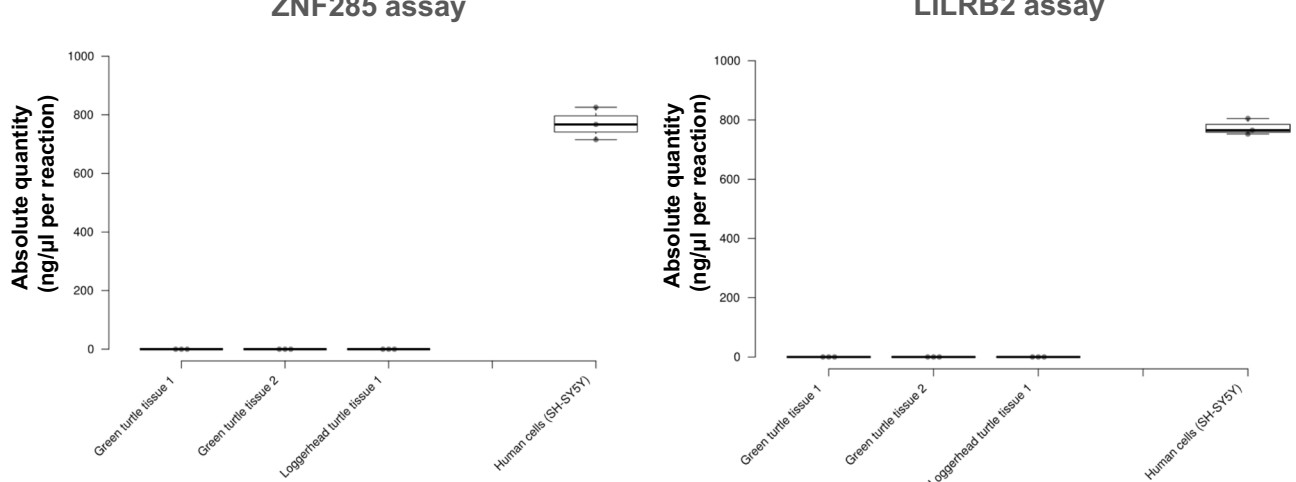

**Extended Data Fig. 9 | See next page for caption.**

**Extended Data Fig. 9 | Green sea turtle genome coverage map from room air eDNA, and human qPCR assay cross-reactivity qPCR.** (**9a**) Green sea turtle (*C. mydas*) genome alignment map of the green turtle aligning reads (minimap2) from the 5 hr 30 min room air eDNA sample. (**9b**) Human eDNA qPCR assays cross-reactivity test with sea turtle (loggerhead and green) tissue samples. Absolute quantity of human genomic DNA present (ng/μl per reaction).

Each qPCR reaction is a 10 μl reaction containing 1 μl of extracted genomic DNA template. Note that both assays were species-specific with no cross-reactivity (no amplification) in either sea turtle tissue sample. Loggerhead and green sea turtles can be added to the list of 27 other species (from mice to plants, see methods section) experimentally validated with which these human assays do not cross-react.

# Reporting Summary

## Statistics

For all statistical analyses, confirm that the following items are present in the figure legend, table legend, main text, or Methods section.

| n/a | Confirmed | |
|---|---|---|
| ☐ | ☒ | The exact sample size (*n*) for each experimental group/condition, given as a discrete number and unit of measurement |
| ☐ | ☒ | A statement on whether measurements were taken from distinct samples or whether the same sample was measured repeatedly |
| ☒ | ☐ | The statistical test(s) used AND whether they are one- or two-sided<br>*Only common tests should be described solely by name; describe more complex techniques in the Methods section.* |
| ☒ | ☐ | A description of all covariates tested |
| ☒ | ☐ | A description of any assumptions or corrections, such as tests of normality and adjustment for multiple comparisons |
| ☒ | ☐ | A full description of the statistical parameters including central tendency (e.g. means) or other basic estimates (e.g. regression coefficient) AND variation (e.g. standard deviation) or associated estimates of uncertainty (e.g. confidence intervals) |
| ☒ | ☐ | For null hypothesis testing, the test statistic (e.g. *F*, *t*, *r*) with confidence intervals, effect sizes, degrees of freedom and *P* value noted<br>*Give P values as exact values whenever suitable.* |
| ☒ | ☐ | For Bayesian analysis, information on the choice of priors and Markov chain Monte Carlo settings |
| ☒ | ☐ | For hierarchical and complex designs, identification of the appropriate level for tests and full reporting of outcomes |
| ☒ | ☐ | Estimates of effect sizes (e.g. Cohen's *d*, Pearson's *r*), indicating how they were calculated |

*Our web collection on statistics for biologists contains articles on many of the points above.*

## Software and code

Policy information about availability of computer code

| | |
|---|---|
| Data collection | No software was used to collect research data. Supplemental Figure 1's number of shotgun eDNA publications was retrived from PubMed and Web of Science, as follows: Graph of "shotgun environmental DNA" papers in PubMed by year published. A literature search for "shotgun environmental DNA" returned 499 results in NCBI's PubMed (https://pubmed.ncbi.nlm.nih.gov) and 654 results in Clarivate's Web of Science (www.webofscience.com). Search conducted on March 11th 2022. |
| Data analysis | No custom software was used for analysis. All analysis was conducted using previously published software and tools, and referenced appropriately. |

For manuscripts utilizing custom algorithms or software that are central to the research but not yet described in published literature, software must be made available to editors and reviewers. We strongly encourage code deposition in a community repository (e.g. GitHub). See the Nature Portfolio guidelines for submitting code & software for further information.

## Data

Policy information about availability of data

All manuscripts must include a data availability statement. This statement should provide the following information, where applicable:
- Accession codes, unique identifiers, or web links for publicly available datasets
- A description of any restrictions on data availability
- For clinical datasets or third party data, please ensure that the statement adheres to our policy

All Illumina sequenced samples including raw reads are deposited in NCBI (https://www.ncbi.nlm.nih.gov/) under BioProject ID: PRJNA449022 (https://www.ncbi.nlm.nih.gov/bioproject/PRJNA449022). All Oxford Nanopore sequenced samples including raw reads are deposited in NCBI (https://www.ncbi.nlm.nih.gov/) under BioProject ID: PRJNA874696 (https://www.ncbi.nlm.nih.gov/bioproject/ PRJNA874696).

# Field-specific reporting

Please select the one below that is the best fit for your research. If you are not sure, read the appropriate sections before making your selection.

☐ Life sciences  ☐ Behavioural & social sciences  ☒ Ecological, evolutionary & environmental sciences

For a reference copy of the document with all sections, see [nature.com/documents/nr-reporting-summary-flat.pdf](nature.com/documents/nr-reporting-summary-flat.pdf)

# Ecological, evolutionary & environmental sciences study design

All studies must disclose on these points even when the disclosure is negative.

| Study description | Observational study, no treatments. Assessed the number of human reads recovered from environmental DNA shotgun sequencing data. Additionally, collect water, sand and air samples eDNA from areas with high and low human habitation and quantified the level of human eDNA present with species-specific qPCR. Collected Oxford Nanopore long read sequencing of selected intentionally collected human eDNA samples (and Illumina Exome sequencing), including water from areas close to towns and human footprints in beach sand. |
|---|---|
| Research sample | Environmental DNA samples. Samples consisted of water eDNA (DNA recovered form sea water) or beach sand eDNA (DNA recovered form sea turtle nesting beaches).<br>Comprises of complex metagenomic data, not data purely from a single individual or species. Human originating DNA reads are reported here, sea turtle and pathogen DNA reads from these samples has been reported elsewhere (as referenced in the manuscript. Additionally, collected water eDNA samples form areas with high and low human activity, and human footprint beach sand samples and sand samples from an island with restricted human access.  Air eDNA (air filter from rooms with or without humans present), human, sea turtle and turtle pathogen reads for air samples are  reported here, as for air samples no turtle-related information was previously report, and as it demonstrates the feasibility of collecting data from multiple species and pathogens from the same air eDNA sample (which is potentially applicable to human medical settings also). |
| Sampling strategy | Illumina sequencing, sample sizes were dictated by the cost of high-depth shotgun sequencing (non-targeted). The maximum number of samples it was cost effective to sequence were analyzed. n = 20 NGS libraries. For qPCR analysis 68 eDNA samples were collected. Water eDNA was collected from a temperate and sub-tropical region to confirm the applicability of human eDNA in both climates, as there are known links between eDNA stability and climate type. Seven samples were used for Oxford Nanopore sequencing and five samples were used for Illumina exome sequencing, sample number primarily dictated by cost. |
| Data collection | All Illumina samples (shotgun and exome enriched) were sequenced at the University of Florida's (UF) Interdisciplinary Center for Biotechnology Research Core Facilities. All Nanopore samples were sequenced in the Duffy Lab at UF's Whitney Laboratory for Marine Bioscience. Water samples were collected by Jessica Farrell, David Duffy and Todd Osborne. Sand samples were collected by licensed nesting beach permit holders under the supervision of David Duffy or Jessica Farrell, Illumina samples). Human sand samples and air samples were collected by David Duffy in accordance with UF Institutional Review Board ethical approvals. Sand samples from Rattlesnake Island were collected by Jessica Farrell, David Duffy and Samantha Koda, with the permission and guidance of US NPS staff (Kurt Foote and Andrew Rich ). qPCR was conducted by Jessica Farrell, David Duffy and Victoria Summers in the  Duffy Lab at UF's Whitney Laboratory for Marine Bioscience, and by David Duffy in the Conway Core Facility, University College Dublin. |
| Timing and spatial scale | Illumina samples: Ad hoc. Initial set of 4 water samples (1 pooled tank library, and 4 wild water samples) were sequenced to confirm the viability of the approach (2017). Another set of 6 water samples, form more geographically diverse locations were sequenced (2021), and finally a set of sand samples (12 libraries) were sequenced. Sequencing was conducted in tandem with the optimization and development of sea turtle and pathogen qPCR assays. Water sampling sites range from northeast Florida to southern Florida (Florida Keys), all sequenced sand samples are from northeast Florida sea turtle nesting beaches, with the exception on one tank library and one rehab sand sample which were collected at the University of Florida's Whitney Laboratory for Marine Bioscience and Sea Turtle Hospital in northeast Florida.<br>qPCR & Nanopore samples: Intentional human eDNA sampling was conducted between May and July 2022 (water and sand) at sites in Northeast Florida, US and in East Ireland, room air eDNA sampling was conducted between Oct. and Nov. 2022 in Northeast Florida, US. |
| Data exclusions | No data were excluded from the analysis. Low quality reads were trimmed from each sample as part of the analysis pipeline, as is standard convention, but data form all sequenced samples are reported. Raw reads (prior to trimming) have also been publicly deposited, as is convention. |
| Reproducibility | Illumina & Nanopore samples: All samples represent individual replication, being independent of each other. No treatments etc. were conducted, these are observational data. Sequencing was done over 9 distinct library prep. and machine runs, on two different Illumina sequencers (HiSeq and NovaSeq) and a Nanopore MinION.<br>For qPCR, both biological and technical replicates were used. In addition, the human related eDNA water sampling was reproduced in both Florida and Ireland to confirm the reproducibility of this approach. Each sample set was analysed  in its respective country, demonstrating reproducibility across labs and spatial scales. |
| Randomization | No groups were assigned. Samples were either water eDNA, sand eDNA or air eDNA depending on the environmental type originally samples. Illumina samples: Samples were random for human eDNA reads. Sequenced water samples were selected based on geographic spread and potential of the presence of sea turtles in a given location. Sand eDNA samples were selected based on sea turtle nesting activity, not human activity.<br>qPCR & Nanopore samples: Water sampling sites were selected based on their proximity to human habitation. Sand eDNA samples |

were either from human footprints or from non-footprint sand. Room air eDNA samples were either from rooms with or without humans present.

**Blinding**

Blinding was not performed, although the data was not collected solely for human eDNA applications (Illumina), and no human eDNA information was available to investigators prior to sequencing/qPCR and analysis (Illumina, qPCR and Nanopore).

**Did the study involve field work?**    ☒ Yes    ☐ No

# Field work, collection and transport

**Field conditions**

Nesting beach sand samples were collected during the summer sea turtle nesting season in northeast Florida, during dry weather. Sand footprints and non-footprint sand samples were collected in the summer during dry weather.
Water samples were collected during in-water sea turtle surveying expeditions, and sampling trips to locations with high and low human habitation levels.

**Location**

State of Florida, location of sampling described in the manuscript and related sea turtle paper cited in the manuscript. For all (non-rehab) water samples, these were collect just below the surface (less than 1m depth). For all beach sand samples, these were collected by dragging the 50ml tube along the surface of the sand, or from nest spoil heaps.
The location of all intentional human eDNA sampling (Ireland and Florida) has been provided in the Supplemental Figures.

**Access & import/export**

No import/export permits were required as samples did not cross state or country lines. Samples were analyses in the country they were collected in. Permitting re the original samples is described in the sea turtle paper related to these samples, essentially: Sampling was carried out under permit number MTP-22-236 from the Florida Fish and Wildlife Conservation Commission (FWC) and with ethical approval for tissue sampling from the University of Florida's Institutional Animal Care and Use Committee (IACUC). Rehabilitation and nest patrol activities are conducted under FWC permit numbers MTP-21-228, MTP-21-103, MTP-21-084, MTP-21-029, MTP-21-041, MTP-21-140, MTP-21-023, MTP-21-046 and MTP-21-101 (rehabilitation and conservation activities were in no way impacted by this study). The Inwater Research Group conduct in-water sea turtle monitoring surveys under NMFS permit numbers 19528 and 16598 and FWC permit numbers MTP-18-125 and MTP-18-139. The Florida Hawksbill project conducts conduct in-water sea turtle monitoring surveys under NMFS permit number 22988 and FWC permit number MTP-21-077.
Human-related eDNA sampling was conducted with University of Florida Institutional Review Board (IRB-01) ethical approval under project number IRB202201336. Sampling at the Fort Matanzas National Monument (Rattlesnake Island) was conducted under a United States Department of the Interior National Park Service permit, permit number FOMA-2022-SCI-0003.

**Disturbance**

No disturbance was caused by the study as these are environmental samples. Water samples were obtained from rehab tanks while the turtle was not present. Sand samples were obtained after nesting events and after nest evaluations (conducted by already permitted nesting beach patrols). Human samples were obtained directly from the water (relatively small volumes) and from sand (relatively small volumes).

# Reporting for specific materials, systems and methods

We require information from authors about some types of materials, experimental systems and methods used in many studies. Here, indicate whether each material, system or method listed is relevant to your study. If you are not sure if a list item applies to your research, read the appropriate section before selecting a response.

## Materials & experimental systems

| n/a | Involved in the study |
|-----|----------------------|
| ☒ ☐ | Antibodies |
| ☒ ☐ | Eukaryotic cell lines |
| ☒ ☐ | Palaeontology and archaeology |
| ☒ ☐ | Animals and other organisms |
| ☐ ☒ | Human research participants |
| ☒ ☐ | Clinical data |
| ☒ ☐ | Dual use research of concern |

## Methods

| n/a | Involved in the study |
|-----|----------------------|
| ☒ ☐ | ChIP-seq |
| ☒ ☐ | Flow cytometry |
| ☒ ☐ | MRI-based neuroimaging |

# Human research participants

Policy information about studies involving human research participants

**Population characteristics**

Not applicable, as no population characteristics or personal information (beyond being male or female) was recorded from anonymous participants as it was not required due to the study design. The only criteria for inclusion was that the participants were human, and that they were capable of providing informed consent, and fully aware of qPCR and genomic sequencing technologies (as part of being able to provide fully informed consent). Human-related sampling was conducted with University of Florida Institutional Review Board (IRB-01) ethical approval under project number IRB202201336, with all participants providing informed consent. Four participants (three female and one male) provided sand footprint samples and six participants (five female and one male) provided room air samples (pooled room air).

**Recruitment**

No biases. Only the presence of and sequence of human eDNA was analysed, this was assessed from each footprint with no other correlating factors (i.e. no participant medical information or identifying information recorded). Voluntary participants

were sought at the institutional level. All participants providing informed consent. Participation was on a voluntary basis with no compensation received by the participants.

Ethics oversight

Human-related eDNA sampling was conducted with University of Florida Institutional Review Board (IRB-01) ethical approval under project number IRB202201336.

Note that full information on the approval of the study protocol must also be provided in the manuscript.

