## [Peer Review File · Nature Ecology & Evolution]

Peer Review Information

Journal: Nature Ecology & Evolution

Manuscript Title: Inadvertent human genomic bycatch and intentional capture raises novel beneficial applications and ethical concerns with environmental DNA.

Corresponding author name(s): David J. Duffy

Editorial Notes:

Reviewer Comments & Decisions:

Decision Letter, initial version:

10th May 2022

Dear Dave,

As discussed previously, we sent your manuscript entitled "Environmental DNA (eDNA): inadvertent human genomic bycatch raises ethical concerns about consent, privacy and surveillance" out to review by two referees, one specialising in eDNA and the other in privacy, ethics and human genetic materials. You'll see from the reports that they found the manuscript of considerable interest, but they raise significant issues that mean that we cannot offer to publish the manuscript. The principle issue, as raised by reviewer 1, is whether the data presented are suitable to make the point about the potential privacy issues of human genomic bycatch. Concerns regarding potential lab contamination seem to us editorially to be likely difficult to resolve. We are interested in the issues you raise (as illustrated by a previously published Comment touching on some of the ethics issues <https://www.nature.com/articles/s41559-020-01351-6>) and feel that a research paper demonstrating the phenomenon would be a valuable resource, but wonder if it is the case that a research programme would need to be explicitly designed to target these questions rather than using previously derived data. Such a programme could even set out a case study approach of how to integrate ethics approval for human materials into eDNA and as such provide an important step for the field.

So while our decision at this point is unfortunately negative, please keep in touch if you feel that future data might prove fruitful to resolve the present issues with the manuscript--and of course please don't hesitate to let us know if you feel there are any misunderstandings either in the peer review reports or in our rationale for this decision.
Best wishes,

[REDACTED]

Reviewers Comments:

Reviewer #1 (Remarks to the Author):

Dear team,

2I congratulate you on your efforts of re-utilizing existing samples and data on turtle eDNA to answer a completely new question, namely if and to what degree human populations can be monitored using eDNA. While this is a very urgent and important research topic, I fear that the presented study presents several problems: I will present some of my major concerns and would be happy to discuss them with the authors; I at this stage can however not recommend the study for publication.

Major comments:

- This study is purely based on samples/data from past research, which I in general fully support to make research as efficient as possible. I, however, see a few problems in the study design and sample size for answering the posed question.
- The negative controls (especially from the sand samples) have been taken as negative controls for the sea turtle, which means that this sample expectedly contains human DNA and contamination during sampling can not be ruled out (it is actually interesting that the sand negative control shows the least amount of human DNA, both in terms of the full human genome and the Y chromosome, which is something that would not have been expected, see Supplementary Table 1 - do you have an explanation for that?). For the water samples, the negative control consisted of MilliQ water that was brought to the side and then filtered in situ, making sure that no human contamination had happened from the point of filtering to the end result. I am, however, wondering if contamination could have happened during sampling, e.g. by being in touch with the water while sampling (which would have not been a concern for the original turtle study this experiment had been designed for)? This could also explain why the samples from wild water show more human DNA content than the rehabilitation tank samples, since I would expect the latter to contain more human DNA due to humans working with and around the turtles, but maybe they can be sampled more easily without directly getting in touch with the water?
- It is also striking that the human DNA content in wild vs. tank water samples could be explained by the employed sequencing machine, i.e. NovaSeq vs HiSeq. While I am not sure if the modest increase in mismatch rates when performing alignments with NovaSeq in comparison to HiSeq, and the general systematic bias that is expected when changing protocols, could result in such a substantial difference, I would like to ask the authors to look into this.
- Coming back to the negative controls, the authors do not report any RPTM in Figure 1 for the water control; when looking at Supplementary Table 1, I understand that this is the case because less than TM (ten million) reads have been sequenced for this sample. The authors correctly state in the table legend that "The lack of sequenced reads reflects an overall lack of DNA in this control sample." However, 196,876 reads have been obtained for this negative control, out of which 3,184 align to the human genome - which suggests strong human contamination. While no RPTM can be calculated, this still seems to be an important result, which I would like the manuscript to discuss. [On a minor note, how are the % alignments calculated since, e.g., 3,184/196,876 is not 1.59% but 1.62%? Is this due to partial alignments or slightly variable read lengths?]
- I think it's an important idea to write a landmark paper about sampling human DNA in eDNA studies. I however think that the sample size of this study is too small to create such a landmark study; while I don't want to criticize small sample size in general, I feel that in this case technical, spatial and temporal replicates could have been taken to really study the content of human DNA in eDNA data. Especially understanding the spatial axis would be important since the "wild" samples taken by the team are reportedly relatively far away from any human civilization. Would the ratio of human DNA

2increase when approaching civilization?

- Have the authors conducted a metagenomic mapping approach against a database of expected mammals? I am wondering if some reads would map to other mammals in such an approach - something that could explain the difference in human read assignment when only using the more human-specific Y chromosome.
- Have the authors looked into coverage of the human genome in their shotgun data? It would be interesting to know if most of it is, e.g., mitochondrial, and to give an idea of what sort of information could be retrieved using this data. Especially given that short-read sequencing is being used in this approach and one would expect to sequence a pool of different human individuals, it might be difficult to actually make any individual-specific inferences. I, however, see that this is not the focus of this study, but rather showing that human DNA can be discovered at all.

Minor comments:

- I would suggest rewriting the introduction a bit to make clear that it is the shift from metabarcoding-based approaches to shotgun sequencing that potentially enables retrieving enough genetic data to identify human individuals/make phenotypic predictions. At the moment, those points read a bit unconnected.
- ll. 157: The authors suggest correlating human susceptibility loci with pathogen load in environmental samples; while this might be a prospective at some point in the future, I think it's necessary to discuss that this is a very difficult endeavor given that each sample is a pooled sample with intrinsically high variance in both phenotypic as well as genotypic measurements, plus the environment will have a major impact on any detected correlation, given that pooled samples will per definition come from different locations.
- ll. 185: I understand the author's intentions of mentioning these examples, but I would strictly adhere to the problem of genetic data sharing in this article (since it's a big enough problem on its own).
- Please provide any supplementary data as tsv/excel files so that they can easily be re-analyzed by the reader. Please also include any raw data for the Y chromosome coverage.

Thank you again for your work and I am sorry that I can't be more positive at this stage. I find it very encouraging that you are looking into this very important research subject, but would either recommend tackling a few of the mentioned concerns (I am happy to discuss details or if I misunderstood something) or "toning the message of the paper down" from a landmark paper to a pilot study that for the first time shows the vast amount of human data that can be found in eDNA shotgun data.

I wish you all the best with your exciting research,
Lara Urban

Reviewer #2 (Remarks to the Author):

3This piece addresses a timely and important topic, given increasing capabilities and interest in sampling and analyzing wastewater for public health or other purposes and the often-similar ethical issues that analysis may raise. My comments here are restricted to the ethical discussion in your article.

First, I encourage you to make the connection between human genetic bycatch (HGB) and wastewater monitoring arising from the Covid-19 pandemic earlier in the article. You allude to this use at the outset of your article, but only mention it explicitly (and briefly) on page 4 of your manuscript. As part of motivating the reason to consider these ethical issues now, the incredible investments that have been made to develop wastewater monitoring during the pandemic—and the ways in which human genetic material is collected and may be exploited—are surely relevant.

Second, I want to suggest a response to one of the ethical issues you raise. You appear concerned about “with whom should the responsibility” lie to police the adequate filtering and scrubbing of eDNA sequencing to shield human privacy in public data repositories. But it seems to me that norms about policing protections for human privacy ought already to be well established in other fields. Who is responsible for policing the adequate filtering, scrubbing, or de-identifying of human genetic material in instances in which human subjects are unquestionably involved? It would seem sensible to locate responsibility for that work with the same entities or actors when eDNA is at issue, too. But perhaps this is too facile a response. If so, it might be helpful to explain why.

Third, I recommend that, in addition to identifying the ethical issues that arise because eDNA captures human DNA too, you explicitly call for researchers, funders, and other stakeholders to develop responses to these ethical issues before the technology at issue becomes even more wide-spread and entrenched and before human genetic bycatch comes to be exploited in the ways you suggest. Ex ante planning is crucial for ensuring that law and ethics stay ahead of technology, and you should be explicit about that.

Fourth, it may be worthwhile to identify circumstances under which HGB is more or less likely to give rise to ethical concerns. Wastewater monitoring may raise serious privacy concerns about human genetic privacy because our most frequently visited places don't change very much and our home addresses are readily knowable. That means that human genetic data retrieved from a manhole may be traceable to a small group of possible human contributors with relative ease. That is less likely to be the case where seawater is collected from a popular tourist beach. You identify some instances in which HGB raises serious ethical concerns, as in the exploitation of HGB to identify genetic data of populations less willing to participate knowingly and voluntarily in genetic research. It may be helpful to try to demarcate the circumstances under which ethical issues are more or less likely to arise (small, stable populations of humans contributing to an eDNA watershed would likely be one).

Fifth, turning to Table 1:

- I recommend that “Unintentional consequences” would be better as “Unintended consequences.”
- It is not entirely apparent how “genomic harvesting” differs from “genetic surveillance – illegal/unethical collection of whole ethnic groups/populations.” Both describe genomic harvesting. Is the distinction you seek that the first identifies the harm of lack of consent, while the second identifies a more specific harm of exploitation that is experienced by minoritized groups? If so, I am not sure

4the latter is genuinely a harm of "surveillance" as opposed to a harm of "harvesting." And if so, please work to clarify the distinctions between these two identified harms. I think both are present, and both merit identification.

- The last harm identified, about "bio-piracy," includes "flora/fauna genetic data." How is this a function of HGB? Please clarify.

**Although we cannot publish your paper, it may be appropriate for another journal in the Nature Portfolio. If you wish to explore the journals and transfer your manuscript please use our <https://mts-natecolevol.nature.com/cgi-bin/main.plex?el=A6Cn7GTZ7A3BAGa7X7A9ftdxx5oQaAUk4keXKN8TEWwZ> manuscript transfer portal. You will not have to re-supply manuscript metadata and files, but please note that this link can only be used once and remains active until used. For more information, please see our http://www.nature.com/authors/author_resources/transfer_manuscripts.html?WT.mc_id=EMI_NPG_1511_AUTHORTRANSF&WT.ec_id=AUTHOR manuscript transfer FAQ page.

Note that any decision to opt in to In Review at the original journal is not sent to the receiving journal on transfer. You can opt in to [In Review](https://www.nature.com/nature-research/for-authors/in-review) at receiving journals that support this service by choosing to modify your manuscript on transfer. In Review is available for primary research manuscript types only.

** For Nature Research general information and news for authors, see <http://npg.nature.com/authors>.

Decision Letter, first revision:

26th May 2022

Dear Dave,

Thank you for your letter asking us to reconsider our decision on your Brief Communication entitled "Environmental DNA (eDNA): inadvertent human genomic bycatch raises ethical concerns about consent, privacy and surveillance.". After careful consideration we have decided that we would be willing to consider a revised version of your manuscript.

Along with your revised manuscript, you should also submit a separate point-by-point response to all of the concerns raised by the reviewers, in each case describing what changes have been made to the manuscript or, alternatively, if no action has been taken, providing a compelling argument for why

5that is the case. If we feel that a substantial attempt has been made to address the reviewers' comments, this response will be sent back to the reviewers - along with the revised manuscript - so that they can judge whether their concerns have been addressed satisfactorily or otherwise.

I should stress, however, that we would be reluctant to trouble our reviewers again unless we thought that their comments had been addressed in full--it sounds like the contamination issue is easily dealt with, but we will also need to see evidence of a more detailed sampling strategy to answer the questions posed, including technical, spatial and temporal replicates as recommended by reviewer 1. And we will need to see reviewer 2's comments on the ethical implications addressed as well.

- ensure it complies with our format requirements for Brief Communications as set out in our guide to authors at www.nature.com/natecolevol/authors/index.html

- state in a cover note the length of the text, methods and legends; the number of references and the number of display items.

Please ensure that all correspondence is marked with your Nature Ecology & Evolution reference number in the subject line.

Please use the following link to submit your revised manuscript:

[REDACTED]

I would appreciate it if you could tell me if you think you will be able to submit a revised manuscript, and also the likely timescale.

I look forward to hearing from you soon.

[REDACTED]

Author Rebuttal, first revision:

We would like to thank all the reviewers for their careful consideration of the manuscript and for their generous, thoughtful and highly constructive comments. In line with the Reviewers' comments we have now carried out extensive additional human eDNA experiments, where human eDNA capture was the sole focus of the research. This includes species-specific qPCR-based assessment of human eDNA loads from sites with high and low human habitation in both sub-tropical and temperate environments, comparing human eDNA loads in human sand footprints and from areas with restricted human access, and long read Oxford Nanopore sequencing of human eDNA. We have also clarified that while not the target of our original eDNA collection and not involved in site selection, we did always plan on analyzing the human aligning reads from our original Illumina shotgun data (it was part of the initial study design), and as such we took relevant precautions to avoid investigator contamination. Additionally, we have also implemented all of the requested changes to the manuscript, which we feel have helped to greatly improve the manuscript.

All author initial responses are provided in blue text.

- A question for both reviewers and the editor. We have publicly deposited non-human sequencing samples, but would like your opinion as to whether we should also deposit the high human water sequence and the human footprint data?

Reviewers Comments:

Reviewer #1 (Remarks to the Author):

Dear team,

I congratulate you on your efforts of re-utilizing existing samples and data on turtle eDNA to answer a completely new question, namely if and to what degree human populations can be monitored using eDNA. While this is a very urgent and important research topic, I fear that the presented study presents several problems: I will present some of my major concerns and would be happy to discuss them with the authors; I at this stage can however not recommend the study for publication.

We thank the Reviewer for their support for our study and the importance of the research topic, positive comments, and constructive criticism. We also appreciate that you see the importance of HGB to the field of eDNA. Taking all reviewer comments into account we have designed novel human-centric eDNA sampling and analysis to demonstrate the feasibility of intentional human eDNA capture also (please see the expanded results section of the revised manuscript). This novel sampling and analysis was conducted solely for the purposes of this human eDNA study. This includes intentional human eDNA water sampling in sub-tropical (Florida) and temperate (Ireland) locations, the use of human species-specific qPCR assays to quantify the level of human eDNA in each novel sample and additional long read deep sequencing data, using Oxford Nanopore Technology (MinION). In addition, we have shown that human DNA can be recovered from human footprints in beach sand, while eDNA extracted from sand at restricted human access locations contains no human eDNA.

We have also further clarified the rigorous nature of our original sampling (first version of the manuscript). Although it was collected primarily for the purposes of our sea turtle research, it was also always intended to be used for human eDNA analysis (and sea turtle viral and metagenomic analysis). In hindsight, our efforts to provide a concise paper meant that we did not adequately and explicitly describe how investigator DNA contamination was intentionally avoided throughout this research. We apologise for this earlier brevity, and now explicitly describe our rationale and approach in the Methods section.

Major comments:

- This study is purely based on samples/data from past research, which I in general fully support to make research as efficient as possible. I, however, see a few problems in the study design and sample size for answering the posed question.

Based on the Reviewers' comments, this is no longer the case, and the study now includes a larger sample size and additional studies designed specifically to address human eDNA questions. We have also retained the original data and explained more clearly that, in the collection of this original data, while study design was multipurpose and sea turtle-centric, that we were cognizant of and planned for human eDNA investigation throughout the project as an additional objective. While we always intended to quantify the number of human aligning reads in our original samples, we did not stack the original samples in favour of human eDNA detection (i.e. we did not seek samples from densely populated areas). This research multi-use efficiency was part of our initial study design (to maximise the returns of the expensive shotgun sequencing). In our initial batch of sampling and sequencing (2017) we had expected that the rehabilitation tank water would contain more human eDNA than the wild environmental samples. We took all appropriate steps to avoid any investigator contamination in order to be able to assess this difference. However, the converse was the case, the tank water had lower human eDNA levels than the wild samples. For subsequent rounds of sampling, we were fully aware of the likelihood of relatively high human eDNA capture and continued to plan sampling protocols accordingly. We agree with the reviewer that we did not make these points sufficiently explicitly clear in the initially submitted version and have now revised the methods section accordingly.

As part of the revision, and in line with Reviewer suggestions we have now conducted additional experiments and quantified the level of human eDNA from densely and less densely populated areas, and shown that human eDNA load correlates to proximity to population centres (Figs 2-4, Supplemental Figs 2-6).

- The negative controls (especially from the sand samples) have been taken as negative controls for the sea turtle, which means that this sample expectedly contains human DNA and contamination during sampling can not be ruled out (it is actually interesting that the sand negative control shows the least amount of human DNA, both in terms of the full human genome and the Y chromosome, which is something that would not have been expected, see Supplementary Table 1 - do you have an explanation for that?).

This is partially the case. The water negative control (MilliQ water transported to and from sampling sites) functions equally well for human and sea turtle eDNA control (and any other species) and was designed as such, in accordance with standard practice for water-based eDNA sampling.

Yes, we have an explanation for the sand negative control, and it is relevant to the interpretation. In hindsight we should have included these details in the manuscript, and apologize for omitting them. The negative field control sand sample was taken from higher up the beach than the other samples. Sand dunes are protected in Florida, and to prevent dune erosion and habitat destruction there are signs posted stating that it is an offence to walk on the dunes. Boardwalks above the dunes are provided for beach access. Therefore, there is less human activity at the negative field control sample site, and sea turtles and humans use the same lower areas of the beach. Negative field control samples were collected within the permissible zone, but very close to the non-permitted zone. Although not necessarily completely free of all human contact or windborne impacts, the sea turtle negative control also equates to a human negative control. We have now included this information in the manuscript.

We have now also obtained human ethical approval and conducted intentional human eDNA sampling. This includes sampling human footprints in beach sand from consenting study participants. For this study we have two additional controls. 1. Our usual negative controls, taken on the same day from a site with no obvious recent human activity (higher up the beach, as with the turtle negative controls discussed above). These were mostly (but not completely) free of human eDNA (Fig. 3b). In addition, we had more elaborate sand negative field controls. For these, we obtained US National Park Service permits to access a restricted access island. All sand samples taken from the island with restricted human access were completely free from human eDNA, as assessed by human species-specific qPCR. Nanopore sequencing the human footprint eDNA sample returned 7,077 human aligning reads, while the restricted access island sample returned only 11 human eDNA aligning reads.

For the water samples, the negative control consisted of MilliQ water that was brought to the site and then filtered in situ, making sure that no human contamination had happened from the point of filtering to the end result. I am, however, wondering if contamination could have happened during sampling, e.g. by being in touch with the water while sampling (which would have not been a concern for the original turtle study this experiment had been designed for)?

No, this is not the case. Although sea turtles were our priority, we were aware during all sampling that we would also have a human-focused eDNA project. We were careful to avoid investigator contact, with PPE worn throughout sampling and processing, and no human (investigator) contact was made with the water at any time. We intentionally avoided human contamination as we did not want to bias our sample composition (metagenomics, or waste precious sequencing reads sequencing investigator DNA rather than 'wild' sample eDNA). We also knew that we wanted to investigate the presence of human reads, and planned accordingly. We have now made this point clearer in the methods section.

We originally believed that the HiSeq samples would be focused on HGB in captive settings (hospital staff, not investigators). However, upon analysis of that initial dataset in 2017, we realized that the wild samples had higher levels of HGB. Therefore, for all subsequent samples (NovaSeq) we continued to avoid any potential investigator contamination during sampling and processing.

The new qPCR-based studies also demonstrate that the human eDNA being detected in wild Floridian and Irish samples is not due to investigator contamination during sampling, with no detection in negative field controls (Figs 2,3). Additionally, they show that the quantity of human eDNA present increases in proximity to human towns.

This could also explain why the samples from wild water show more human DNA content than the rehabilitation tank samples, since I would expect the latter to contain more human DNA due to humans working with and around the turtles, but maybe they can be sampled more easily without directly getting in touch with the water?

There is no investigator (eDNA sampler) contact with the water in either the tank or the wild samples. PPE is worn in all cases and water contact avoided. Yes, we also originally expected the tank water to contain more human eDNA than the wild water samples, as the rehabilitation staff do come in contact with the tank water (see below). Surprisingly, the results of this original HiSeq sample set showed more human eDNA was present in environmental samples than tank samples. Given that finding, we continued to avoid investigator contact with all subsequent sampling. The higher abundance of human eDNA in the wild samples compared with the tank samples convinced us it was worth pursuing this novel finding further. We had obtained our initial human eDNA results in 2017 (initial sequencing sample cohort analysis). For cost and effort efficiency (and as our main funding is sea turtle focused) we relied on samples also useful for our sea turtle eDNA research, but we were conscious that we would also be investigating human eDNA throughout the project so continued to carefully avoid eDNA sampler contact with the water/sand for all subsequent samples.

Apologies if we caused confusion as originally written. Our aim was to highlight that these are genuine wildlife eDNA samples and they were not biased towards heavily human populated areas. We did not mean to imply that we were oblivious of the potential of human contamination from the investigators. We were careful to avoid such contamination throughout the project. We have now revised the text to more clearly make this point.

The hospitalized sea turtles are high intensity cancer patients, so they are removed by handling at least weekly for check-up, and often more frequently depending on condition and stage of rehab (e.g. laser surgery, post-surgical wound care etc.). They also sometimes require hand feeding. Given these circumstances we expected more human DNA in the tank samples. The fact that environmental samples have more human eDNA highlights that the level of human DNA making its way into our waterways and oceans far exceeds the level entering rehab tank water during turtle care and husbandry.

- It is also striking that the human DNA content in wild vs. tank water samples could be explained by the employed sequencing machine, i.e. NovaSeq vs HiSeq. While I am not sure if the modest increase in mismatch rates when performing alignments with NovaSeq in comparison to HiSeq, and the general systematic bias that is expected when changing protocols, could result in such a substantial difference, I would like to ask the authors to look into this.

This was unexpected for us also; we did not anticipate such variation when switching to NovaSeq. Interestingly, the magnitude of the change from HiSeq to NovaSeq was greater than when the same sand sample was sequenced with either a conventional library prep (PCR) or PCR free library prep kit (7SM36HN and 12SM36H respectively), indicating that the cause may be due to factors other than sequencing protocol. Temporal variation or seasonality could also be at play, as the HiSeq water

samples were collected in 2017, while the NovaSeq water samples were collected in 2020 and temporal factors such as heavy rain are known to dilute salinity (and presumably eDNA) in the same geographic areas which were sampled (inlet) in both years. The location is also tidal. One immediate point of difference is that the concentration of retrieved eDNA (total eDNA extracted) varied between the 2017 and 2020 samples. 2017 range: 137.4 - 217.3 ng/μl. 2020 range: 16.1 - 62.9 ng/μl (excluding neg. field control which was 0.0 ng/μl).

We have taken tide into account for the new intentional human sampling sites taking samples on a low tide with the exception of the "Beyond Inlet" samples which were intentionally taken on an incoming tide to act as low a human sampling site as possible.

- Coming back to the negative controls, the authors do not report any RPTM in Figure 1 for the water control; when looking at Supplementary Table 1, I understand that this is the case because less than TM (ten million) reads have been sequenced for this sample. The authors correctly state in the table legend that "The lack of sequenced reads reflects an overall lack of DNA in this control sample." However, 196,876 reads have been obtained for this negative control, out of which 3,184 align to the human genome - which suggests strong human contamination. While no RPTM can be calculated, this still seems to be an important result, which I would like the manuscript to discuss.

We have now discussed this further in the manuscript, as requested (please see the results section). It is true that 196,876 is a significant level of total reads. However, it must be noted that this sample was given equal pooling ratios to the other NovaSeq water samples which have read numbers in the 152 million - 199 million range. In this context, 196 thousand reads is rather low and reflects a lack of input DNA in that sample.

It is known that Illumina samples can contain small contaminations from previous runs, which is why we have this negative control sample. Subtracting the number of human-aligning reads in this sample (potentially caused by sequencing contamination) from the genuine samples provides an indication of how many human reads are genuinely coming from those environmental samples (from the same NovaSeq batch) and not from sequencing contamination. The range of environmentally-originating human eDNA reads is 42,128 to 122,131 reads, or 13 to 38 times more environmentally-derived human eDNA. When quantified immediately after eDNA extraction this field negative control sample had 0.0 ng/ul DNA (NanoDrop) and after library prep had 0.000 ng/ul DNA (Qubit), suggesting that those few reads detected likely arose within the sequencing machine itself, potentially coming from previous runs.

Our new qPCR results show that no human contamination occurs from sampling, extraction or qPCR. While no contamination occurred in our qPCR experiments, and a limited level of contamination may have occurred due to the use of a high-throughput core facility machine likely also used for human samples by other customers (max. contamination approx. 1/1,0000 of the total reads generated for the NovaSeq samples), we still wished to generate NGS data which minimises any late-stage contamination. Therefore, for our new sampling we conducted Oxford Nanopore sequencing in our own lab using a MinION. This MinION had not previously been used for any human-related application (only invertebrate and sea turtle). Additionally, we ran the negative control samples on fresh flow cells. These new experiments produced far fewer human aligning reads in the negative control samples (range: 2 to 26 reads), while environmental samples still returned a high level of human aligning reads (7,077 for footprint sand and 330,775) (Fig. 4).

This new experiment confirms the qPCR findings that human DNA is specifically present in environmental samples, and shows that recovered human eDNA is of sequenceable quality.

[On a minor note, how are the % alignments calculated since, e.g., 3,184/196,876 is not 1.59% but 1.62%? Is this due to partial alignments or slightly variable read lengths?]

This is from the difference between total returned sequenced reads (reported in the Table) and total post-QC and post-trimming read number being used to generate the percentage.

- I think it's an important idea to write a landmark paper about sampling human DNA in eDNA studies. I however think that the sample size of this study is too small to create such a landmark study; while I don't want to criticize small sample size in general, I feel that in this case technical, spatial and temporal replicates could have been taken to really study the content of human DNA in eDNA data. Especially understanding the spatial axis would be important since the "wild" samples taken by the team are reportedly relatively far away from any human civilization. Would the ratio of human DNA increase when approaching civilization?

The reviewer has posed some very interesting questions, and we thank you for your suggestions. We originally tried to highlight that human eDNA is recoverable even if it is not the intended target and when sampling was not occurring in high human population areas. In our original effort to avoid biasing our sampling towards human eDNA we neglected the possibility of also obtaining samples from high human population sites. However, we agree with the reviewer that this is an interesting question, and have now also addressed this. In addition to showing human eDNA can be recovered as bycatch we have now also investigated intentional human eDNA sampling, including in relation to spatial and temporal dimensions.

This includes eDNA sampling at and away from sites of human habitation in both temperate and sub-tropical zones, and the sampling of human sand footprints and sand from areas with restricted human access.

This new sampling dramatically increases the sample size of this study, across much larger spatial dimensions, with more replicates. We have also established two species-specific qPCR assays as useful human eDNA assays. This opens the possibility of cost-effective human eDNA investigations with a wide range of potential applications. Just like wildlife eDNA, these human qPCR assays and sequencing approaches could be applied to a range of substrates (from air, solid substrates, water and ancient sediments and ice cores). We believe this study now establishes the feasibility of a whole field of human eDNA applications.

Re: Would the ratio of human DNA increase when approaching civilization?

Yes. Our new qPCR-based studies confirm this, showing higher abundance of human eDNA as one approaches civilization.

- Have the authors conducted a metagenomic mapping approach against a database of expected mammals? I am wondering if some reads would map to other mammals in such an approach - something that could explain the difference in human read assignment when only using the more human-specific Y chromosome.

This is an interesting suggestion. However, our new data (species-specific qPCR and long read sequencing) largely negates this question, and confirms the specificity of the recovered eDNA as being human, our primary focus.

We agree with the reviewer that this is potentially a source of the observed variation, although so is losing any human female reads in the Y analysis (i.e. potentially sub-sampling the total (male and female) human read number by approximately half). Additionally, the Y chromosome represents only a small portion of the entire human genome, with such additional subsampling expected to alter

the proportions between samples. However, we believe that the Y chromosome analysis provides added specificity and complements the whole human genome analysis (original samples). We have not found a well curated database allowing easy alignment of our data against all mammals, hence our focus on the rapidly evolving Y chromosome. We have also conducted microbial analysis on the new nanopore sequencing data (WIMP analysis), but similar bioinformatic tools for mammals seem to be lacking currently with most pan-vertebrate/mammalian tools still focussed on single gene (or a small handful of gene) metabarcoding approaches. We are still interested in conducting such (not microbial restricted) metagenomic analyses on our shotgun samples but believe it is beyond the scope of this present human-focused study.

In order to confirm the presence of human eDNA with the new data we have focussed on long reads (nanopore) and species-specific qPCRs.

- Have the authors looked into coverage of the human genome in their shotgun data? It would be interesting to know if most of it is, e.g., mitochondrial, and to give an idea of what sort of information could be retrieved using this data. Especially given that short-read sequencing is being used in this approach and one would expect to sequence a pool of different human individuals, it might be difficult to actually make any individual-specific inferences. I, however, see that this is not the focus of this study, but rather showing that human DNA can be discovered at all.

We agree both that this is an interesting question, and that this is largely beyond the scope of the current study. However, we have included an indicative graph (Fig. 3b) showing the coverage of chromosome 2 (which had the largest proportion of aligning reads, likely as it is the second largest chromosome) for the high human water sample. There was broad coverage across almost the entire chromosome, suggesting that there is no major bias in terms of which regions of the human genome are being recovered by shotgun sequencing from eDNA samples. We have also included graphs showing the coverage of the X chromosome, Y chromosome and mitochondrial genome (Supplemental Fig. 7a), as these are important genomic regions often used in ethnicity and parental origin studies.

We have also conducted structural variant analysis on the nanopore data, providing proof-of-principle evidence that structural variants can be recovered from human eDNA samples, even without prior enrichment (Fig. 3c).

Minor comments:

- I would suggest rewriting the introduction a bit to make clear that it is the shift from metabarcoding-based approaches to shotgun sequencing that potentially enables retrieving enough genetic data to identify human individuals/make phenotypic predictions. At the moment, those points read a bit unconnected.

In addition to the original introductory metabarcoding versus shotgun sequencing text in paragraph 1, we have now added the sentence below to paragraph 2 to more clearly connect these points, as suggested.

“Current targeted qPCR and metabarcoding-based eDNA approaches do not recover any substantial human genomic information. However, as eDNA shifts towards shotgun sequencing, potentially large volumes of human eDNA will be retrieved, including potentially sufficient DNA volumes to identify human individuals or make phenotypic predictions.”

- II. 157: The authors suggest correlating human susceptibility loci with pathogen load in

environmental samples; while this might be a prospective at some point in the future, I think it's necessary to discuss that this is a very difficult endeavor given that each sample is a pooled sample with intrinsically high variance in both phenotypic as well as genotypic measurements, plus the environment will have a major impact on any detected correlation, given that pooled samples will per definition come from different locations.

The reviewer is correct, this is currently a difficult endeavour, especially for pooled samples. However, the rate of advancement of eDNA (both wildlife and human wastewater) has been staggering. While certainly a future endeavour, it is likely achievable in the medium term, especially if one was to consider sampling from specific sources, e.g. air eDNA (from specific locations, home rooms, hospital rooms, or even specially designed diagnostic rooms), or water eDNA (e.g. wastewater before it leaves the home, or even after a specific toilet flush). More long-term refinements may enable similar applications from pooled samples. We have added text expanding on this in the discussion. As the paper's results now covers intentional human eDNA capture also, we have also included text on both pooled and individualised pathogen and host loci sampling.

- ll. 185: I understand the author's intentions of mentioning these examples, but I would strictly adhere to the problem of genetic data sharing in this article (since it's a big enough problem on its own). We felt that these points are indicative of how big data have been exploited despite previous regulatory assurances. We feel that the medical example is particularly pertinent, given that human medical data falls under the remit of the same agencies as human genomic data, as genomic approaches have progressed primarily in the medical field. We also selected these two examples as these companies originate in countries with relatively stringent regulation, thus highlighting the temptation to mis-use large datasets even by reputable companies. For these reasons we would like to retain these examples (especially the medical records one), but if the Reviewer insists then we are willing to remove them.

- Please provide any supplementary data as tsv/excel files so that they can easily be re-analyzed by the reader. Please also include any raw data for the Y chromosome coverage.

All supplemental tables are now in Excel format as requested. We now show X chromosome, Y chromosome and mitochondrial genome coverage for the high human water and sand footprint eDNA samples (Supplemental Figure 7a), as these three regions of the genome contain important information about ethnicity and parental origin.

Thank you again for your work and I am sorry that I can't be more positive at this stage. I find it very encouraging that you are looking into this very important research subject, but would either recommend tackling a few of the mentioned concerns (I am happy to discuss details or if I misunderstood something) or "toning the message of the paper down" from a landmark paper to a pilot study that for the first time shows the vast amount of human data that can be found in eDNA shotgun data.

I wish you all the best with your exciting research,
Lara Urban

We appreciate your positive comments, and apologize for the lack of clarity regarding the negative controls and our intentional procedures to avoid investigator sampling contamination. We hope that in light of the new experiments and the additional information you agree that the manuscript has been greatly improved.

Our original aim was to i) show proof-of-principle that human eDNA can be recovered, and then ii) discuss the broader potentially serious implications. Apologies if we did not segregate the pilot proof-of-principle and broader implications sufficiently. With the addition of the new data we feel that the broader implications are no longer theoretical, and all of the technological capabilities are now in place to enable both potentially beneficial and worrying human eDNA applications.

Reviewer #2 (Remarks to the Author):

This piece addresses a timely and important topic, given increasing capabilities and interest in sampling and analyzing wastewater for public health or other purposes and the often-similar ethical issues that analysis may raise. My comments here are restricted to the ethical discussion in your article.

Thank you for your considerate review and positive comments. It is highly valuable to also have input from an ethics expert. We have now implemented all of your comments.

First, I encourage you to make the connection between human genetic bycatch (HGB) and wastewater monitoring arising from the Covid-19 pandemic earlier in the article. You allude to this use at the outset of your article, but only mention it explicitly (and briefly) on page 4 of your manuscript. As part of motivating the reason to consider these ethical issues now, the incredible investments that have been made to develop wastewater monitoring during the pandemic—and the ways in which human genetic material is collected and may be exploited—are surely relevant. As suggested, we have explicitly included references to wastewater monitoring and HGB earlier in the manuscript, now including it in the first two paragraphs of the introduction, and including the recent use of this technology for outbreaks of pathogens beyond SARS-CoV-2.

Second, I want to suggest a response to one of the ethical issues you raise. You appear concerned about “with whom should the responsibility” lie to police the adequate filtering and scrubbing of eDNA sequencing to shield human privacy in public data repositories. But it seems to me that norms about policing protections for human privacy ought already to be well established in other fields. Who is responsible for policing the adequate filtering, scrubbing, or de-identifying of human genetic material in instances in which human subjects are unquestionably involved? It would seem sensible to locate responsibility for that work with the same entities or actors when eDNA is at issue, too. But perhaps this is too facile a response. If so, it might be helpful to explain why. It is a valid question. A larger issue is that animal studies and human studies have little to no overlap in terms of oversight and regulatory legislation and oversight bodies (different ethical bodies, different required training, different governmental regulators etc.). For human studies, responsibility for anonymisation of data and adherence to regulations sits with the investigators, though enforcement and oversight sits with institutional and governmental bodies.

Currently in most countries research oversight is completely separated into animal and human spheres with no overlap. Therefore, the wider question is whether eDNA researchers employing shotgun sequencing but not focussed on human sequences need to be brought into the human system and obtain dual approvals (animal and human) from both systems and comply with the responsibilities of both systems simultaneously. It is currently burdensome (time and financially) for environmental researchers to even establish whether their sequencing data contains human reads. This is a requisite to knowing whether data needs to be scrubbed. Additionally, with current human approvals researchers need to obtain approval prior to commencing work, so this raises the issue of

whether environmentally-focused researchers would need to obtain human approvals for every eDNA project because there ‘might’ be human eDNA present.

In order to obtain ethical approval for human studies, specific training is required before being able to submit an application. Environmental researchers are not subject to the same system (having an independent system), and one of the benefits of non-invasive eDNA techniques is reduced permitting requirements (e.g. wildlife, ethical and endangered species permitting). The constraints of more arduous human permitting requirements could hamper the non-human eDNA field in terms of adoption and progress.

In the future, submission pathways could be integrated automatically into the eDNA submission processes of data repositories to filter out human reads. However, setting the threshold of those filters could become contentious, needing to effectively remove human data but not so stringently that they may remove some animal reads also. It is not necessarily straightforward to ascertain for every read whether it is human or animal in origin (particularly from short read data, currently the most widely utilized). This is especially true as not even the full diversity of human genomes is known yet, let alone that of every other species. The field of genomics is still developing for wildlife, barely progressing to pan-genomics, which is required to characterise genomic diversity within a species. In summary, while filtering and responsibility for filtering initially seems straightforward, regulatory, responsibility and technical implications of this issue alone could have serious implications for the practicality of eDNA studies and the continued adoption and implementation of eDNA approaches.

Third, I recommend that, in addition to identifying the ethical issues that arise because eDNA captures human DNA too, you explicitly call for researchers, funders, and other stakeholders to develop responses to these ethical issues before the technology at issue becomes even more widespread and entrenched and before human genetic bycatch comes to be exploited in the ways you suggest. Ex ante planning is crucial for ensuring that law and ethics stay ahead of technology, and you should be explicit about that.

Thank you for your suggestion. You are correct that due to the importance of these issues we needed to make this call explicitly, rather than just implying it. We now end the Discussion section with an explicit call.

Fourth, it may be worthwhile to identify circumstances under which HGB is more or less likely to give rise to ethical concerns. Wastewater monitoring may raise serious privacy concerns about human genetic privacy because our most frequently visited places don’t change very much and our home addresses are readily knowable. That means that human genetic data retrieved from a manhole may be traceable to a small group of possible human contributors with relative ease. That is less likely to be the case where seawater is collected from a popular tourist beach. You identify some instances in which HGB raises serious ethical concerns, as in the exploitation of HGB to identify genetic data of populations less willing to participate knowingly and voluntarily in genetic research. It may be helpful to try to demarcate the circumstances under which ethical issues are more or less likely to arise (small, stable populations of humans contributing to an eDNA watershed would likely be one).

We have now included a section in the Discussion detailing how HGB consideration may be more applicable in different settings, as suggested. Furthermore, as we have now demonstrated the feasibility and relative ease of intentional human eDNA sampling, we have strengthened the discussion in relation to intentional beneficial applications and misuse of this technology.

Fifth, turning to Table 1:

- I recommend that “Unintentional consequences” would be better as “Unintended consequences.”

Changed as suggested.

- It is not entirely apparent how “genomic harvesting” differs from “genetic surveillance – illegal/unethical collection of whole ethnic groups/populations.” Both describe genomic harvesting. Is the distinction you seek that the first identifies the harm of lack of consent, while the second identifies a more specific harm of exploitation that is experienced by minoritized groups? If so, I am not sure the latter is genuinely a harm of “surveillance” as opposed to a harm of “harvesting.” And if so, please work to clarify the distinctions between these two identified harms. I think both are present, and both merit identification.

Thank you for pointing this out. We have now clarified genomic harvest (being interested in genomic data) as being distinct from population genetic surveillance (being able to monitor populations, and the presence of specific ethnic groups within a larger population). While both applications are worrisome, the combination of air or wastewater monitoring for human ethnic DNA signatures is particularly concerning considering the frequency of ethnic persecution and genocide throughout human history.

- The last harm identified, about “bio-piracy,” includes “flora/fauna genetic data.” How is this a function of HGB? Please clarify.

You are correct, flora/fauna bio-piracy is not a function of HGB, but the issues surrounding flora/fauna bio-piracy are well established and are instructive for HGB. With shotgun sequencing one could simultaneously conduct flora/fauna/human bio-piracy. We have modified this point to clarify.

Decision Letter, second revision:

6th October 2022

Dear Dave,

Thanks for bearing with us during the second round review process for "Human environmental DNA: inadvertent human genomic bycatch and overt capture raises novel applications and ethical concerns". As you know, while one of the original reviewers was able to re-review, reviewer 1 was not. This meant that we needed to find another reviewer with similar expertise (labelled 'reviewer 3') to both check the comments from the first round and continue the peer review process. The good news is that they feel that the contamination/study design issues have been satisfactorily dealt with. Reviewer 2 also feels that the issues they raised in their previous review have been substantively addressed and has only minor concerns remaining. Reviewer 3, however, while they were satisfied with the responses to the first round, has their own concerns to raise. I realise it's frustrating to receive substantive comments in the second round of review, especially when these are extensive, but we feel that reviewer 3 makes some very constructive suggestions to strengthen the manuscript that should be doable. These fall into two main areas. The first is better contextualisation among literature from fields that have experienced human genetic bycatch or similar issues (this also ties in to reviewer 2's remaining concerns about risk/monitoring). The second is the need to drill into your data to discuss likelihoods and feasibility of human bycatch--for publication I don't think we necessarily need you to demonstrate that bycatch is inevitable or even likely, but given that this is a research manuscript (rather than an opinion piece) it's important to maximise the data presented and take the opportunity to discuss the problem scientifically, rather than speculatively. Editorially we feel that integrating reviewer 3's suggestions should result in a substantially strengthened manuscript--partly for this reason, and due to practicality, we recommend you revise as an Article rather than Brief Communication. That is, a maximum main text word length of 3,500 words, up to six display items, and unlimited references and methods.

If you wish to submit a substantially revised manuscript, please bear in mind that we will be reluctant to approach the reviewers again in the absence of major revisions.

* Include a "Response to reviewers" document detailing, point-by-point, how you addressed each referee comment. If no action was taken to address a point, you must provide a compelling argument.

18This response will be sent back to the referees along with the revised manuscript.

* If you have not done so already we suggest that you begin to revise your manuscript so that it conforms to our Brief Communication format instructions at <http://www.nature.com/natecolevol/info/final-submission>. Refer also to any guidelines provided in this letter.

[REDACTED]

If you wish to submit a suitably revised manuscript we would hope to receive it within 6 months. If you cannot send it within this time, please let us know. We will be happy to consider your revision so long as nothing similar has been accepted for publication at Nature Ecology & Evolution or published elsewhere.

Nature Ecology & Evolution is committed to improving transparency in authorship. As part of our efforts in this direction, we are now requesting that all authors identified as 'corresponding author' on published papers create and link their Open Researcher and Contributor Identifier (ORCID) with their account on the Manuscript Tracking System (MTS), prior to acceptance. This applies to primary research papers only. ORCID helps the scientific community achieve unambiguous attribution of all scholarly contributions. You can create and link your ORCID from the home page of the MTS by clicking on 'Modify my Springer Nature account'. For more information please visit www.springernature.com/orcid.

Thank you for the opportunity to review your work.

[REDACTED]

Reviewer expertise:

Reviewer #2: as before

Reviewer #3: eDNA, conservation genomics

Reviewers' comments:

Reviewer #2 (Remarks to the Author):

Thank you for the opportunity to review this paper again, and for your patience as I completed this review. As before, my comments are restricted to the ethical discussion in the article.

I think this paper is substantially improved. I appreciate the changes that the authors have made in response to reviewer comments. I have some additional comments, but they likely require minor, rather than major, revision.

First, on page 2, ll. 51-52, you note that human wastewater monitoring has already been repurposed to track other pathogens. This is true. But this recitation misses a forest for a single, and arguably the least objectionable, tree. Wastewater monitoring is also already being put to use to detect illicit drugs, antidepressants, stress markers, and alcohol consumption, among other things. There may be a risk that wastewater monitoring could be used to surveil for pregnancy-related hormones or abortion-related drugs, particularly in states that have now made abortion care unlawful. So in addition to noting the disease surveillance to which wastewater monitoring may be put, it may be worthwhile to flag broader uses already contemplated or in use. In addition to the sources cited here, you may wish to consider: Ram, N., Gable, L., & Ram, J.L. The Future of Wastewater Monitoring for the Public Health, Univ. Rich. L. Rev. 56(2022).

Second, in the discussion, I encourage you to attempt to be even clearer about scenarios where HGB may be neither beneficial nor exploitative. Surely there are some such uses (I previously suggested that seawater collected from a popular tourist beach is likely to be less ethically concerning than wastewater monitoring of a small, defined, stable population). Identifying these kinds of scenarios will help make clear that you are able to parse from the range of possible uses the genuinely problematic ones.

Third, at ll. 266-267, please reconsider whether to place forensic and criminal investigative applications under "benefits" as opposed to "worrying" applications. Solving crime is a good thing, to be sure. But exploiting involuntarily shed genetic information for investigative aims risks putting all of us under perpetual genetic surveillance in ways that may raise genuine questions about the appropriate limits of policing and surveillance.

The changes to Table 1 are much appreciated. This Table is now clear and easy to understand.

Reviewer #3 (Remarks to the Author):

20The basic premise of this paper is that 1) human eDNA may be commonly captured and sequenced either inadvertently (“by-catch”) or intentionally (“overt”) and 2) this potentially raises ethical and legal concerns. I think this premise is interesting and relevant to the field. Thank you to the authors for attempting to tackle this topic. I also want to acknowledge that the authors clearly took previous reviews seriously and made meaningful attempts to address the concerns raised there about contamination. I don’t have any comments specifically about their response to previous reviews. Thank you.

I do believe there are two important weaknesses to the paper. First, to be blunt, the scholarship in this paper is poor. The topics here are highly interdisciplinary, so I appreciate and am sympathetic to the fact that this is hard to do. Second, I believe that the authors at times have made claims about what is feasible that extend well beyond what is strictly demonstrated from their empirical data. I will address these two topics in comments one at a time.

Primary concern #1: Engagement with existing literature

As noted by the authors, there are two different situations for human eDNA data: By-catch and intentional sampling of humans. Like the authors, I’ll walk through each of these separately.

Intentional human sampling: One might intentionally capture and sequence human eDNA (either with enrichment or simply shotgun sequencing). In this case, you are conducting research on humans and your research needs to comply with any special ethical and legal considerations for research on human subjects. This would be “overt capture” (paper title) and relates to the Discussion in this manuscript on lines 228 – 267. In this discussion the authors cover a lot of potential ground for overt study of human eDNA. However, I feel like the authors fail to engage fully with recent literature on human medicine, forensic, and anthropology fields that they suggest innovations for.

Human medicine

Line 243: Along with Reviewer #1, I’m skeptical of feasibility (reviewer comments on previous line #157). I think the authors in their revision are trying to suggest that, although pooled samples are difficult to use, you could get eDNA from a single individual at a time by sampling a specific hospital room or toilet flush (Line 248). In this case, you are attempting to sample a specific, knowable individual. From an ethics standpoint, I think you clearly should gain consent from that individual (like the authors of this paper did for their footprint work). In which case, why not just request a cheek swab (a superior sample type)? I can only imagine eDNA sampling being useful when you cannot gain consent because 1) the application is forensic, for a legal investigation or 2) the sample represents a pool of individuals. The authors in their response argue that this application “is likely achievable in the medium term”. Maybe, but the authors only cite a perspective paper (citation #1) and a paper on COVID-19 monitoring (citation #40; please note that there is also now a peer-reviewed version of this ms). My request of the authors: If there is evidence that “one could utilize wastewater eDNA-based sampling to correlate level of pathogens with abundance of susceptibility loci in a given population”, please cite at least one peer-reviewed, empirical demonstration.

Forensics

The authors correctly identify eDNA sampling as potentially useful for forensic applications, but do not engage in any way with forensic genetics literature (only cited literature regarding forensics is the

21speculative discussion from a paper on naked mole rats). There is no engagement with literature on forensic genetics/eDNA (e.g., recent papers with clear connections like “Assessing the use of environmental DNA (eDNA) as a tool in the detection of human DNA in water”; *Journal of Forensic Sciences*). There is also no engagement with forensic genetic ethics, such as phenotyping which is a really important issue connected to racial issues that might arise if one were to “identify the geographic human populations” as the authors here suggest (dozens of papers exist on this topic). My request of the authors: Please deeply and meaningfully engage with existing forensic methods/ethics in the literature, or at least point readers to the most relevant discussions of this topic.

Anthropology

The authors suggest that eDNA sampling might “identify undiscovered sacrificial sites”. There is a wealth of literature on ethical considerations when studying the genetic material of ancestral human populations. There is also a wealth of literature specifically on human ancient DNA, including soil samples (i.e., ancient eDNA). None of this literature is mentioned. My request to the authors: Please deeply and meaningfully engage with existing anthropology/ancient DNA methods and ethics in the literature, or at least point readers to the most relevant discussions of this topic.

Human “by-catch”: One might also retrieve reads from humans unintentionally. The original researcher would clearly be violating basic research ethics if they “repurposed” these reads for research on human subjects. They may also face issues with posting this data publicly where other data users might “repurpose” these reads without the original research team’s consent. Thank you to the authors for articulating this. However, the authors do not acknowledge that human “by-catch” in ecological studies is not new or unique to eDNA sampling. Inadvertent (or even intentional) capture of humans has been described/discussed relative to wildlife camera trapping (e.g., DOI: 10.1002/2688-8319.12033) and eco-acoustic data (e.g., DOI: 10.1101/2022.02.08.479660). My request to the authors: Seek to make connections with similar “by-catch” issues using other methodologies and note if there are best practices which are directly applicable to eDNA sampling.

Although not my area of expertise, the discussion of potential nefarious application of human eDNA data seems better. Links to recent literature on genetic research in indigenous communities are relevant. The idea of using genetic material to screen for the presence of specific populations is indeed “chilling” (Line 306), but the authors seem to imply that there are nearly-fixed genetic differences that are diagnostic for ethnicity. I’m guessing this is not what the authors intend to communicate. A more nuanced message here might explain more explicitly that even if there is not a good scientific basis for using eDNA to infer ethnicity (or some other human attribute), we have historically seen pseudoscientific approaches used as a weapon (e.g., so-called “craniology” in the eugenics movement, Nazi racial “science”, etc). A possible, partial analogue with gender and disability ethical discussions would be non-invasive prenatal testing (i.e., genetic evidence of embryo sex or disability being used to make a pregnancy termination decision). I do note, again, that this is not my own area of expertise.

Bioinformatics: None of the bioinformatic tools except for Galaxy appear to be cited. This includes FastQC, Trim GaloreI, Porechip, Bowtie2, StringTie, Samtools, minimap2, and mosdepth. Please cite the associated peer-reviewed papers for each. Also watch for your dependencies (e.g., I believe WIMP uses Centrifuge). Further, no parameter values for any programs are reported. I would minimally have

22expected to see parameter values listed (even if they are the current “default”) and summaries of intermediate and final bioinformatic results. Better would be rationales for parameter values selected and links to the supplement where multiple parameter values were assessed.

Primary concern #2: What can we really do with human eDNA?

Reviewer #1 made the observation “it might be difficult to actually make any individual-specific inferences” and I think this is a critical point. I do not agree that this point is “beyond the scope of the current study”. If it is not simply difficult but practically impossible to make individual-specific inferences from human by-catch eDNA, then many of the ethical/policy concerns for researchers are ameliorated. This is important: Should we as a scientific community require human subject research approval for most amplification-free eDNA studies? Failing to protect the rights of potential human subjects would be unfortunate, but so would creating regulatory barriers to research that are unnecessary if privacy risks are low.

Short-read data (Illumina): The authors report the highest observed proportion human-aligning reads for sample Inlet1 (Fort) with 42,030 reads per 10 million total reads. That is, 0.420% of sequenced bases were potentially human (probably less; as alluded to by the authors, much of what you can align is not necessarily human, particularly if repetitive regions were not masked). So 1X genome coverage for a single human genome (3.2 Gb) would require around 750 Gb of sequencing effort. That’s ¼ of a NovaSeq 6000 run or an entire HiSeq 3000 run of sequencing effort for a single sample. Do I have my back-of-the-envelope math right? This is a lot of sequencing effort. Is it possible that, as long as sequencing efforts are not incredibly deep, inadvertently generating data that allows human individual identification is a non-issue? The authors cite two empirical papers as examples of “deep” sequencing of environmental samples. One is the Farrel et al study using the same samples as in this study. In Table S1, the highest sequencing effort sample has 304 million reads. If these are 150 bp reads, that’s 45 Gb sequenced (i.e., about 4% of the sequencing effort you would minimally need to get 1X coverage for a single human genome based on this study). The other empirical paper (not really eDNA, but bulk arthropods) had a shotgun sequencing effort around 2 billion per sample (~300 Gb). Of course, to tease apart population or individual-level variation, you probably need much more than 1X mean coverage across an unknown individual pool size.

Long-read data: Figure 4 seems to imply that coverage with the Nanopore sequencing was extremely high for an environmental sample (4b; mean coverage on Chromosome 2 appears to be around 20X), but coverage for the Y Chromosome is extremely low. First, I’m surprised that in the supplement the Moultrie Creek B sample reports ~10% of reads being putatively human (20X more than the highest RPTM environmental sample on short reads). I suppose this speaks to how high of a human use area this is. However, how many contributing individuals do we think there are? If this 10% of DNA comes from potentially hundreds of individuals, can meaningful inference be made? Can the authors either 1) dig into the literature and show some evidence of feasibility or 2) build on their quick proof-of-concept “structural variant” list to show how one would derive some population inference from this information?

In summary, if this is not intended simply to be a “perspective” piece, I would like to see the authors spend more time engaging in empirical tests of what is feasible from the current dataset.

Minor comments

If the authors seek a revision, please spend some time ensuring that figures are readable at print scale and that the figure legends provide enough information to understand new figure types and terms (this is particularly relevant for Figure 4).

Regards, Taylor Wilcox

Author Rebuttal, second revision:

We would like to thank the Reviewers for their careful consideration of the manuscript and for their thoughtful and constructive comments. In line with the Reviewers' comments, we have now carried out deeper analysis of our existing datasets, included novel data in the manuscript and expanded the discussion.

The main new analyses/data included are:

- Human eDNA sequencing and qPCR from room air samples;
- Human exome enrichment sequencing from water and sand eDNA samples. We had initially hoped to include these samples in the previous submission (sent for sequencing around the same time that we conducted our initial in-house MinION sequencing), but covid delays and management changes at the sequencing facility meant that the data hadn't been generated in time;
- Genetic ancestry assignment from all sample types (air, substrate, water);
- Fuller description of lengths obtained from human long read sequences (after consultation with forensic scientists);
- Enhanced disease-association mutation analysis.

Together with the analytical and experimental enhancements, we have implemented the other requested changes to the manuscript, which we feel have helped to further greatly improve and strengthen its impact.

All author initial responses are provided in blue text.
All changes to the main manuscript have been tracked.

Reviewers' comments:

Reviewer #2 (Remarks to the Author):

Thank you for the opportunity to review this paper again, and for your patience as I completed this review. As before, my comments are restricted to the ethical discussion in the article.

I think this paper is substantially improved. I appreciate the changes that the authors have made in response to reviewer comments. I have some additional comments, but they likely require minor, rather than major, revision.

First, on page 2, ll. 51-52, you note that human wastewater monitoring has already been repurposed to track other pathogens. This is true. But this recitation misses a forest for a single, and arguably the least objectionable, tree. Wastewater monitoring is also already being put to use to detect illicit drugs, antidepressants, stress markers, and alcohol consumption, among other things. There may be a risk that wastewater monitoring could be used to surveil for pregnancy-related hormones or abortion-related drugs, particularly in states that have now made abortion care unlawful. So in addition to noting the disease surveillance to which wastewater monitoring may be put, it may be worthwhile to flag broader uses already contemplated or in use. In addition to the sources cited here, you may wish to consider: Ram, N., Gable, L., & Ram, J.L. The Future of Wastewater Monitoring for the Public Health, *Univ. Rich. L. Rev.* 56(2022).

In line with this suggestion we have now included the following in the manuscript:

“Even for pathogens the legal and ethical implications of wastewater monitoring have not been adequately considered, and wastewater is also being utilized to detect illicit drugs, antidepressants, stress markers, and alcohol consumption⁴⁴. There may be a risk that wastewater monitoring could be used to survey for pregnancy-related hormones or abortion-related drugs, particularly in localities in which abortion care is unlawful.” Due to the manuscript already being over the allowed word limit we have tried to cover these additional aspects as concisely as possible, and have included the above text in the wastewater section of the discussion.

Second, in the discussion, I encourage you to attempt to be even clearer about scenarios where HGB may be neither beneficial nor exploitative. Surely there are some such uses (I previously suggested that seawater collected from a popular tourist beach is likely to be less ethically concerning than wastewater monitoring of a small, defined, stable population). Identifying these kinds of scenarios will help make clear that you are able to parse from the range of possible uses the genuinely problematic ones.

As suggested we have added clarification that differences in the level of ethical concerns are likely to exist between different study setups (including using the example tourist beach versus

25defined stable population the Reviewer provided) and have specifically noted that in some cases HGB may be neutral, rather than exploitive or beneficial, with different studies existing along a spectrum of concern. To further aid with clarity we have now also included a separate ‘benefits’ table (Table 2).

Third, at ll. 266-267, please reconsider whether to place forensic and criminal investigative applications under “benefits” as opposed to “worrying” applications. Solving crime is a good thing, to be sure. But exploiting involuntarily shed genetic information for investigative aims risks putting all of us under perpetual genetic surveillance in ways that may raise genuine questions about the appropriate limits of policing and surveillance.

We agree with the Reviewer. To further clarify the potential benefits, forensic and criminal investigative applications are included in the new benefits table (Table 2) and as noted are discrete from population surveillance.

The changes to Table 1 are much appreciated. This Table is now clear and easy to understand. Thank you, and for your previous suggestions in this regard.

Reviewer #3 (Remarks to the Author):

The basic premise of this paper is that 1) human eDNA may be commonly captured and sequenced either inadvertently (“by-catch”) or intentionally (“overt”) and 2) this potentially raises ethical and legal concerns. I think this premise is interesting and relevant to the field. Thank you to the authors for attempting to tackle this topic. I also want to acknowledge that the authors clearly took previous reviews seriously and made meaningful attempts to address the concerns raised there about contamination. I don’t have any comments specifically about their response to previous reviews. Thank you.

Thank you, we appreciate your comments and your acknowledgement that we addressed the previous review comments.

I do believe there are two important weaknesses to the paper. First, to be blunt, the scholarship in this paper is poor. The topics here are highly interdisciplinary, so I appreciate and am sympathetic to the fact that this is hard to do. Second, I believe that the authors at times have made claims about what is feasible that extend well beyond what is strictly demonstrated from their empirical data. I will address these two topics in comments one at a time.

Primary concern #1: Engagement with existing literature

26As noted by the authors, there are two different situations for human eDNA data: By-catch and intentional sampling of humans. Like the authors, I'll walk through each of these separately.

Intentional human sampling: One might intentionally capture and sequence human eDNA (either with enrichment or simply shotgun sequencing). In this case, you are conducting research on humans and your research needs to comply with any special ethical and legal considerations for research on human subjects. This would be “overt capture” (paper title) and relates to the Discussion in this manuscript on lines 228 – 267. In this discussion the authors cover a lot of potential ground for overt study of human eDNA. However, I feel like the authors fail to engage fully with recent literature on human medicine, forensic, and anthropology fields that they suggest innovations for.

Please note: We appreciate the Reviewer's constructive suggestions on further exploration of multidisciplinary links in the discussion and have endeavoured to expand these sections as much as was feasible. However, this was difficult within the scope of the paper with the previous manuscript already being at the journal word limit for primary research papers. We have now been granted an additional 500 words by the Editor, though much of this allotment has had to go towards describing our new data. We have included the additional reference types as suggested by the Reviewer, and have kept their exploration concise as necessitated by the word limit.

Human medicine

Line 243: Along with Reviewer #1, I'm skeptical of feasibility (reviewer comments on previous line #157). I think the authors in their revision are trying to suggest that, although pooled samples are difficult to use, you could get eDNA from a single individual at a time by sampling a specific hospital room or toilet flush (Line 248). In this case, you are attempting to sample a specific, knowable individual. From an ethics standpoint, I think you clearly should gain consent from that individual (like the authors of this paper did for their footprint work). In which case, why not just request a cheek swab (a superior sample type)? I can only imagine eDNA sampling being useful when you cannot gain consent because 1) the application is forensic, for a legal investigation or 2) the sample represents a pool of individuals. The authors in their response argue that this application “is likely achievable in the medium term”. Maybe, but the authors only cite a perspective paper (citation #1) and a paper on COVID-19 monitoring (citation #40; please note that there is also now a peer-reviewed version of this ms). My request of the authors: If there is evidence that “one could utilize wastewater eDNA-based sampling to correlate level of pathogens with abundance of susceptibility loci in a given population”, please cite at least one peer-reviewed, empirical demonstration.

As suggested by the Reviewer, we have now demonstrated from our own data that relevant information about human disease susceptibility loci can indeed be determined from eDNA samples (Fig. 5a, Supplemental Fig. 7a and Supplemental Tables 5 and 6).

Our air and water eDNA analysis from pooled individuals also shows that it is possible to obtain individual reads long enough for haplotyping or disease allele analysis, even from pooled samples, as well as from eDNA samples obtained from a single identifiable individual. Not only was this achievable from shotgun sequencing, but our exome enrichment shows that it is feasible to generate large volumes of human-specific reads from pooled eDNA samples. With shotgun, exome enrichment, or enrichment for other specific loci of interest it is feasible to generate human loci information from pooled samples, while quantification of pathogens from eDNA samples has been well established for both human and animal pathogens. This includes the simultaneous detection of host and pathogens from the same eDNA samples.

It should also be noted that the feasibility of human eDNA qPCR demonstrated in this manuscript shows that more cost-effective non-sequencing qPCR-based loci identification/quantification should be readily feasible too.

Citation #40: Thank you, we have updated the citation from the preprint to the published paper.

Citation #1 is a review paper, cited for brevity as we cannot cover all of the recent rapid primary research developments relating to pathogen detection from aquatic sources in the discussion of this research article. Other instances of pathogen monitoring have already been cited throughout the manuscript, and we have now explicitly added some of these references to this sentence also. Pathogen quantification and pathogen genomic surveillance have been robustly established for a range of pathogens. Combined with the ability to recover human genomic information from pooled eDNA samples reported in the revised manuscript, there is no technical roadblock to the design and implementation of susceptibility loci-focused studies.

Re cheek swabs: contact-free sampling has benefits over more conventional sampling for some applications, although of course not in all instances. The focus of such non-invasive eDNA-enabled sampling is not so much on inherited mutations (inherited diseases) where a single sampling event is often sufficient, but rather for diseases arising from spontaneous somatic mutations, such as cancer. As these mutations can occur throughout life there are initiatives to improve screening success by implementing more frequent automated routine biomarker monitoring, and these fall under the auspices of Precision Medicine and Connected Health. It is these physician-free approaches that would most benefit from human eDNA tools (clinical follow-up diagnosis recommendations are made based on the automated biomarker monitoring). In these cases, the most appropriate sample type can vary by the disease being screened for, as not all cells in the body harbour the biomarker mutations. For instance, air eDNA may be suitable for lung cancer detection while wastewater eDNA would be more suitable for colon and bladder cancers. Indeed, stool samples are already directly utilized for colon cancer screening, outperforming plasma-based screening.

Automatic monitoring systems for eDNA that are currently in development (such as the NS²) could be readily re-purposed in the near-term for such Connect Health human genetic biomarker applications. We have now made the linkage between human eDNA sampling and Connected Health continual biomarker monitoring more explicitly in the manuscript.

Forensics

The authors correctly identify eDNA sampling as potentially useful for forensic applications, but do not engage in any way with forensic genetics literature (only cited literature regarding forensics is the speculative discussion from a paper on naked mole rats). There is no engagement with literature on forensic genetics/eDNA (e.g., recent papers with clear connections like “Assessing the use of environmental DNA (eDNA) as a tool in the detection of human DNA in water”; *Journal of Forensic Sciences*). There is also no engagement with forensic genetic ethics, such as phenotyping which is a really important issue connected to racial issues that might arise if one were to “identify the geographic human populations” as the authors here suggest (dozens of papers exist on this topic). My request of the authors: Please deeply and meaningfully engage with existing forensic methods/ethics in the literature, or at least point readers to the most relevant discussions of this topic.

Thank you for this recent laboratory-based citation, which was only published shortly before our resubmission (after we had completed our literature review). We have now cited it and some other recent preprints in the discussion (including recent air eDNA advances), as well as forensic ethics reviews.

We also engaged with forensic scientists regarding the minimum read length they would consider informative for individual identification from eDNA data. In short, 16kb reads covering the entire mitochondrion would be sufficient for US missing person databases. Following that consultation we have now included more information on human aligning read lengths in the manuscript (Supplemental Table 4 and Supplemental Fig. 6a).

We also now explicitly demonstrate that human haplotyping and haplogrouping is possible from pooled eDNA samples for each sequencing method employed, Illumina shotgun, Nanopore shotgun and Illumina human exome enrichment (Fig. 5c).

Anthropology

The authors suggest that eDNA sampling might “identify undiscovered sacrificial sites”. There is a wealth of literature on ethical considerations when studying the genetic material of ancestral human populations. There is also a wealth of literature specifically on human ancient DNA, including soil samples (i.e., ancient eDNA). None of this literature is mentioned. My request to the authors: Please deeply and meaningfully engage with existing anthropology/ancient DNA

methods and ethics in the literature, or at least point readers to the most relevant discussions of this topic.

As suggested by the Reviewer we have now included additional references to ancient and contemporary DNA ethical considerations in the discussion section. For sacrificial site detection, quantitative approaches such as species-specific qPCR would likely be the analysis type of choice. While still having ethical implications, these are not as complex as those which arise if sequencing ancient DNA.

Human “by-catch”: One might also retrieve reads from humans unintentionally. The original researcher would clearly be violating basic research ethics if they “repurposed” these reads for research on human subjects. They may also face issues with posting this data publicly where other data users might “repurpose” these reads without the original research team’s consent. Thank you to the authors for articulating this. However, the authors do not acknowledge that human “by-catch” in ecological studies is not new or unique to eDNA sampling. Inadvertent (or even intentional) capture of humans has been described/discussed relative to wildlife camera trapping (e.g., DOI: 10.1002/2688-8319.12033) and eco-acoustic data (e.g., DOI: 10.1101/2022.02.08.479660). My request to the authors: Seek to make connections with similar “by-catch” issues using other methodologies and note if there are best practices which are directly applicable to eDNA sampling.

Thank you for the suggestion. For conciseness we tried to primarily focus on genetic comparisons, but do see the value in linking to non-genetic examples of human bycatch. We have included references to other types of conservation-orientated non-genetic bycatch in the discussion section, as suggested.

Although not my area of expertise, the discussion of potential nefarious application of human eDNA data seems better. Links to recent literature on genetic research in indigenous communities are relevant. The idea of using genetic material to screen for the presence of specific populations is indeed “chilling” (Line 306), but the authors seem to imply that there are nearly-fixed genetic differences that are diagnostic for ethnicity. I’m guessing this is not what the authors intend to communicate. A more nuanced message here might explain more explicitly that even if there is not a good scientific basis for using eDNA to infer ethnicity (or some other human attribute), we have historically seen pseudoscientific approaches used as a weapon (e.g., so-called “craniology” in the eugenics movement, Nazi racial “science”, etc). A possible, partial analogue with gender and disability ethical discussions would be non-invasive prenatal testing (i.e., genetic evidence of embryo sex or disability being used to make a pregnancy termination decision). I do note, again, that this is not my own area of expertise.

30Yes, we are referring to genetic ancestry markers, and have made this more explicit in the text. We have also included as suggested that minority groups and vulnerable individuals have historically been subjected to pseudoscientific-based persecution, as these issues could easily be convoluted in such a way.

Bioinformatics: None of the bioinformatic tools except for Galaxy appear to be cited. This includes FastQC, Trim GaloreI, Porechip, Bowtie2, StringTie, Samtools, minimap2, and mosdepth. Please cite the associated peer-reviewed papers for each. Also watch for your dependencies (e.g., I believe WIMP uses Centrifuge). Further, no parameter values for any programs are reported. I would minimally have expected to see parameter values listed (even if they are the current “default”) and summaries of intermediate and final bioinformatic results. Better would be rationales for parameter values selected and links to the supplement where multiple parameter values were assessed.

Apologies for any oversight; all tools have now been cited. All tools utilized are well-established, and since we did not generate or repurpose tools, the well-supported default parameters were appropriate (with exception of Sniffles v2.0.7, in which we lowered the sensitivity parameter to single split-read resolution via “minsupport=1”). We have also cross-validated the broad findings by utilizing a number of tools (e.g. alignment tools) on the same dataset, and by aligning to 3 different human reference genomes including the most current recently released T2T reference genome. While each tool/genome produces slight changes in the exact number of aligned reads, these are within expected inter-tool variation range and do not alter the findings of the manuscript. We have also included a new Supplemental Table 4, rationalizing all intentional human eDNA sample sequencing reporting for easy cross-reference. This involved re-analysis of all intentional human sequenced samples against the recent T2T human reference genome.

Primary concern #2: What can we really do with human eDNA?

Reviewer #1 made the observation “it might be difficult to actually make any individual-specific inferences” and I think this is a critical point. I do not agree that this point is “beyond the scope of the current study”. If it is not simply difficult but practically impossible to make individual-specific inferences from human by-catch eDNA, then many of the ethical/policy concerns for researchers are ameliorated. This is important: Should we as a scientific community require human subject research approval for most amplification-free eDNA studies? Failing to protect the rights of potential human subjects would be unfortunate, but so would creating regulatory barriers to research that are unnecessary if privacy risks are low.

We agree that this is an important topic for thoughtful analysis and debate, before the implementation or otherwise of new approval processes, hence conducting this study. We have deepened our analysis of the eDNA shotgun sequencing (population inferences and additional

mutational analysis) and included new air eDNA sampling and included novel human exome capture from eDNA. Together these data confirm our assertion that recovered human eDNA is of sufficient quality that the implications of both human genetic-bycatch and intentional human eDNA sampling should be considered by the research community and society more broadly. We do not determine what response or final regulatory frameworks are required. Rather, we encourage considered discussion and debate, informed by further experimentation. We believe that this is a discussion that should be initiated immediately, taking account not just of current technological capabilities but also future trends and the inevitable continued improvement in sequencing capacity and sophistication of computational analyses. We show that many human eDNA applications are already feasible, whether shotgun/targeted sequencing or qPCR is utilized.

Short-read data (Illumina): The authors report the highest observed proportion human-aligning reads for sample Inlet1 (Fort) with 42,030 reads per 10 million total reads. That is, 0.420% of sequenced bases were potentially human (probably less; as alluded to by the authors, much of what you can align is not necessarily human, particularly if repetitive regions were not masked). So 1X genome coverage for a single human genome (3.2 Gb) would require around 750 Gb of sequencing effort. That's $\frac{1}{4}$ of a NovaSeq 6000 run or an entire HiSeq 3000 run of sequencing effort for a single sample. Do I have my back-of-the-envelope math right? This is a lot of sequencing effort. Is it possible that, as long as sequencing efforts are not incredibly deep, inadvertently generating data that allows human individual identification is a non-issue? The authors cite two empirical papers as examples of “deep” sequencing of environmental samples. One is the Farrel et al study using the same samples as in this study. In Table S1, the highest sequencing effort sample has 304 million reads. If these are 150 bp reads, that's 45 Gb sequenced (i.e., about 4% of the sequencing effort you would minimally need to get 1X coverage for a single human genome based on this study). The other empirical paper (not really eDNA, but bulk arthropods) had a shotgun sequencing effort around 2 billion per sample (~300 Gb). Of course, to tease apart population or individual-level variation, you probably need much more than 1X mean coverage across an unknown individual pool size. The ability to generate human population demographic, individual identification etc. will depend as much on the sequencing approach applied as overall genome coverage. For example, coverage of entire genomes is not necessarily required to identify individuals. Rather, coverage of informative regions is the primary requirement. Particularly for long read sequencing technologies, even a single read could be sufficient to pinpoint an individual. As discussed in the manuscript, the minimum requirement for missing persons DNA databases in the US is a full mitochondrial sequence (i.e. only approx. 16kb). Even without targeting specific regions or using high molecular weight DNA extraction protocols our longest human nuclear aligning eDNA read

was 148 kb, and a human mitochondrial read of 16kb was obtained (Supplemental Table 3 and Supplemental Fig. 6a).

In addition to highlighting current capabilities, this manuscript is cognizant of the rapid and continued increase in sequencing output (a couple of decades ago it took billions of dollars and huge international effort to sequence a single human genome; now, an individual can sequence multiple human genomes in a day for less than \$1k each). Over the short, medium and long term the output of deep sequencers will continue to increase, while costs continue to fall and devices become even more portable, making pan-species shotgun metagenomics even more feasible and cost effective. This is especially important given the improvements and focus of sequencing companies on improved output and accuracy of long read sequencing capabilities. We show here that the quality of captured eDNA is already more than sufficient for human population and mutational inferences (even from a MinION, ONT's lowest output device). Furthermore, real-time sequencing enrichment (adaptive sequencing) on ONT devices, where only targets of interest are sequenced, with DNA sequences not aligning to target regions of interest being bypassed is allowing enrichment without requiring any additional laboratory manipulation or target capture probe-design. Therefore, regulatory and ethical considerations are currently needed, and the requirement for this discussion becomes even more imperative in light of the recent history and future trajectory of deep sequencing advances.

Long-read data: Figure 4 seems to imply that coverage with the Nanopore sequencing was extremely high for an environmental sample (4b; mean coverage on Chromosome 2 appears to be around 20X), but coverage for the Y Chromosome is extremely low. First, I'm surprised that in the supplement the Moultrie Creek B sample reports ~10% of reads being putatively human (20X more than the highest RPTM environmental sample on short reads). I suppose this speaks to how high of a human use area this is. However, how many contributing individuals do we think there are? If this 10% of DNA comes from potentially hundreds of individuals, can meaningful inference be made? Can the authors either 1) dig into the literature and show some evidence of feasibility or 2) build on their quick proof-of-concept "structural variant" list to show how one would derive some population inference from this information?

In summary, if this is not intended simply to be a "perspective" piece, I would like to see the authors spend more time engaging in empirical tests of what is feasible from the current dataset. We have expanded our datasets (including air eDNA and exome enrichment) and as suggested conducted more in-depth analysis of our original data. A new Supplemental Table 4 has been included to more readily compare coverage between sequenced intentional human samples. This includes reporting the human aligning data in bases to enable direct comparison between Illumina and Nanopore outputs. It also includes the percentage of all sequenced reads per library which were human aligning reads, for both shotgun and the newly added exome enriched

libraries. This new table reports alignment rate to the recently released human reference genome (T2T).

As suggested by the Reviewer we have conducted additional in-depth analysis (Fig. 4b, Fig. 5 and Supplemental Figs 6 and 7) of these samples, and of the new exome data and air eDNA data. This includes haplotype and haplogroup calling from every sample type to derive population inference (Fig 5c).

Minor comments

If the authors seek a revision, please spend some time ensuring that figures are readable at print scale and that the figure legends provide enough information to understand new figure types and terms (this is particularly relevant for Figure 4).

Figure 4 has been revised with a new graphing approach taken. Some of the loss of resolution occurred during the manuscript submission process. If accepted, post-review we will work with the production office to ensure all figures retain high print quality. Figure legends have been expanded and additional graphical approaches to genomic data employed.

Decision Letter, third revision:

20th February 2023

Dear Dave,

Thank you for submitting your revised manuscript "Human environmental DNA: inadvertent human genomic bycatch and intentional capture raises novel beneficial applications and ethical concerns" (NATECOLEVOL-220416342C). It has now been seen again by the original reviewers and their comments are below. The reviewers find that the paper has improved in revision, and therefore we'll be happy in principle to publish it in Nature Ecology & Evolution, pending minor revisions to satisfy the reviewers' final requests and to comply with our editorial and formatting guidelines.

Thank you again for your interest in Nature Ecology & Evolution. Please do not hesitate to contact me

34if you have any questions.

[REDACTED]

Reviewer #2 (Remarks to the Author):

Thank you for the opportunity to review this paper again, and for your patience as I completed this review. As before, my comments are restricted to the ethical discussion in the article.

I think this paper has once again substantially improved, and I thank the authors for grappling seriously with prior Reviewer comments. My comments here are limited.

I think this paper now works quite well as demonstration of how difficult it is to decouple “human subjects research” from other types of research. The authors have done a good job of identifying ethical lines of inquiry that arise from their work, but not attempting to resolve these ethical quandaries (which would be impossible to do in the space of a single article for the reasons I and other Reviewers have noted before).

One possible intervention the authors have not mentioned (and may reasonably decline to mention for reasons of space constraints) is that the federal funding agencies might expand the scope of federally-funded studies for which Certificates of Confidentiality are automatically issued. Certificates provide powerful protection against access to research data in a host of legal proceedings. Traditionally, NIH has taken the lead in administering and explaining Certificates, but this study suggests that NSF and other funding agencies may need to consider the applicability of Certificates to the research they fund as well.

Finally, I will once again caution against treating all forensic investigative uses as “beneficial.” As I noted before, while solving crime is a good thing, exploiting involuntarily shed genetic information for investigative aims risks putting all of us under perpetual genetic surveillance in ways that may raise genuine questions about the appropriate limits of policing and surveillance. I do not think this concern can be addressed merely by categorizing “forensic and criminal investigative applications” as a benefit while hiving off “population surveillance” as problematic. Even investigative methods that focus on the individual level may implicate privacy interests at the population level. For instance, as the Supreme Court has recognized, cell phone location tracking requires a judicially-authorized warrant because, even when that investigative method is targeted to a single individual, the information on which it relies is compiled against all of us on an ongoing basis. Similarly, because we all perpetually shed DNA, investigative methods that exploit such DNA sources (including eDNA) may be exploited to learn about or target any and all of us. But I will concede that this is a difficult distinction to set out in a non-law-focused article, and so I do not fault the authors for attempting to square this circle as they have.

Reviewer #3 (Remarks to the Author):

35Thank you to the authors for their careful revision of this manuscript.

Although I would still enjoy a manuscript with more room for engagement with existing literature in fields like human genetic forensics and non-genetic human by-catch in detection data, I now understand that the journal format may be limiting and that there simply is not enough room. I hope that my critique of the literature review before didn't come across as rude. I found the incorporation of additional data analysis since the previous version convincing. Thank you for making the effort. I don't have many comments to add.

Line 48: Generally, I think it's best to write "eDNA" as the noun and "eDNA sampling" or "eDNA research" as the verb. So on this line, I would write "Environmental DNA sampling...", where as simply "Environmental DNA" is appropriate on Line 56. Consider checking for consistent usage throughout.

Line 466: I'm not sure that I understand. How were standard curves run after eDNA samples? Don't they need to be on the same PCR plate?

Line 468: I found it kind of hard to keep track of which samples were collected when and where and which samples were used in which sequencing effort and subsequent analysis. There are two sets of samples: (1) Tank, ocean, and sand samples from publications #8 and #19, and (2) beach, estuary, ocean, private well, and air samples from this study. There is Illumina shotgun sequencing, which are all samples from publications #8 and #19. There is Nanopore sequencing from some of the samples specifically for this study (including air). There is qPCR data for samples from both studies. There is exome capture data using some samples specifically for this study. Is that right? I think breaking this down with a supplemental flow figure (e.g., Sankey) and/or reminding the reader within the Methods of sample types and sample sizes would be helpful. I had to bounce between the Methods, Results, and a few supplemental tables. Also, does the comment about tank sample pooling (Line 387) belong perhaps just before header "Intentional human samples" on Line 471?

Our ref: NATECOLEVOL-220416342C

9th March 2023

Dear Dr. Duffy,

36Thank you for your patience as we've prepared the guidelines for final submission of your Nature Ecology & Evolution manuscript, "Human environmental DNA: inadvertent human genomic bycatch and intentional capture raises novel beneficial applications and ethical concerns" (NATECOLEVOL-220416342C). Please carefully follow the step-by-step instructions provided in the attached file, and add a response in each row of the table to indicate the changes that you have made. Please also check and comment on any additional marked-up edits we have proposed within the text. Ensuring that each point is addressed will help to ensure that your revised manuscript can be swiftly handed over to our production team.

****We would like to start working on your revised paper, with all of the requested files and forms, as soon as possible (preferably within two weeks). Please get in contact with us immediately if you anticipate it taking more than two weeks to submit these revised files.****

In recognition of the time and expertise our reviewers provide to Nature Ecology & Evolution's editorial process, we would like to formally acknowledge their contribution to the external peer review of your manuscript entitled "Human environmental DNA: inadvertent human genomic bycatch and intentional capture raises novel beneficial applications and ethical concerns". For those reviewers who give their assent, we will be publishing their names alongside the published article.

Nature Ecology & Evolution offers a Transparent Peer Review option for new original research manuscripts submitted after December 1st, 2019. As part of this initiative, we encourage our authors to support increased transparency into the peer review process by agreeing to have the reviewer comments, author rebuttal letters, and editorial decision letters published as a Supplementary item. When you submit your final files please clearly state in your cover letter whether or not you would like to participate in this initiative. Please note that failure to state your preference will result in delays in accepting your manuscript for publication.

Cover suggestions

As you prepare your final files we encourage you to consider whether you have any images or illustrations that may be appropriate for use on the cover of Nature Ecology & Evolution.

Nature Ecology & Evolution has now transitioned to a unified Rights Collection system which will allow our Author Services team to quickly and easily collect the rights and permissions required to publish your work. Approximately 10 days after your paper is formally accepted, you will receive an email in providing you with a link to complete the grant of rights. If your paper is eligible for Open Access, our Author Services team will also be in touch regarding any additional information that may be required to arrange payment for your article.

Please note that *Nature Ecology & Evolution* is a Transformative Journal (TJ). Authors may publish their research with us through the traditional subscription access route or make their paper immediately open access through payment of an article-processing charge (APC). Authors will not be required to make a final decision about access to their article until it has been accepted. [Find out more about Transformative Journals](https://www.springernature.com/gp/open-research/transformative-journals)

Authors may need to take specific actions to achieve [compliance with funder and institutional open access mandates](https://www.springernature.com/gp/open-research/funding/policy-compliance-faqs). If your research is supported by a funder that requires immediate open access (e.g. according to [Plan S principles](https://www.springernature.com/gp/open-research/plan-s-compliance)) then you should select the gold OA route, and we will direct you to the compliant route where possible. For authors selecting the subscription publication route, the journal's standard licensing terms will need to be accepted, including [those licensing terms](https://www.nature.com/nature-portfolio/editorial-policies/self-archiving-and-license-to-publish) will supersede any other terms that the author or any third party may assert apply to any version of the manuscript.

[REDACTED]

[REDACTED]

Reviewer #2:

Remarks to the Author:

Thank you for the opportunity to review this paper again, and for your patience as I completed this review. As before, my comments are restricted to the ethical discussion in the article.

I think this paper has once again substantially improved, and I thank the authors for grappling seriously with prior Reviewer comments. My comments here are limited.

I think this paper now works quite well as demonstration of how difficult it is to decouple “human subjects research” from other types of research. The authors have done a good job of identifying ethical lines of inquiry that arise from their work, but not attempting to resolve these ethical quandaries (which would be impossible to do in the space of a single article for the reasons I and other Reviewers have noted before).

One possible intervention the authors have not mentioned (and may reasonably decline to mention for reasons of space constraints) is that the federal funding agencies might expand the scope of federally-funded studies for which Certificates of Confidentiality are automatically issued. Certificates provide powerful protection against access to research data in a host of legal proceedings. Traditionally, NIH has taken the lead in administering and explaining Certificates, but this study suggests that NSF and other funding agencies may need to consider the applicability of Certificates to the research they fund as well.

Finally, I will once again caution against treating all forensic investigative uses as “beneficial.” As I noted before, while solving crime is a good thing, exploiting involuntarily shed genetic information for investigative aims risks putting all of us under perpetual genetic surveillance in ways that may raise genuine questions about the appropriate limits of policing and surveillance. I do not think this concern can be addressed merely by categorizing “forensic and criminal investigative applications” as a benefit while hiving off “population surveillance” as problematic. Even investigative methods that focus on the individual level may implicate privacy interests at the population level. For instance, as the Supreme Court has recognized, cell phone location tracking requires a judicially-authorized warrant because, even when that investigative method is targeted to a single individual, the information on which it relies is compiled against all of us on an ongoing basis. Similarly, because we all perpetually shed DNA, investigative methods that exploit such DNA sources (including eDNA) may be exploited to learn about or target any and all of us. But I will concede that this is a difficult distinction to set out in a non-law-focused article, and so I do not fault the authors for attempting to square this circle as they have.

39Reviewer #3:

Remarks to the Author:

Thank you to the authors for their careful revision of this manuscript.

Although I would still enjoy a manuscript with more room for engagement with existing literature in fields like human genetic forensics and non-genetic human by-catch in detection data, I now understand that the journal format may be limiting and that there simply is not enough room. I hope that my critique of the literature review before didn't come across as rude. I found the incorporation of additional data analysis since the previous version convincing. Thank you for making the effort. I don't have many comments to add.

Line 48: Generally, I think it's best to write "eDNA" as the noun and "eDNA sampling" or "eDNA research" as the verb. So on this line, I would write "Environmental DNA sampling...", where as simply "Environmental DNA" is appropriate on Line 56. Consider checking for consistent usage throughout.

Line 466: I'm not sure that I understand. How were standard curves run after eDNA samples? Don't they need to be on the same PCR plate?

Line 468: I found it kind of hard to keep track of which samples were collected when and where and which samples were used in which sequencing effort and subsequent analysis. There are two sets of samples: (1) Tank, ocean, and sand samples from publications #8 and #19, and (2) beach, estuary, ocean, private well, and air samples from this study. There is Illumina shotgun sequencing, which are all samples from publications #8 and #19. There is Nanopore sequencing from some of the samples specifically for this study (including air). There is qPCR data for samples from both studies. There is exome capture data using some samples specifically for this study. Is that right? I think breaking this down with a supplemental flow figure (e.g., Sankey) and/or reminding the reader within the Methods of sample types and sample sizes would be helpful. I had to bounce between the Methods, Results, and a few supplemental tables. Also, does the comment about tank sample pooling (Line 387) belong perhaps just before header "Intentional human samples" on Line 471?

Author Rebuttal, third revision:

40Response to Reviewers' comments

We would like to once again thank the Reviewers for their careful consideration of the paper and their constructive comments throughout the entire review process. We believe the manuscript has benefited greatly from the reviewers' insights and input. We have now implemented all comments from this round of review.

Author responses are in blue font.

Reviewer #2 (Remarks to the Author):

Thank you for the opportunity to review this paper again, and for your patience as I completed this review. As before, my comments are restricted to the ethical discussion in the article.

I think this paper has once again substantially improved, and I thank the authors for grappling seriously with prior Reviewer comments. My comments here are limited.

I think this paper now works quite well as demonstration of how difficult it is to decouple "human subjects research" from other types of research. The authors have done a good job of identifying ethical lines of inquiry that arise from their work, but not attempting to resolve these ethical quandaries (which would be impossible to do in the space of a single article for the reasons I and other Reviewers have noted before).

One possible intervention the authors have not mentioned (and may reasonably decline to mention for reasons of space constraints) is that the federal funding agencies might expand the scope of federally-funded studies for which Certificates of Confidentiality are automatically issued. Certificates provide powerful protection against access to research data in a host of legal proceedings. Traditionally, NIH has taken the lead in administering and explaining Certificates, but this study suggests that NSF and other funding agencies may need to consider the applicability of Certificates to the research they fund as well.

This is an interesting point, but, as the Reviewer indicates, due to space constraints it is not one we can follow up on here. Future discussions should consider this point, as well as comparisons from other jurisdictions where similar solutions may also be in place.

Finally, I will once again caution against treating all forensic investigative uses as "beneficial." As I noted before, while solving crime is a good thing, exploiting involuntarily shed genetic information for investigative aims risks putting all of us under perpetual genetic surveillance in ways that may raise genuine questions about the appropriate limits of policing and surveillance. I do not think this concern can be addressed merely by categorizing "forensic and criminal investigative applications" as a benefit

while hiving off “population surveillance” as problematic. Even investigative methods that focus on the individual level may implicate privacy interests at the population level. For instance, as the Supreme Court has recognized, cell phone location tracking requires a judicially-authorized warrant because, even when that investigative method is targeted to a single individual, the information on which it relies is compiled against all of us on an ongoing basis. Similarly, because we all perpetually shed DNA, investigative methods that exploit such DNA sources (including eDNA) may be exploited to learn about or target any and all of us. But I will concede that this is a difficult distinction to set out in a non-law-focused article, and so I do not fault the authors for attempting to square this circle as they have.

To further highlight the potential positive and negative forensics implications we have now also included investigative applications in the potential problematic table (Table 1):

- Genetic surveillance – Potential for involuntarily genetic surveillance from investigative applications, including recovery of bystander shed genetic information, and / or intentional overreach.

As well as its original appearance in the benefits table (Table 2):

- Forensic and criminal investigative applications, to aid in solving crime. Air eDNA holds particular novel promise.

Reviewer #3 (Remarks to the Author):

Thank you to the authors for their careful revision of this manuscript.

Although I would still enjoy a manuscript with more room for engagement with existing literature in fields like human genetic forensics and non-genetic human by-catch in detection data, I now understand that the journal format may be limiting and that there simply is not enough room. I hope that my critique of the literature review before didn't come across as rude. I found the incorporation of additional data analysis since the previous version convincing. Thank you for making the effort. I don't have many comments to add.

Line 48: Generally, I think it's best to write “eDNA” as the noun and “eDNA sampling” or “eDNA research” as the verb. So on this line, I would write “Environmental DNA sampling...”, where as simply “Environmental DNA” is appropriate on Line 56. Consider checking for consistent usage throughout.

Changed as suggested.

Line 466: I'm not sure that I understand. How were standard curves run after eDNA samples? Don't they need to be on the same PCR plate?

Not necessarily. A standard curve plate can be run separately and the obtained values used to calculate the quantity of DNA from other plates based on the Cq values obtained. While this is not our common practice (we normally run a standard curve on every plate), given earlier reviewer comments re. the potential source of the human DNA, we felt that in this case it was most appropriate to hold off on generating the standard curve until all eDNA samples had been processed to confirm that no contamination of eDNA samples was possible from the human cell line DNA used to generate the standard curve.

Line 468: I found it kind of hard to keep track of which samples were collected when and where and which samples were used in which sequencing effort and subsequent analysis. There are two sets of samples: (1) Tank, ocean, and sand samples from publications #8 and #19, and (2) beach, estuary, ocean, private well, and air samples from this study. There is Illumina shotgun sequencing, which are all samples from publications #8 and #19. There is Nanopore sequencing from some of the samples specifically for this study (including air). There is qPCR data for samples from both studies. There is exome capture data using some samples specifically for this study. Is that right? I think breaking this down with a supplemental flow figure (e.g., Sankey) and/or reminding the reader within the Methods of sample types and sample sizes would be helpful. I had to bounce between the Methods, Results, and a few supplemental tables. Also, does the comment about tank sample pooling (Line 387) belong perhaps just before header "Intentional human samples" on Line 471?

The reviewer is correct, with the exception of "There is qPCR data for samples from both studies." All qPCRs were from intentional human eDNA samples generated for this study.

Thank you for the suggestion re. sample schematic; we have now included such a schematic (Supplemental Fig. 1).

Pooling moved as suggested, and methods re. the 3 sample types (water, sand and air) have been further clarified.

Final Decision Letter:

29th March 2023

Dear Dave,

We are pleased to inform you that your Article entitled "Inadvertent human genomic bycatch and intentional capture raises novel beneficial applications and ethical concerns with environmental DNA.", has now been accepted for publication in Nature Ecology & Evolution.

Over the next few weeks, your paper will be copyedited to ensure that it conforms to Nature Ecology and Evolution style. Once your paper is typeset, you will receive an email with a link to choose the appropriate publishing options for your paper and our Author Services team will be in touch regarding any additional information that may be required

You will not receive your proofs until the publishing agreement has been received through our system

Due to the importance of these deadlines, we ask you please us know now whether you will be difficult to contact over the next month. If this is the case, we ask you provide us with the contact information (email, phone and fax) of someone who will be able to check the proofs on your behalf, and who will be available to address any last-minute problems . Once your paper has been scheduled for online publication, the Nature press office will be in touch to confirm the details.

Acceptance of your manuscript is conditional on all authors' agreement with our publication policies (see www.nature.com/authors/policies/index.html). In particular your manuscript must not be published elsewhere and there must be no announcement of the work to any media outlet until the publication date (the day on which it is uploaded onto our web site).

Please note that *Nature Ecology & Evolution* is a Transformative Journal (TJ). Authors may publish their research with us through the traditional subscription access route or make their paper immediately open access through payment of an article-processing charge (APC). Authors will not be required to make a final decision about access to their article until it has been accepted. [Find out more about Transformative Journals](https://www.springernature.com/gp/open-research/transformative-journals)

Authors may need to take specific actions to achieve [compliance](https://www.springernature.com/gp/open-research/funding/policy-compliance-faqs) with funder and institutional open access mandates. If your research is supported by a funder that requires immediate open access (e.g. according to [a](https://www.springernature.com/gp/open-research/funding/policy-compliance-faqs)

44[Plan S principles](https://www.springernature.com/gp/open-research/plan-s-compliance)) then you should select the gold OA route, and we will direct you to the compliant route where possible. For authors selecting the subscription publication route, the journal's standard licensing terms will need to be accepted, including <https://www.nature.com/nature-portfolio/editorial-policies/self-archiving-and-license-to-publish>. Those licensing terms will supersede any other terms that the author or any third party may assert apply to any version of the manuscript.

We welcome the submission of potential cover material (including a short caption of around 40 words) related to your manuscript; suggestions should be sent to Nature Ecology & Evolution as electronic files (the image should be 300 dpi at 210 x 297 mm in either TIFF or JPEG format). Please note that such pictures should be selected more for their aesthetic appeal than for their scientific content, and that colour images work better than black and white or grayscale images. Please do not try to design a cover with the Nature Ecology & Evolution logo etc., and please do not submit composites of images related to your work. I am sure you will understand that we cannot make any promise as to whether any of your suggestions might be selected for the cover of the journal.

You can generate the link yourself when you receive your article DOI by entering it here: <http://authors.springernature.com/share>.

[REDACTED]

P.S. Click on the following link if you would like to recommend Nature Ecology & Evolution to your librarian <http://www.nature.com/subscriptions/recommend.html#forms>

** Visit the Springer Nature Editorial and Publishing website at http://editorial-jobs.springernature.com?utm_source=ejp_NEcoE_email&utm_medium=ejp_NEcoE_email&utm_campaign=ejp_NEcoE for more information about our career opportunities. If you have any questions please click [here](mailto:editorial.publishing.jobs@springernature.com).**